# SENTINEL: STAGEWISE INTEGRITY VERIFICATION FOR PIPELINE PARALLEL DECENTRALIZED TRAINING

## ABSTRACT

Decentralized training introduces critical security risks when executed across untrusted, geographically distributed nodes. While existing Byzantine-tolerant literature addresses data parallel (DP) training through robust aggregation methods, pipeline parallelism (PP) presents fundamentally distinct challenges. In PP, model layers are distributed across workers where the activations and their gradients flow between stages rather than being aggregated, making traditional DP approaches inapplicable. We propose SENTINEL, a verification mechanism for PP training *without computation duplication*. SENTINEL employs lightweight momentum-based monitoring using exponential moving averages (EMAs) to detect corrupted inter-stage communication. Unlike existing Byzantine-tolerant approaches for DP that aggregate parameter gradients *across replicas*, our approach verifies sequential activation/gradient transmission *between layers*. We provide theoretical convergence guarantees for this new setting that recovers classical convergence rates when relaxed to standard training. Experiments demonstrate successful training of billion-parameter LLMs across untrusted distributed environments with hundreds of workers while maintaining model convergence and performance.

## 1 INTRODUCTION

Large Language Models (LLMs) have fundamentally reshaped artificial intelligence, demonstrating exceptional performance across diverse tasks (OpenAI, 2023; Yang et al., 2024; Jiang et al., 2024; Dubey et al., 2024; MetaAI, 2025; Bi et al., 2024; DeepSeek-AI et al., 2025). Training state-of-the-art LLMs, however, requires substantial computational resources (reportedly tens of thousands of co-located GPUs for models like GPT-4 (Walker II, 2023), Llama-4 (MetaAI, 2025), Qwen2.5 (Yang et al., 2024), etc.) with corresponding energy and financial costs. This has motivated research into decentralized training approaches to broaden participation in LLM development (Ryabinin et al., 2021; Yuan et al., 2022; Ryabinin et al., 2023). Decentralized training, which extends distributed training to trustless settings, allows independent collaborators to pool their computational resources, potentially over large distances, to develop models without relying on massive centralized infrastructure.

Decentralized training of LLMs over networks of interconnected devices is made possible through two primary parallelization approaches: data and pipeline parallelism. Data parallelism (DP) (Li et al., 2020; Zhao et al., 2023) distributes different batches of training data across workers nodes, however requires each node to fit the entire model which is not practical for billion-parameter models in decentralized settings. Pipeline parallelism (PP) (Krizhevsky et al., 2017; Huang et al., 2019) partitions the model across stage-wise across worker nodes, with each responsible for distinct model stages (groups of layers), but requires high-bandwidth connections and suffers from node dropout. Combining these complementary approaches reduces the size limitation of DP and the vulnerability of PP, and have enabled frameworks such as SWARM (Ryabinin et al., 2023) to train billion-parameter LLMs through internet-scale communication among distributed nodes. By leveraging these parallelization techniques, such frameworks aim to achieve high node utilization while minimizing bandwidth requirements, hoping to make large-scale model training more widely accessible.

While optimizing communication bandwidth and fault tolerance have been the primary focus in decentralized training research (Ryabinin et al., 2021; Douillard et al., 2023; 2025; Ajanthan et al., 2025), the success of incentive-driven decentralized training critically hinges on the integrity and trustworthiness of participating nodes (Lu et al., 2024).

Malicious actors in DP configurations can corrupt global updates through parameter gradient poisoning, while in PP (layer-wise model parallelism), adversaries can sabotage intermediate activations or activation gradients between model stages. Such vulnerabilities underscore the need for robust mechanisms in a decentralized PP training setting. Traditional Byzantine-tolerant methods, designed designed to prevent simpler DP threat models (Malinovsky et al., 2024; Gorbunov et al., 2022; Mhamdi et al., 2018) fail to address the cascading failures induced by partitioned model execution in pipeline parallel (Lu et al., 2024).

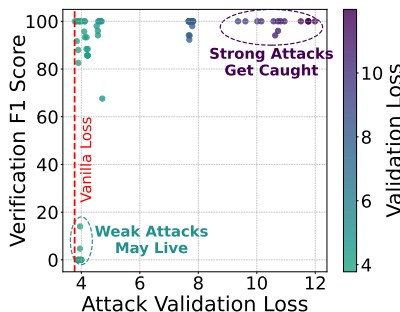

Figure 1: Scatter plot of F1-scores vs. validation loss for more than 75 different attack setups. Strong attacks (higher loss) are caught more often (high F1-score), while weak attacks may slip through. Our verification method thus catches the most harmful attacks that would disrupt training.

In this paper, we provide the first comprehensive exploration of secure and verifiable PP decentralized training by identifying and addressing vulnerabilities unique to this setting. We formalize a suite of malicious attacks tailored for this setting, for which traditional checkpoint-based verification is ineffective (Arun et al., 2025). To counter this, we propose a lightweight verification mechanism using verifier nodes, trusted intermediaries placed between stages, that continuously monitor computational integrity without requiring full model replication or impeding training throughput.

Our method, called SENTINEL, implements a momentum-based anomaly detection system that tracks exponential moving averages (EMAs) of activations and gradients across pipeline stages. At each stage of the model, verifier nodes compute statistical divergence metrics between observed signals and their EMA baselines. Deviations exceeding adaptively calibrated thresholds, determined via inter-quartile range (IQR) analysis, are flagged for potential malicious activity. This lightweight verification introduces minimal computational overhead while enabling early detection of both gradient and activation tampering attacks. Empirical evaluations demonstrate that our system successfully detects and mitigates various attacks in decentralized PP setups scaling beyond hundreds of workers, maintaining training integrity and convergence stability despite the presence of malicious participants.

The primary contributions of our work are summarized as follows:

- We present the first comprehensive study of vulnerabilities unique to decentralized training with hybrid data–pipeline parallelism, and introduce a suite of training-interruption attacks that serve as benchmarks for evaluating the security of future systems.
- We propose a lightweight verification method, dubbed SENTINEL, that leverages momentum-based monitoring at verifier nodes. Our theoretical analysis demonstrates that undetected malicious workers have a negligible impact on the convergence properties (see Fig. 1).
- We perform extensive experiments in distributed settings involving hundreds of workers, validating the effectiveness of our verification framework in mitigating malicious behaviors within realistic decentralized training scenarios by achieving consistently high ($> 90\%$) F1 scores.
- We integrate our method with SWARM parallelism to demonstrate its remarkable versatility in real-world decentralized training ecosystems.

## 2 PROBLEM STATEMENT

In this section, we outline our hybrid DP-PP architecture and threat model for decentralized training, focusing on malicious worker behavior. Additional vulnerabilities are detailed in App. B.

### 2.1 THREAT MODEL

We consider a distributed pipeline parallel neural network composed of $p$ stages and $n$ worker nodes (see Fig. 2b & c). The network outputs $\boldsymbol{y} = F(\boldsymbol{x}) = f_p \circ f_{p-1} \circ \cdots \circ f_1(\boldsymbol{x})$. At iteration $t$ the parameters are $\boldsymbol{\theta}_t = (\boldsymbol{\theta}_t^{(1)}, \ldots, \boldsymbol{\theta}_t^{(p)}) \in \mathbb{R}^{D_{\text{total}}}$, where each stage $s$ has parameters $\boldsymbol{\theta}_t^{(s)} \in \mathbb{R}^{D_s}$ and $D_{\text{total}} = \sum_{s=1}^{p} D_s$. Each stage $s$ is is replicated across $d_s$ worker nodes operating in parallel, thus $n = \sum_{s=1}^{p} d_s$. Parameters $\boldsymbol{\theta}^{(s)}$ are shared across stage replicas, however

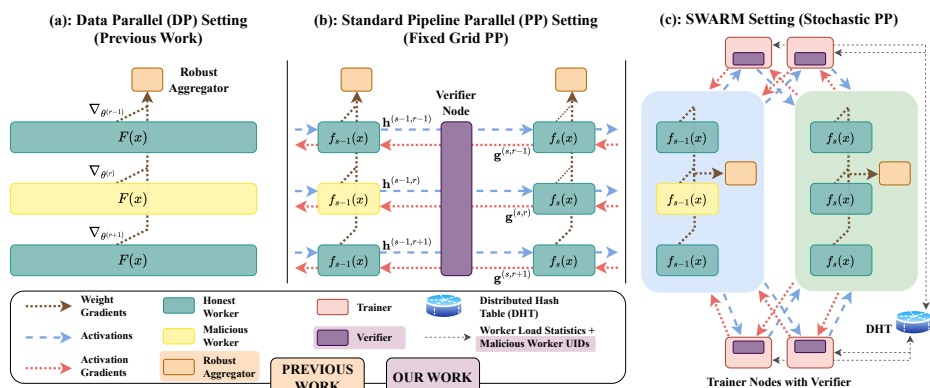

Figure 2: Distributed threat models. **(a)** In DP, workers hold full model replicas and only send *weight gradients*. **Traditional Byzantine-tolerant methods consider this case and use robust aggregation**. (**b & c**) **The threat model considered in this paper** (see Sec. 2.1). In PP, workers hold individual layers, and send intermediate *activations $h$* and *activation gradients $g$*, thus corruptions directly affect other workers. **(b)** In the standard setting there is a fixed grid of pipeline stages and data parallel replicas, and communication is routed through our verifier nodes. **(c)** In the SWARM setting designed for decentralized setups, data is stochastically routed by trainer nodes. Workers send their computations ($h$ and $g$) to trainers, who then route them to an available worker in the next stage. Our proposed verifiers can seamlessly be added on top of these trainer nodes (App. G).

each replica processes a distinct mini-batch. Thus, worker $(s, r)$ computes replica-specific activations $h^{(s,r)} = f_s(h^{(s-1,r)}; \theta^{(s)}) \in \mathbb{R}^m$ on the forward pass, and replica-specific activation gradients $g^{(s,r)} = \nabla_{h^{(s-1,r)}} \mathcal{L}(\theta)$ on the backward pass where $\mathcal{L}(\theta)$ is the training loss.

In current decentralized PP training frameworks such as SWARM (Ryabinin et al., 2023), workers exchange activations and activation gradients between pipeline stages without verification. Unlike federated learning (McMahan et al., 2017) where attacks primarily target weight gradient poisoning, PP setups are vulnerable to training-interruption attacks, where malicious workers can silently disrupt training by sending corrupted signals. This is particularly challenging because corruptions in early-stage workers *may only become apparent in workers of later layers*: errors can amplify due to model non-linearities and only surpass detection thresholds after several stages, allowing attackers to avoid detection and potentially flagging honest workers due to this *cascading effect*. For a comprehensive comparison between the DP and PP setup, please refer to Q0 in App. A.

To address these vulnerabilities, we introduce dedicated "verifier nodes" as trusted intermediaries for lightweight verification (see Fig. 2b). These verifier nodes intercept and validate all training signals exchanged between stages, operating efficiently even on CPU hardware with minimal overhead. Their key advantage is localizing malicious behavior at specific stages, preventing cascading corruption effects inherent to pipeline parallelism. In frameworks like SWARM(Ryabinin et al., 2023), verifier nodes are a simple extension to "trainer nodes" that handle orchestration between workers (see Fig. 2c, App. G). Formally, our threat model with verifier nodes is defined as follows:

**Definition 1** (Data and Pipeline Parallel Threat Model). *Consider a neural network trained in a distributed system with the PP setup described above. We position trusted "verifier" nodes between consecutive stages, through which all communication is routed and thus can be validated, and our goal is to detect and exclude malicious participants. Explicitly, let $B_s \subset \{1, 2, \ldots, d_s\}$ be the subset of malicious worker nodes at stage $s$, with the fraction of malicious workers defined as $\gamma_s = |B_s|/d_s$, and we assume $\gamma_s < 1/2$. Our objective is to detect the malicious subset $B_s$.* [1]

In our threat model, malicious actors can employ various strategies to disrupt training. Next, we introduce these attacks based on their computational requirements and potential impact on the training.

**Attack Variants.** Let $x$ represent any activation or gradient vector potentially manipulated by a compromised node, with $\hat{x}$ denoting its manipulated version. We consider the following attacks:

- **Constant Attack:** The attacker submits a constant vector without performing the assigned computation, i.e., $\hat{x} = c$, such as vectors of negative ones or zeros.

---

[1] We assume the first and last layers are managed by honest workers, as they process user data and compute the loss and thus require protection. We also assume that the malicious workers cannot collude with each other.

- **Random Value Attack:** The attacker replaces vector elements with values randomly drawn from a standard normal distribution: $\hat{\boldsymbol{x}} \sim \mathcal{N}(\mathbf{0}, I)$.
- **Scaling Attack:** The attacker scales the vector by a factor $\alpha$, i.e., $\hat{\boldsymbol{x}} = \alpha\boldsymbol{x}$.
- **Random Sign Attack:** The attacker flips each vector element's sign with probability $p$.
- **Bias Addition Attack:** The attacker introduces random noise to the vector: $\hat{\boldsymbol{x}} = \boldsymbol{x} + \boldsymbol{\epsilon}$, where $\boldsymbol{\epsilon} \sim \mathcal{N}(\mathbf{0}, \sigma^2 I)$. For stealthier attacks, $\sigma$ can be chosen to match the original vector's magnitude.
- **Delay Attack:** The attacker use past values: $\hat{\boldsymbol{x}}_t = \boldsymbol{x}_{t-k}$ where $k$ denotes the delay steps
- **Invisible Noise Attack:** Inspired by the ALIE attack (Baruch et al., 2019), the attacker replaces benign values with statistically subtle boundary values: $\hat{\boldsymbol{x}} = \boldsymbol{\mu} + z_{\max}(\boldsymbol{\sigma} \odot \boldsymbol{\epsilon})$, where $\boldsymbol{\mu}$ is the original vector's mean, $\boldsymbol{\sigma}$ its element-wise standard deviation, $z_{\max} = \sqrt{2} \cdot \text{erfinv}(2p - 1)$, with $p = 1 - \alpha$ being the quantile threshold, erfinv is the inverse error function, and $\boldsymbol{\epsilon} \sim \mathcal{N}(\mathbf{0}, I)$.

## 3 MOMENTUM-BASED VERIFICATION OF WORKER NODES

Vanilla pipeline parallelism remains vulnerable to corrupted communications in both forward and backward propagation. When malicious workers inject corrupted activations or activation gradients, the effects can cascade through the network, potentially compromising model convergence with minimal detectability. Naïve methods like full computation duplication (Rajput et al., 2019; Lu et al., 2024) guarantee detection but reduce training throughput, while random sampling verification fails as some attacks that can damage training within just a few iterations (see Fig. 7).

We introduce SENTINEL: a lightweight, statistically principled verification mechanism that leverages EMA of activations and their gradients to establish reliable reference points for detecting anomalous behavior. We design an algorithm to adaptively set the thresholds of our anomaly detection tests using the IQR. Under relaxed assumptions, we analytically prove that undetected corrupted worker nodes under our verification framework have negligible impact on final model convergence.

### 3.1 PROPOSED METHOD

**Motivation.** The key insight driving our approach is that in healthy distributed training scenarios, each worker's activations and gradients should exhibit statistical consistency with the overall population. Existing work duplication methods such as Lu et al. (2024) would require significant computational resources as they need to allocate half of their resources for work verification. EMAs offer three critical advantages as the foundation for our proposed lightweight verification:

- **Computational Efficiency**: Computing and updating EMA statistics requires minimal computation and memory overhead ($\mathcal{O}(m)$ complexity where $m$ is the activation/gradient size), making it suitable for resource-constrained verifier nodes.
- **Temporal Smoothing**: The EMA naturally smooths out mini-batch noise while capturing the underlying distribution of legitimate worker outputs, creating a robust reference point.
- **Adaptivity to Training Dynamics**: As training distributions shift, the EMA automatically adjusts to these shifts while remaining resistant to abrupt deviations from malicious workers.

Thus, we design SENTINEL to contain four key components: (1) using EMAs as statistical reference points, (2) selecting appropriate distance measures for deviation detection, (3) implementing adaptive thresholds for anomaly detection, and (4) handling cascading effects in the distributed architecture. Below, we elaborate on each of these components. For a detailed step-by-step overview of our approach, please refer to Alg. 1 in the Appendix. Furthermore, we refer the interested reader to App. G for an overview of how SENTINEL gets integrated in SWARM (Ryabinin et al., 2023).

**Exponential Moving Average as Reference Point.** We leverage the EMA of activations and gradients at each layer as a statistical reference point to detect deviations. Since the EMA serves as a robust approximation of the expected value (Robbins & Monro, 1951), it is effective for detecting anomalies. For non-malicious actors, we only expect small deviations since the optimization trajectory typically remains smooth. Formally, each verifier node maintains a running EMA of activations:

$$\boldsymbol{m}_t^{(s)}(\boldsymbol{h}) = \beta_h \boldsymbol{m}_{t-1}^{(s)}(\boldsymbol{h}) + (1 - \beta_h) \frac{1}{d_s} \sum_{r=1}^{d_s} \boldsymbol{h}_t^{(s,r)}, \tag{1}$$

where $m_t^{(s)}(h)$ denotes the EMA of activations at stage $s$, and $\beta_h \in [0,1)$ is the decay rate. **A similar equation is used to capture the EMA of gradients $m_t^{(s)}(g)$ with decay rate** $\beta_g \in [0,1)$. To establish a reliable initial estimate, we employ a "warm-up" phase with only honest workers, during which the EMA statistics are collected and stabilized before verification begins (for additional explanation, please see Q4 in App. A.) After the warm-up period, each time workers submit new signals, the verifier conducts a lightweight statistical test comparing these signals with the established EMA. The deviation determines whether a worker is flagged as malicious. Formally, for activation $h_t^{(s,r)}$ submitted by worker $(s,r)$, the verifier calculates:

$$\Gamma_t^{(s,r)} := \Omega\left(h_t^{(s,r)}, m_{t-1}^{(s)}(h)\right), \tag{2}$$

where $\Omega(\cdot, \cdot)$ is a suitable distance measure. A worker is flagged as malicious if $\Gamma_t^{(s,r)} > \tau$, and we skip updating the EMA to maintain verification integrity. We use a similar detector for $g_t^{(s,r)}$.

**Choice of Distance Measure $\Omega$.** The distance function critically impacts verification sensitivity. Rather than using a single metric, we employ a collection of metrics $\mathcal{M}$, including *mean absolute difference*, *normalized Euclidean distance*, *sliced Wasserstein distance*, and *sign flip ratio*, to robustly detect various attack types. These metrics capture different aspects of distributional shifts that may indicate malicious behavior:

- **Mean Absolute Difference ($L_1$)** measures the average absolute deviation between signals: $\Omega_{L_1}(h_t^{(s,r)}, m_{t-1}^{(s)}(h)) = \mathbb{E}[\|h_t^{(s,r)} - m_{t-1}^{(s)}(h)\|_1]$, where $\mathbb{E}[\cdot]$ denotes the expectation over all features.
- **Normalized Euclidean Distance ($L_2$)** computes the squared difference between whitened representations:$\Omega_{L_2}(h_t^{(s,r)}, m_{t-1}^{(s)}(h)) = \mathbb{E}[\|\bar{h}_t^{(s,r)} - \bar{m}_{t-1}^{(s)}(h)\|^2]$,where $\bar{x}$ denotes the whitened (z-scored) version of $x$.
- **Sign Flip Ratio (SFR)** quantifies the fraction of coordinates with opposing signs, bounded in $[0,1]$: $\Omega_{SFR}(h_t^{(s,r)}, m_{t-1}^{(s)}(h)) = \mathbb{E}[\mathbf{1}(\text{sign}(h_t^{(s,r)}) \neq \text{sign}(m_{t-1}^{(s)}(h)))]$.
- **Sliced Wasserstein Distance (SW)** approximates the Wasserstein distance through random projections, i.e., $\Omega_{SW}(h_t^{(s,r)}, m_{t-1}^{(s)}(h)) = \mathbb{E}_{u \sim \mathcal{S}^{d-1}}[W_1((h_t^{(s,r)})_{\#}u, (m_{t-1}^{(s)}(h))_{\#}u)]$, where $W_1$ is the 1-Wasserstein distance, and $(\cdot)_{\#}\theta$ denotes the projection onto the random unit vector $u$ from the unit sphere $\mathcal{S}^{d-1}$.

We track all these metrics for each stage throughout training. The same metrics are used to measure the deviations between submitted gradients $g_t^{(s,r)}$ and their momentum $m_{t-1}^{(s)}(g)$. A worker is flagged if it exceeds the threshold for any metric. The diverse set of metrics provides robustness against various attack vectors, as different attacks may manifest in different statistical properties. Neural network-based distance measures could also be a promising candidate but left for future work. For an ablation study on the impact of each metric on the final performance, please refer to App. F.3.

**Automatic Threshold $\tau$ Updates.** Each distance metric requires a threshold $\tau$ to flag anomalous behavior. During the initial "warm-up" period mentioned earlier, we collect valid deviations for each metric at every stage and compute the IQR. We then use Tukey's fences (Tukey, 1977) to establish valid deviation bounds. In particular, let $\mathcal{H}_l^s$ denote the history of valid deviations over the past $l$ iterations collected from all $d_s$ workers at stage $s$. Let $q_1$, $q_2$, and $q_3$ be the 25th, 50th (median), and 75th percentiles of these deviations. We define our test statistic as:

$$\text{if} \quad |\Gamma_t^{(s,r)} - q_2| \geq k(q_3 - q_1) \Rightarrow \text{flag node } (s,r) \text{ as malicious} \tag{3}$$

where $k = 1.5$ is the conventional choice (Tukey, 1977). We choose to adaptively adjust $k$ through an iterative process that widens or narrows thresholds to maintain a chosen false positive rate (e.g., $<$ 1%). This dynamic threshold continuously incorporates new benign deviations into the historical window, enabling the verification system to automatically adapt to evolving data changes throughout training (see Fig. 12). The details of this method is given in Alg. 5 in the Appendix.

**Handling Cascading Effects.** In our PP-based distributed architecture, corrupted activations from an early stage can affect downstream workers, potentially causing misclassification of honest nodes. We address this with two complementary mechanisms:

1. **Bottom-up Malicious-Node Identification**: When a worker at stage $s$ is flagged as malicious, the verifier notifies all downstream verifiers to pause their deviation statistics for the affected

mini-batch and label subsequent anomalies as "tainted by upstream." To maintain uninterrupted training, in the backward pass verifiers send the stored gradient EMA, enabling parameter updates without revealing any behavioral change. For more information see App. D.2 and Fig. 5.

2. **Violation Counter with Forgiveness**: Rather than banning a worker after one deviation, each verifier maintains a violation counter. Severe deviations ($\times 100$ above the threshold) result in immediate bans, while milder ones increment the counter by one. A worker is banned after $c$ violations, but the counter decrements after $T_{\text{forgiveness}}$ consecutive clean steps, allowing recovery from transient anomalies. We use $c = 5$ and $T_{\text{forgiveness}} = 100$ in our experiments.

## 3.2 THEORETICAL ANALYSIS

To complement our practical approach, we provide theoretical guarantees under relaxed conditions, analyzing (1) the convergence behavior of the distributed training under bounded malicious perturbations, and (2) the conditions under which our system can maintain an honest majority at each stage.

> **Theorem 1** (Convergence Under Bounded Perturbations (informal)). *Consider a distributed training setup that utilizes PP to split the model layers across workers and uses momentum-based verification to verify each worker's contribution in the forward or backward pass.[a] Also, assume that less than half of workers at each stage are malicious (i.e., $\gamma_s < 1/2$) and we use a fixed threshold $\tau$ for worker verification using Eq. (2).[b] Under such relaxed conditions, training with non-convex loss functions optimized with momentum SGD converges to a neighborhood of a stationary point where the size of this neighborhood is directly proportional to $\tau$.*

Theorem 1 states malicious workers who evade detection by keeping perturbations below threshold $\tau$ can only cause the final solution to deviate from the optimal solution by an amount proportional to $\tau$. The formal theorem is given in App. E.1. Please also visit Q8 in the FAQ (App. A).

Recall that SENTINEL relies on the assumption that fewer than half of workers at any stage are malicious (i.e., $\gamma_s < 1/2$). Next, we quantify the conditions under which this assumption holds with high probability, given a total budget of malicious workers. The proof is given in App. E.2.

> **Lemma 1** (Honest Majority Guarantee). *Consider our distributed training system with $p$ pipeline stages, each replicated across $d$ worker nodes. Let $b$ be the total number of malicious workers, and $\epsilon \in (0, 1)$ be a small positive constant. If workers are assigned to each stage randomly and $b \leq dp/2 - p\sqrt{d/2 \ln (p/\epsilon)}$, then with probability at least $1 - \epsilon$ every pipeline stage has strictly fewer than $d/2$ malicious workers.*

## 4 RELATED WORK

Our momentum-based verification approach intersects three primary research directions that have largely evolved independently. For a more comprehensive review of related work, see App. C.

**Decentralized LLM Training** has emerged as a democratizing force in AI development. While frameworks like Tasklets (Yuan et al., 2022) or SWARM (Ryabinin et al., 2023) have made significant advances in communication efficiency and fault tolerance for non-malicious failures, they remain vulnerable to adversarial participants. Our work aims to address these vulnerabilities.

**Byzantine Robustness in Machine Learning** has traditionally focused on federated learning contexts where each worker computes complete model updates. Classic approaches like Krum (Blanchard et al., 2017), Bulyan (Mhamdi et al., 2018), and CENTEREDCLIP (Karimireddy et al., 2021) rely on comparing full gradients across workers, a fundamental mismatch with pipeline parallel architectures where each worker computes only a fraction of the model. SENTINEL is specifically designed for the unique constraints of pipeline parallelism.

---

[a]We do not consider dishonest activity during the "all-reduce" operation for syncing parameter gradients between DP replicas (which is the setting that all prior Byzantine-tolerant literature address). This is a complimentary axis and one can utilize any prior Byzantine-tolerant work.

[b]This is to relax our conditions. In practice we set this threshold automatically each iteration using the IQR.

Table 1: Attack detection performance for Llama-3-0.6B on C4 dataset.

| MODE | ATTACK | SENTINEL (OURS) | | | | | NO VERIF. |
|---|---|---|---|---|---|---|---|
| | | PR. (%) ↑ | RE. (%) ↑ | F1 (%) ↑ | DET. SPEED ↓ | VAL. LOSS ↓ | VAL. LOSS ↓ |
| - | None (Vanilla) | 100.0 | 100.0 | 100.0 | N/A | 3.819 | 3.821 |
| ACTIVATION | Scaling ($\alpha = -1$) | 100.0 | 100.0 | 100.0 | 6.38 | 3.824 | 4.109 |
| | Random Value | 100.0 | 100.0 | 100.0 | 6.48 | 3.827 | 7.778 |
| | Delay (100-steps) | 88.9 | 100.0 | 94.1 | 13.21 | 3.841 | 7.675 |
| | Bias Addition | 84.6 | 91.7 | 88.0 | 14.57 | 3.830 | 3.892 |
| | Invisible Noise (99%) | 100.0 | 100.0 | 100.0 | 6.48 | 3.826 | 7.682 |
| GRADIENT | Scaling ($\alpha = -1$) | 0.0 | 0.0 | 0.0 | N/A | 3.893 | 3.893 |
| | Random Value | 100.0 | 100.0 | 100.0 | 1.0 | 3.818 | 9.595 |
| | Delay (100-steps) | 100.0 | 100.0 | 100.0 | 7.33 | 3.826 | 10.157 |
| | Bias Addition | 100.0 | 100.0 | 100.0 | 1.0 | 3.828 | 10.813 |
| | Invisible Noise (99%) | 100.0 | 79.2 | 88.4 | 211.0 | 3.943 | 4.176 |

Table 2: Detection performance for training Llama-3-0.6B against mixed attacks.

| DATASET | SENTINEL (OURS) | | | | | VANILLA |
|---|---|---|---|---|---|---|
| | PR. (%) ↑ | RE. (%) ↑ | F1 (%) ↑ | DET. SPEED ↓ | VAL. LOSS ↓ | VAL. LOSS ↓ |
| COMMONCRAWL | 83.7 | 92.3 | 87.8 | 78.14 | 3.831 | 3.821 |
| FINEWEB | 81.8 | 92.3 | 86.7 | 66.00 | 3.827 | 3.840 |
| OPENWEBTEXT | 91.9 | 87.2 | 89.5 | 52.70 | 3.784 | 3.778 |

**Secure Distributed Systems** principles inform our verifier node architecture, which draws inspiration from trusted intermediaries in distributed computing. While preliminary work by Lu et al. (2024) identified potential vulnerabilities in pipeline parallel training, their redundancy-based solution would significantly reduce throughput, negating the primary benefit of distributed training. Our lightweight verification mechanism provides robust security guarantees with minimal computational overhead.

## 5 EXPERIMENTAL RESULTS

In this section, we present our experimental results. Unless stated otherwise, we use a 0.6B-parameter Llama-3 (Dubey et al., 2024; Liang et al., 2025) model (16 layers, 32 attention heads, 1024 hidden dimension and context length) distributed across 128 workers in a $8 \times 16$ data-pipeline parallel mesh. We use AdamW (Loshchilov & Hutter, 2019) with initial learning rate $6\mathrm{e}{-4}$ as our optimizer and train our models on FineWeb (FW) (Penedo et al., 2024), OpenWebText (OW) (Gokaslan et al., 2019), and Common Crawl (C4) (Raffel et al., 2020) datasets for 5k steps. We randomly designate 25% of workers at each pipeline stage as malicious (2:6 malicious vs. honest ratio), with only 25% of these activated simultaneously to soften our "no-collusion" assumption. Training begins with a 1k-step warm-up period before verification is activated. Based on validation runs on vanilla case, we set $\beta_h = 0.9$ and $\beta_g = 0.8$ for activation and gradient verification. Finally, for our adaptive threshold we use a window of past 100 steps. We relax these assumptions through various ablation studies to study their impact. Detailed experimental settings and extended results are provided in App. F.

**Metrics.** We evaluate our verification method using precision (Pr), recall (Re), and F1-score to measure effectiveness in detecting malicious workers while minimizing the false positives. To quantify detection efficiency, we report *detection speed* as the average number of iterations between the start of malicious behavior till the malicious worker gets banned. We also compare convergence rates across methods using the average loss at the last training step on a held-out validation set.

**Verification Performance.** We trained Llama-3-0.6B models on the C4 dataset with and without our verification mechanism and report the results in Tab. 1 (for more comprehensive results on C4 and other datasets, please see App. F.) Our experiments yield three key findings:

1. In pipeline parallelism, activation attacks are as threatening as gradient attacks, but their risk has been neglected in the Byzantine-tolerant literature.

2. Attack effectiveness varies significantly between activation and gradient domains. The same technique can severely disrupt training when targeting activations but have minimal impact when applied to gradients (e.g., invisible noise attacks).

3. Our EMA verification method achieves high F1-scores across attack types. When attacks produce negligible deviations and evade detection, their impact on convergence remains limited, confirming our theoretical analysis in Theorem 1 that undetected attacks can only shift parameters to a neighborhood of the optimum. Fig. 1 also confirms this relationship through a scatter plot of F1-scores against validation loss of more than 75 different attack setups from Tabs. 12 to 14.

Table 3: Activation attack detection performance for large-scale Llama-3 training on C4 dataset.

| SETUP | ATTACK | SENTINEL (OURS) | | | | | 
|---|---|---|---|---|---|---|
| | | PR. (%) ↑ | RE. (%) ↑ | F1 (%) ↑ | DET. SPEED ↓ | VAL. LOSS ↓ |
| 0.6B w/ 16 × 16 MESH | Random Value | 100.0 | 100.0 | 100.0 | 7.96 | 3.900 |
| | Delay (100-steps) | 85.7 | 100.0 | 92.3 | 14.39 | 3.945 |
| | Bias Addition | 100.0 | 25.6 | 40.8 | 65.15 | 3.981 |
| | Invisible Noise (99%) | 100.0 | 100.0 | 100.0 | 7.96 | 3.898 |
| 1.2B w/ 8 × 8 MESH | Random Value | 100.0 | 100.0 | 100.0 | 4.33 | 3.723 |
| | Delay (100-steps) | 37.5 | 100.0 | 54.5 | 67.0 | 3.774 |
| | Bias Addition | 0.0 | 0.0 | 0.0 | N/A | 3.738 |
| | Invisible Noise (99%) | 100.0 | 100.0 | 100.0 | 4.33 | 3.727 |

Table 4: Detection performance for training alternative models against mixed activation attacks.

| ARCHITECTURE | SENTINEL (OURS) | | | | VANILLA |
|---|---|---|---|---|---|
| | PR. (%) ↑ | RE. (%) ↑ | F1 (%) ↑ | VAL. LOSS ↓ | VAL. LOSS ↓ |
| LLAMA-4-0.4B | 73.5 | 92.3 | 81.8 | 3.617 | 3.628 |
| DEEPSEEK-V3-1B | 94.6 | 97.2 | 95.9 | 3.421 | 3.393 |
| LLAMA-3-4B | 94.9 | 68.6 | 79.6 | 3.714 | 3.668 |

Table 5: Detection performance for training Llama-3-0.6B against mixed attacks.

| METHOD | ATTACKED TRAINING | | | | VANILLA |
|---|---|---|---|---|---|
| | PR. (%) ↑ | RE. (%) ↑ | F1 (%) ↑ | VAL. LOSS ↓ | VAL. LOSS ↓ |
| SENTINEL + Krum | 93.6 | 80.6 | 86.6 | 3.873 | 3.855 |
| SENTINEL + Bulyan | 85.3 | 80.6 | 82.9 | 3.883 | 3.855 |

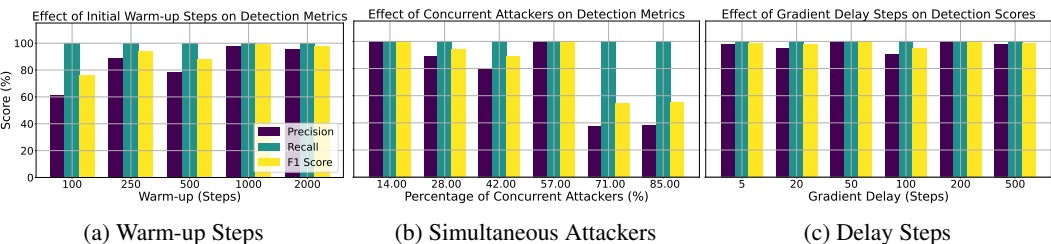

(a) Warm-up Steps  (b) Simultaneous Attackers  (c) Delay Steps

Figure 3: Ablation studies on the effect of various elements on the verification performance.

**Mixed Attacks.** While previous experiments examined individual attack types in isolation, real-world scenarios typically involve adversaries employing various attack strategies simultaneously across different nodes. To evaluate our approach under these more realistic conditions, we conduct experiments with the Llama-3-0.6B model in a distributed training environment where 37.5% of workers per stage are malicious (specifically, 3 malicious versus 5 honest workers per stage in an 8 × 16 mesh). All attackers begin their malicious activities simultaneously, with each attacker randomly assigned both an attack mode (activation or gradient manipulation) and a specific attack strategy from Sec. 2. Tab. 2 summarizes our experimental results across three datasets. The findings demonstrate that our verification method successfully identifies most detrimental attacks, resulting in validation loss metrics comparable to non-attacked baseline models. This robustness against heterogeneous attack vectors highlights the effectiveness of our approach in securing pipeline parallel-based LLM training against sophisticated adversarial scenarios.

**Large-scale Experiments.** To validate our approach at scale, we conduct experiments in two other settings: (1) a 16 × 16 mesh configuration with 256 total workers, and (2) a larger 1.2B parameter model trained on an 8 × 8 mesh with 64 workers. In both scenarios, we maintained 37.5% malicious workers per stage (6:10 malicious-to-honest ratio for the 16 × 16 mesh and 3:5 for the 8 × 8 mesh), with 25% of attackers active during each attack round. Results in Tab. 3 (see Tab. 15 for extended results) show that our verification mechanism effectively preserves training integrity across all attack scenarios, with validation losses comparable to non-attacked baselines. These results were achieved without extensive hyperparameter tuning compared to our 0.6B setting, though improvements are needed to reduce false positives in activation delay attacks on the 1.2B model.

**Ablation Studies.** Next, we study how key factors affect our verification method in detecting malicious workers. We evaluate using our strongest attacks, 99% invisible noise activation attacks (Studies 1 & 2), 100-step delayed gradient attacks (Study 3), and a mix of all activation attacks (Study 4 & 5), with a consistent ratio of 3 malicious to 5 honest workers per stage (37.5% malicious).

1. **Initial Warm-up Period**: Our method requires an initial warm-up phase to ensure that training has reached a stable point. Fig. 3a shows that while early detection achieves high recall, precision is initially low due to insufficiently robust thresholds because of training volatility. After roughly 1k steps, precision stabilizes as the verification method establishes reliable bounds.

2. **Attacker Collusion**: When malicious workers coordinate their attacks, detection becomes more challenging. As shown in Fig. 3b, our verification method maintains effectiveness with up to 60% collusion among malicious workers. Beyond this threshold, false positives increase significantly, which could adversely impact precision.

3. **Gradient Delay Impact**: Fig. 3c demonstrates our method's robustness to various delay lengths in gradient attacks. Even with minimal delays, where malicious gradients closely resemble current legitimate gradients, our verification method maintains high detection rates.

4. **Alternative Architectures**: To demonstrate the transferability of SENTINEL across transformer architectures, we train two Mixture-of-Expert (MoE) (Shazeer et al., 2017) architectures, namely Llama-4-0.4B (MetaAI, 2025) and DeepSeek-V3-1B (DeepSeek-AI, 2024), on the FineWeb-EDU (Penedo et al., 2024) dataset. We assume that 33% of the adversaries send their malicious vectors simultaneously as a form of collusion. We train these models as well as their vanilla baselines till 5000 steps. To demonstrate the transferability of our selected hyperparameters, we use the same hyperparameters used for the Llama-3-0.6B experiments for our verification. As shown in Tab. 4, SENTINEL can successfully be applied to these MoE-based models, showcasing its remarkable transferability across architectures. For more details, please see App. F.

5. **Large-scale Models**: Finally, we show that increasing model size from medium scale to large scale can be done without major performance degradation. To this end, we extend our Llama-3-0.6B experiment by increasing the hidden dimension and number of layers to bring the total parameter count to around 4B. We then train this model using a stringent total batch-size of 96 to fit within our available computational resources and train for 5k steps on the FineWeb dataset. As seen in Tab. 4, SENTINTEL protects the training from interruption attacks and delivers a model which has a validation loss close to the vanilla, non-attacked baseline (0.04 difference only) while keeping the detection accuracy around 80%. This highlights that our findings are scalable to larger models with wider hidden dimension and deeper layers.

For more results on the impact of EMA, longer runs, datasets, etc. please see App. F.2.

**Adaptive Attacks.** In this section, we investigate an adaptive attack that knows how SENTINEL is verifying its signals. In particular, let us assume that the malicious workers maintain an EMA of signals sent to subsequent layers, using the

Table 6: Verification performance against adaptive attacks.

| MODE | SENTINEL (OURS) | | | | NO VERIF. |
|------|-----------|-----------|-----------|------------|-----------|
| | PR. (%) ↑ | RE. (%) ↑ | F1 (%) ↑ | VAL. LOSS ↓ | VAL. LOSS ↓ |
| ACTIVATION | 100.0 | 100.0 | 100.0 | 3.835 | 7.776 |
| GRADIENT | 100.0 | 100.0 | 100.0 | 3.818 | 7.777 |

same $\beta$ as the verifier node. After collecting sufficient EMA samples, the attack sends drifted activations/gradients biased toward a predetermined target direction, using stale momentum estimates to create consistent bias while appearing statistically legitimate. The attacker constructs its attacks according to $\hat{x} = m_{t-\delta} + \alpha \cdot (t_{\text{drift}} - m_{t-\delta}) / \|m_{t-\delta}\| + \epsilon$ where $m_{t-\delta}$ is the stale EMA momentum, $\alpha$ is the drift rate, $t_{\text{drift}}$ is the predetermined drift target, $\epsilon \sim \mathcal{N}(0, \sigma^2)$ is Gaussian noise for stealth, and $\delta = \left\lceil \frac{\log(0.1)}{\log(\beta)} \right\rceil$ is the delay factor. We run this attack for activation and activation gradients for training a Llama-3-0.6B on C4 dataset. For activation attack, the verifier/attacker uses $\beta = 0.9$, while for gradient attack, they use $\beta = 0.8$.

Running this attack using our settings from Tab. 1 (25% malicious workers at each stage), we get the results in Tab. 6. As seen, the adaptive EMA attack can be destructive without verification, SENTINEL detects and mitigates it perfectly, validating our resilience against adaptive attacks that have a knowledge of our defense. This is because the attacker's EMA would only comprise part of the total true EMA, and assuming an honest majority, this would not be sufficient to interrupt training.

**Integration with DP Defenses.** As elaborated throughout the paper, SENTINEL targets verifying the signals sent between stages in PP. Unlike the PP axis, we pointed out that existing Byzantine-tolerant literature exclusively target attacks that happen during gradient averaging (a.k.a. all-reduce). As such, these two are complementary axes that operate on different dimensions (please see Q0 in App. A). To demonstrate the complementary nature of these two axes, we use two commonly used robust aggregators, particularly Krum (Blanchard et al., 2017) and Bulyan (Mhamdi et al., 2018), to replace the vanilla all-reduce between workers. We then train our Llama-3-0.6B against mixed attacks with

3:5 malicious to honest ratio for 5k steps on FineWeb. As seen in Tab. 5, addition of these methods on the orthogonal DP axis has no significant impact on the convergence of the model under mixed attacks, supporting our claim that the PP and DP solutions are complementary.

## 5.1 Extending Sentinel to SWARM Parallelism

Finally, we adapt SENTINEL to verify worker node signals in a realistic SWARM (Ryabinin et al., 2023) experiment. SWARM parallelism provides a fault-tolerant distributed training ecosystem powered by the Hivemind framework (Ryabinin & Gusev, 2020). It comprises of worker nodes distributed across both DP and PP coordinates. At each stage, a pool of workers process batches of data, with the coordination managed through trainer nodes that are responsible for stochastically routing activations in the forward pass and activation gradients in the backward pass. From this standpoint, SWARM parallelism is akin to a stochastic DP/PP mesh in comparison to the fixed setting that we have considered so far. As discussed in Sec. 2.1, trainer nodes are in a natural position to be extended as verifier nodes when augmented with SENTINEL.

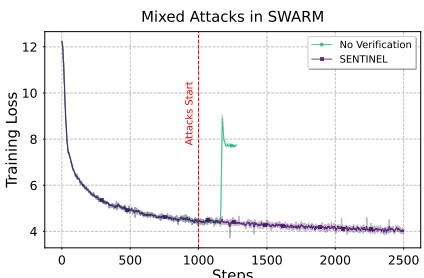

For this experiment, we train our Llama-3-0.6B model across a distributed SWARM configuration with 128 workers (8 × 16 mesh). We employ 32 trainer nodes with verification capability to train our model on FineWeb-EDU, a curated subset of FineWeb. Since trainers do not have P2P communication, each maintain independent EMAs with a single synchronization point at the end of the warm-up.

We evaluate robustness against a mixture of random attacks by designating 37.5% of workers at each stage (except the first and last two) as malicious, maintaining a 3:5 malicious-to-honest ratio with 15% collusion where attackers activate simultaneously. As shown in Fig. 4, the presence of malicious workers significantly disrupts training convergence in the absence of verification. However, SENTINEL successfully maintains the integrity of the training, enabling

Figure 4: Loss when training Llama-3-0.6B models using SWARM (Ryabinin et al., 2023) with 128 distributed workers on preemptible AWS instances. Workers employ various activation/gradient manipulation attacks to disrupt training. While in the absence of verification training gets disrupted, SENTINEL can successfully protect training from divergence.

the training to continue without interruption. This result in a production-grade environment demonstrates the practical applicability of our approach for securing real-world decentralized training ecosystems. For implementation details and full results of SENTINEL in SWARM, see App. G.

## 6 Conclusion

In this paper, we investigated security vulnerabilities in decentralized, pipeline parallel networks, showing how malicious workers can corrupt activations and activation gradients exchanged between pipeline stages. To guide future research, we introduced a suite of training-interruption attacks as benchmarks for evaluating decentralized training security. Our key contribution is SENTINEL, a lightweight momentum-based verification mechanism that utilizes trusted verifier nodes to maintain EMAs of transmitted signals (activations and activation gradients) as statistical reference points. We further developed an IQR-based adaptive thresholding strategy to automatically calibrate detection sensitivity. We complement our approach by theoretical analysis and real-world integration for decentralized training using SWARM. Through extensive experiments with models up to 1.2B parameters distributed across hundreds of workers, we demonstrated its effectiveness in maintaining training integrity with consistently high F1 scores ($> 85\%$) across various attack scenarios.

**Limitations**: While our verification method effectively detects the attack types presented, it may not generalize to all possible adversarial strategies. Future work should explore better adaptive detection mechanisms that require less manual tuning, possibly neural networks for anomaly detection (Pang et al., 2021). Additionally, our approach addresses inter-stage attacks specific to pipeline parallelism, but decentralized training remains vulnerable to other threats including backdoor attacks (Li et al., 2024), membership inference (Shokri et al., 2017), and all-reduce gradient poisoning attacks (Gorbunov et al., 2022) each presenting distinct challenges for truly open collaborative training.

REPRODUCIBILITY STATEMENT

We provide a detailed, step-by-step pseudo-code of our methodology in Algs. 1 to 5 and 7. Additionally, we give a detailed overview of the hyper-parameters used in our empirical evaluations in App. F.1. We are planning to release the code upon acceptance of the paper.

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

# APPENDIX

The appendix is organized as follows:

- In App. A, we present answers to several common questions, hoping to clarify misconceptions that might arise when interpreting our work and its limitations.
- App. B provides a holistic view of vulnerabilities related to decentralized training that are beyond the scope of the current work.
- We provide an extended version of the related work to SENTINEL in App. C that was omitted from the main paper due to space limitations.
- We present our methodology details in App. D.
- App. E contains our theoretical analysis of SENTINEL, including convergence analysis under relaxed assumptions and conditions under which random worker assignment could result in an honest majority at each stage.
- In App. F.2, we provide detailed experimental results and additional ablations.
- Finally, in App. G, we present a step-by-step integration of SENTINEL with SWARM parallelism (Ryabinin et al., 2023). We provide extensive investigation into seamless integration with trainer nodes that coordinate training in SWARM, followed by real-world experiments training LLMs over 128 untrusted worker nodes employing malicious attacks.

## A  FREQUENTLY ASKED QUESTIONS (FAQS)

**Q0: What are the fundamental differences between data parallel (DP) and pipeline parallel (PP) settings that make existing Byzantine tolerant literature inapplicable to this work?**  We provide a clear comparison between DP and PP settings, highlighting why existing Byzantine tolerant literature does not apply to our work:

Table 7: Data Parallel vs. Pipeline Parallel Comparison

| Aspect | Data Parallel (Prior Work) | Pipeline Parallel (Our Work) |
|---|---|---|
| Model Distribution | Full model replica per worker | Model split across workers (layers/stages) |
| Data Distribution | Different batches per worker | Same batch flows through pipeline |
| Communication Pattern | Parameter gradients aggregated | Activations/gradients passed sequentially |
| Byzantine Threat | Corrupted parameter gradients | Corrupted inter-stage activations/gradients |
| Detection Target | Malicious gradient contributions | Malicious activation/gradient transmissions |
| Aggregation Method | Robust gradient aggregation | Sequential verification at each stage |
| Literature Focus | Robust aggregators | No non-trivial prior verification exists |

Thus, the key distinctions are:

- **Prior Byzantine tolerant literature**: Secures the DP axis by developing robust aggregation methods for parameter gradients from multiple model replicas
- **Our work**: Secures the orthogonal PP axis by verifying activations and activation gradients transmitted between sequential pipeline stages using SENTINEL.

Hence, existing Byzantine tolerant works do NOT apply to the PP axis because:

1. **No aggregation possible**: In PP, activations from different workers cannot be aggregated (they represent different layers processing different data batch).
2. **Sequential dependency**: Each stage depends on the previous stage's output, making robust aggregation impossible.
3. **Different threat model**: Malicious workers corrupt intermediate representations rather than final parameter updates.
4. **Verification vs. Aggregation**: Our verifiers monitor communication channels rather than aggregate multiple contributions.

This fundamental difference explains why our threat model and verification approach are necessarily different from "typical Byzantine tolerant" literature.

**Q1: Why would a model owner deliver part of their model to an untrusted entity to train?**
The computational resources required for training LLMs are becoming increasingly unsustainable. As reported by Brown et al. (2020), training a single GPT-3 175B model requires up to 3.6K Petaflop-days, incurring a total cost of $4M in AWS pricing. In the absence of big corporations delivering open source models, decentralized training provides an alternative solution for training such models in a democratized environment where models can be trained openly and participants are reimbursed based on their contributions. As discussed in (Yuan et al., 2022), consumer device GPUs are becoming increasingly available worldwide, many of which are underutilized. "If we could make use of these devices in a decentralized open-volunteering paradigm for foundation model training, this would be a **revolutionary alternative** to the expensive solutions offered by data centers."

**Q2: Is assuming 25-37.5% malicious workers realistic? Why would anyone trust such a system?**
We understand the intuitive concern about our 25-37.5% malicious worker percentages. However, note that these assumptions are standard practice in the Byzantine fault tolerance literature, as evidenced by recent work in this area outlined in Tab. 8. Importantly, our work serves as a preventative security mechanism. The goal is to deter malicious behavior by demonstrating robust detection capabilities, not to operate under the assumption that such high percentages will necessarily occur in practice. As Fig. 3b demonstrates, when malicious nodes are at lower levels, our performance approaches nearly 100% detection accuracy. The worst-case analysis ensures system reliability even under extreme adversarial conditions.

Table 8: Malicious Worker Percentages in Byzantine Fault Tolerance Literature

| Reference | Venue | Malicious Nodes (%) |
|---|---|---|
| Mhamdi et al. (2018) | ICML 2018 | 47.37 |
| Gorbunov et al. (2022) | ICML 2022 | 43.75 |
| Blanchard et al. (2017) | NeurIPS 2017 | 33.00 |
| Karimireddy et al. (2021) | ICML 2021 | 30.55 |
| Malinovsky et al. (2024) | NeurIPS 2024 | 25.00 |
| Gorbunov et al. (2023) | ICLR 2023 | 20.00 |
| Karimireddy et al. (2022) | ICLR 2022 | 20.00 |
| Rammal et al. (2024) | AISTATS 2024 | 18.75 |

**Q3: How practical is the integration of verifier nodes in distributed frameworks?** In real-world systems deploying our algorithm (such as SWARM), trainer nodes are responsible for transmitting activations/activation gradients between layers. We propose modifying these existing trainer nodes to perform verification, essentially obtaining this security functionality "for free" since they already have access to all signals passing between layers. Crucially, trainer nodes represent a centrally controlled role in distributed systems as they are managed by the network coordinator rather than volunteers, making it economically practical to maintain control over them. Since their primary responsibility is coordinating signal transmission, these are lightweight CPU-based nodes with minimal cost. For example, consider training a 4B parameter LLM where each layer/stage requires roughly 18GB of GPU VRAM. On AWS, each worker would require a `g5.2xlarge` EC2 instance with 24GB VRAM at approximately $1.212 per hour. In contrast, trainer nodes can be bundled with 8 nodes per `c5a.8xlarge` instance (32vCPUs, 4vCPU per trainer) at $1.232 per hour (i.e., $0.154 per trainer node per hour). This represents a significant cost reduction compared to worker instances, and since trainers are centrally controlled, it is both economically and operationally feasible for network coordinators to bear these costs while maintaining security and reliability. If trainer roles must be delegated to volunteers, proper authentication mechanisms would be required to prevent malicious behavior, which we leave to future work. For a more in-depth discussion on implementing SENTINEL in SWARM, please see App. G.

**Q4: Why do you assume a "warm-up" period with only honest workers? Does this contradict the spirit of distributed training?** In distributed environments, it is common practice to start training in a controlled environment until training reaches a stabilized point before allowing public workers to join. This approach is important not only for our verification method, but also ensures that training is stable before public participation begins. This is NOT against the spirit of distributed training: consider a 40-layer LLM intended for decentralized training. Initially, we deploy a single replica across 40 nodes under our control, but training throughput is limited. After the initial warm-up phase,

we replicate the pipeline across 7 additional replicas, bringing our total DP workers to 320, where only 12.5% are controlled by the model owner and 87.5% are public workers. Note that maintaining at least one trusted worker at each stage is crucial not only for initial warm-up but also due to system reliability as the volunteer nodes (280 of them in our example) might drop mid-run. We need at least one reliable pipeline to ensure training continuity when volunteers drop.

**Q5: Does the $N$-to-$1$ communication to verifiers create a bandwidth bottleneck?**  There is no bandwidth bottleneck. In model parallel implementations such as SWARM, coordination between stages occurs through trainer nodes that send and receive activations/gradients. Since each trainer node already observes all signals passing from one stage to another, we leverage it as our trusted verifier node. As we discuss in detail in App. G, SWARM utilizes multiple trainer nodes to streamline the data through the stages one-by-one. Thus, each trainer can run their own verification using SENTINEL simultaneously to other trainers sending signals to the workers. For technical details on trainer node operations, see the Hivemind (Ryabinin & Gusev, 2020) library.

**Q6: How does the verification mechanism handle false positives?**  The violation counter with a forgiveness mechanism is a clever way to handle transient anomalies and avoid unfairly banning honest workers. The training curves are given in Figs. 8 and 16 in App. F. As seen, the training/validation curves show no visible impact on training dynamics from transiently replacing submitted signals with EMA values. This makes intuitive sense: when only a few nodes undergo the EMA replacement phase, the remaining workers in the DP setup continue to submit useful signals that guide training effectively.

**Q7: Why does the effectiveness of training-interruption attacks vary across different attack versions and targets?**  When comparing activation-based attacks against gradient-based attacks, two key factors explain the effectiveness differences:

- **Magnitude differences:** Activation values are orders of magnitude larger than activation gradients. Therefore, methods that modify activations can achieve larger perturbation magnitudes during attacks, resulting in more successful disruption.
- **Propagation scope:** Activation manipulation at layer $1 \le \ell \le L$ affects the forward pass for all subsequent layers ($[\ell + 1, \ell + 2, \ldots, L]$) and the backward pass of all layers. In contrast, gradient manipulation at layer $\ell$ only affects the gradients of preceding layers $[1, 2, \ldots, \ell - 1]$. Consequently, if an attacker is positioned in the middle of the network, manipulating activations has broader impact on the entire training process.

**Q8: How does your convergence rate compare to well-known lower bounds from Byzantine-tolerant literature?**  Our convergence guarantee provides an accurate bound given our assumptions. We note a key distinction in our setting: prior Byzantine-tolerant literature considers data parallel training where malicious actors modify "parameter gradients". Our work addresses the *orthogonal* pipeline parallel axis where activations and gradients "between layers" are shared and require verification. These two axes are complementary: securing both pipeline parallel and data parallel axes is important in decentralized settings, and this work focuses on the former. Therefore, prior guarantees from Byzantine-tolerant literature are not directly comparable.

**Q9: How does your method compare to the prior work by Lu et al. (2024)?**  Lu et al. (2024) used a naïve approach of assigning one duplicate replica per worker for verification. For instance, with 320 worker nodes this requires splitting them into two groups: 160 workers performing computation and 160 workers replicating their work for verification. While this approach achieves 100% F1-score on all attacks, it operates at HALF the true distributed network throughput. Our solution does NOT replicate volunteer node work, achieving twice the training speed. Conceptually, on OpenWebText against mixed attacks we will have:

**Q10: Does removing nodes from the training runs degrade the throughput?**  Note that in real-world eco-systems such as SWARM (Ryabinin et al., 2023), we are not using a fixed mesh anymore. Workers join and leave the SWARM as they wish, and within each pipeline stage, there are

Table 9: Comparison with prior work on OpenWebText mixed attacks

| METHOD | METRICS | | | |
|---|---|---|---|---|
| | PR. (%) ↑ | RE. (%) ↑ | F1 (%) ↑ | TPS ↑ |
| DUPLICATE WORK (Lu et al., 2024) | 100.0 | 100.0 | 100.0 | 6483 |
| SENTINEL (OURS) | 91.9 | 87.2 | 89.5 | 12966 |

numerous workers that are serving that stage. This combined with stochastic routing in SWARM (please see the details in App. G.3) ensures that workers would reach their maximum utilization. Thus, in real-world systems where we have an abundance of workers serving each stage where kicking workers out would not necessarily degrade throughput.

## B   VULNERABILITIES OF DECENTRALIZED TRAINING USING DATA AND PIPELINE PARALLELISM

In decentralized settings used for collaborative training, the verification of participant workers is essential to maintain the integrity, security, and overall effectiveness of the training process (Lu et al., 2024). Verification acts as a critical quality-control measure, ensuring that each participant meaningfully contributes to the collective training effort. Without adequate verification mechanisms, malicious actors can infiltrate the SWARM, potentially compromising its integrity. Such attackers might disrupt the collaborative training, degrade its efficiency, or illegitimately benefit by accessing the trained model without genuinely contributing.

Below, we categorize common malicious behaviors that could arise in decentralized collaborative training scenarios. These categories are not mutually exclusive, as attackers may employ several tactics simultaneously. Nevertheless, this classification provides a structured overview of the key threats:

- **Training Disruption (Denial-of-Service or DoS)**: Attackers intentionally impede or halt the training process. This can occur through dropping essential updates, introducing malicious data designed to break communication protocols, or overwhelming the system with excessive or irrelevant submissions.

- **Free-Riding or Minimal Effort Contributions**: Participants contribute minimal computational effort or data yet aim to reap the benefits of the collective process, such as accessing the final model, receiving rewards, or boosting their reputation (Zhu et al., 2025). Common tactics include submitting trivial updates or strategically remaining inactive until training nears completion.

- **Model Poisoning and Backdoor Attacks**: Malicious actors provide adversarial updates designed to introduce subtle vulnerabilities or targeted misbehaviors in the resulting model (Li et al., 2024). Typically concealed under normal operational conditions, these backdoors or compromised models trigger malicious outcomes only under specific, pre-defined scenarios.

- **Privacy Violations (Data Extraction or Inference Attacks)**: Attackers exploit gradients, activations, or other shared information during training to infer sensitive or private information from other participants' datasets, thereby breaching confidentiality and compromising user privacy (Shokri et al., 2017).

- **Reputation or Credit Manipulation**: Participants deliberately falsify or exaggerate their contributions (for instance, by generating seemingly high-quality updates) to unjustly obtain greater rewards, enhanced reputation, or tokens. This form of manipulation undermines the fairness of the system and distorts trust among honest contributors.

In this paper, our primary focus is mitigating threats associated with training disruption. Ensuring the identification and exclusion of malicious participants who submit harmful or disruptive updates is critical. Failure to address these threats effectively would prevent the swarm from achieving model convergence and producing a reliable, functional final model.

## C    EXTENDED RELATED WORK

Secure distributed training has gained significant attention with the proliferation of decentralized machine learning systems. Our work builds upon several research threads while addressing unique challenges posed by pipeline parallelism in LLM training.

**Decentralized Training Frameworks.**    Decentralized training has emerged as a promising approach for democratizing AI capabilities. Ryabinin and Gusev (Ryabinin & Gusev, 2020) introduced the Hivemind framework, enabling mixture-of-experts models to be trained in a decentralized fashion. Building on this foundation, (Ryabinin et al., 2021) proposed Moshpit SGD, a communication-efficient algorithm for training on heterogeneous and unreliable devices. In parallel, (Yuan et al., 2022) introduced Tasklets, a system for decentralized training in heterogeneous environments that adapts to varying network conditions and compute capabilities. SWARM parallelism (Ryabinin et al., 2023) further enhanced this approach by combining pipeline and data parallelism to enable training of models significantly larger than those possible with previous decentralized methods. While these frameworks prioritize fault tolerance against non-malicious failures, they generally lack protection against adversarial participants which is a critical vulnerability in open decentralized training environments.

**Byzantine-Resilient Distributed Training.**    Byzantine fault tolerance in distributed learning has been extensively studied in the context of federated learning (McMahan et al., 2017) and data-parallel training (Li et al., 2020). (Blanchard et al., 2017) introduced Krum, the first Byzantine-tolerant aggregation rule for distributed SGD that could withstand arbitrary gradient manipulations from compromised workers. This was followed by more sophisticated approaches including Bulyan (Mhamdi et al., 2018), median-based aggregation (Baruch et al., 2019), and clipping-based methods (He et al., 2020; Malinovsky et al., 2024).

A fundamentally different approach called CENTEREDCLIP was proposed by (Karimireddy et al., 2021), who leveraged historical gradient information to detect anomalous updates – conceptually similar to our momentum-based verification but applied specifically to gradient aggregation. Recent work by Rammal et al. (Rammal et al., 2024) demonstrated that communication compression could be effectively combined with Byzantine-robust learning, achieving improved convergence rates while maintaining security guarantees.

While these methods provide strong theoretical guarantees, they primarily target scenarios where workers compute complete gradients independently, making them ill-suited for pipeline parallel configurations like SWARM where intermediate activations are communicated between stages. Furthermore, these approaches often involve comparing gradients across workers, which would necessitate parameter replication across stages, contradicting pipeline parallel's objective of enabling training of models too large to fit on a single device.

**Security in Pipeline Parallel Architectures.**    Security considerations specific to pipeline parallel training have received limited attention compared to other distributed paradigms. (Lu et al., 2024) recently presented a position paper exploring robustness challenges in pipeline parallelism-based decentralized training, highlighting activation-based attacks as a critical concern. Their work, however, focused primarily on identifying vulnerabilities rather than proposing comprehensive verification solutions. The redundancy-based approach proposed by (Lu et al., 2024) and (Rajput et al., 2019) could, in principle, be adapted to decentralized pipeline parallel settings. However, these approaches would introduce significant computational overhead (due to duplicating computations across workers) which would greatly diminish the scalability benefits of decentralized training.

## D    DETAILS OF MOMENTUM-BASED VERIFICATION

### D.1    MOMENTUM-BASED VERIFICATION ALGORITHMS

In this section, we present our detailed algorithm for worker verification in decentralized training. Alg. 1 outlines our end-to-end verification mechanism for this setting. We present the algorithm chronologically as training progresses. The verifier nodes perform all verification operations, while worker nodes are solely responsible for computing activations during the forward pass and their

respective gradients during the backward pass. Algorithms 2 and 3 detail the verification procedures for both forward and backward passes, respectively. Alg. 4 presents our approach for mitigating cascading effects as described in Section 3.1. Finally, Alg. 5 specifies our adaptive IQR threshold setting methodology for each metric in our approach.

---

**Algorithm 1** Momentum-based Verification for SWARM Parallelism

---

**Require:** Parameters $\beta_h, \beta_g \in (0, 1)$, violation threshold $c$, forgiveness period $T_{\text{forgiveness}}$, set of metrics $\mathcal{M}$

1: Initialize $\boldsymbol{m}_0^{(s)}(\boldsymbol{h}) = \boldsymbol{0}, \boldsymbol{m}_0^{(s)}(\boldsymbol{g}) = \boldsymbol{0}, v_r^{(s)} = 0, B_s = \emptyset, \mathcal{H}_l^{(s)} = \emptyset, \mathcal{G}_l^{(s)} = \emptyset$ for all $s, r$

2: Initialize $\mathcal{T} = \emptyset$ // *Initialize global tainted set*

3: // *Warm-up phase to establish baseline statistics*

4: **for** $t = 1$ to $T_{\text{warmup}}$ **do**

5:     **for** $s \in \{1, 2, \ldots, p\}$ **do**

6:         Collect $\boldsymbol{h}_t^{(s,r)}$ and $\boldsymbol{g}_t^{(s,r)}$ from all workers $r \in \{1, \ldots, d\}$

7:         Compute $\Gamma_t^{(s,r)} = \Omega(\boldsymbol{h}_t^{(s,r)}, \boldsymbol{m}_{t-1}^{(s)}(\boldsymbol{h})) \; \forall r$, add to $\mathcal{H}_l^{(s)}$

8:         Compute $\Gamma_t^{(s,r)} = \Omega(\boldsymbol{g}_t^{(s,r)}, \boldsymbol{m}_{t-1}^{(s)}(\boldsymbol{g})) \; \forall r$, add to $\mathcal{G}_l^{(s)}$

9:         Update momentum $\boldsymbol{m}_t^{(s)}(\boldsymbol{h})$ and $\boldsymbol{m}_t^{(s)}(\boldsymbol{g})$ using Eq. (1)

10:     **end for**

11: **end for**

12: // *Main training phase with verification*

13: **for** $t = T_{\text{warmup}} + 1$ to $T_{\text{total}}$ **do**

14:     $\mathcal{T}_t = \emptyset$ // *Initialize tainted set for current iteration*

15:     // *Step 1: Forward Pass and Activation Verification*

16:     **for** $s \in \{1, \ldots, p\}$ **do**

17:         $\mathcal{T}_t^{(s)} \leftarrow \text{ACTIVATIONVERIFICATION}(s, t, \boldsymbol{m}_{t-1}^{(s)}(\boldsymbol{h}), \mathcal{H}_l^{(s)}, B_s, v_r^{(s)}, \mathcal{M}, c, T_{\text{forgiveness}}, \mathcal{T}_t)$

18:         $\mathcal{T}_t \leftarrow \mathcal{T}_t \cup \mathcal{T}_t^{(s)}$ // *Accumulate tainted workers*

19:         $R_{\text{clean}} = \{r : (t, s, r) \notin \mathcal{T}_t\}$

20:         $\boldsymbol{m}_t^{(s)}(\boldsymbol{h}) = \beta_h \boldsymbol{m}_{t-1}^{(s)}(\boldsymbol{h}) + (1 - \beta_h) \frac{1}{|R_{\text{clean}}|} \sum_{r \in R_{\text{clean}}} \boldsymbol{h}_t^{(s,r)}$

21:     **end for**

22:     **for** $s \in \{p, p-1, \ldots, 1\}$ **do**

23:         // *Step 2: Backward Pass and Gradient Verification*

24:         $\mathcal{T}_t^{(s)} \leftarrow \text{GRADIENTVERIFICATION}(s, t, \boldsymbol{m}_{t-1}^{(s)}(\boldsymbol{g}), \mathcal{G}_l^{(s)}, B_s, v_r^{(s)}, \mathcal{M}, c, T_{\text{forgiveness}}, \mathcal{T}_t)$

25:         $\mathcal{T}_t \leftarrow \mathcal{T}_t \cup \mathcal{T}_t^{(s)}$ // *Accumulate tainted workers*

26:         // *Step 3: Gradient Replacement for Tainted Workers*

27:         $\boldsymbol{g}_t^{(s)} \leftarrow \text{GRADIENTREPLACEMENT}(s, t, \boldsymbol{m}_{t-1}^{(s)}(\boldsymbol{g}), \boldsymbol{g}_t^{(s)}, \mathcal{T}_t)$

28:         $R_{\text{clean}} = \{r : (t, s, r) \notin \mathcal{T}_t\}$

29:         $\boldsymbol{m}_t^{(s)}(\boldsymbol{g}) = \beta_g \boldsymbol{m}_{t-1}^{(s)}(\boldsymbol{g}) + (1 - \beta_g) \frac{1}{|R_{\text{clean}}|} \sum_{r \in R_{\text{clean}}} \boldsymbol{g}_t^{(s,r)}$

30:     **end for**

31:     $\mathcal{T} \leftarrow \mathcal{T} \cup \mathcal{T}_t$ // *Accumulate tainted entries across iterations*

32: **end for**

33: **return** $\mathcal{T}$ // *Return the complete set of tainted worker-stage-iteration tuples*

---

**Algorithm 2** ACTIVATIONVERIFICATION

**Require:** $s, t, \boldsymbol{m}_{t-1}^{(s)}(\boldsymbol{h}), \mathcal{H}_l^{(s)}, B_s, v_r^{(s)}, \mathcal{M}, c, T_{\text{forgiveness}}, \mathcal{T}_t$

1: $\mathcal{T}_t^{(s)} = \emptyset$ *// Initialize tainted set for current stage*

2: Truncate $\mathcal{H}_l^{(s)} \leftarrow \mathcal{H}_l^{(s)}[-l : \text{end}]$

3: Collect $\boldsymbol{h}_t^{(s,r)}$ from all $r \in \{1, \ldots, d\} \setminus B_s$

4: **for** $r \in \{1, \ldots, d\} \setminus B_s$ not in $\mathcal{T}_t$ **do**

5:     Compute metrics: $\Gamma_t^{(s,r,i)} = \Omega_i(\boldsymbol{h}_t^{(s,r)}, \boldsymbol{m}_{t-1}^{(s)}(\boldsymbol{h}))$ for $i \in \mathcal{M}$

6:     **if** $\exists i \in \mathcal{M} : |\Gamma_t^{(s,r,i)} - q_2^{(s,i)}| \geq k_{\text{tukey}}^{(s,i)}(q_3^{(s,i)} - q_1^{(s,i)})$ **then**

7:         $v_r^{(s)} \leftarrow v_r^{(s)} + 1$ *// Increment violation counter*

8:         $\mathcal{T}_t^{(s)} \leftarrow \mathcal{T}_t^{(s)} \cup \{(t, s, r)\}$ *// Mark as tainted in current stage*

9:         **if** $v_r^{(s)} \geq c$ or $\Gamma_t^{(s,r,i)} \gg k_{\text{tukey}}^{(s,i)}(q_3^{(s,i)} - q_1^{(s,i)})$ **then**

10:            $B_s \leftarrow B_s \cup \{r\}$ *// Ban worker*

11:            Notify stages $s' > s$ to flag affected mini-batches

12:         **end if**

13:     **else**

14:         Add $\Gamma_t^{(s,r,i)}$ to $\mathcal{H}_l^{(s,i)} \forall i \in \mathcal{M}$

15:         $v_r^{(s)} \leftarrow \max(0, v_r^{(s)} - 1)$ if $T_{\text{forgiveness}}$ consecutive clean steps

16:     **end if**

17: **end for**

18: Update IQR statistics and adjust $k_{\text{tukey}}^{(s,i)} \forall i \in \mathcal{M}$ using Alg. 5

19: **if** $|\mathcal{T}_t^{(s)}| > 0.5 \cdot (d - |B_s|)$ **then**

20:     $\mathcal{T}_t^{(s)} \leftarrow \emptyset$ *// Clear if more than 50% flagged (natural shift, see App. D.3)*

21: **end if**

22: **return** $\mathcal{T}_t^{(s)}$ *// Return tainted workers for this stage*

**Algorithm 3** GRADIENTVERIFICATION

**Require:** $s, t, \boldsymbol{m}_{t-1}^{(s)}(\boldsymbol{g}), \mathcal{G}_l^{(s)}, B_s, v_r^{(s)}, \mathcal{M}, c, T_{\text{forgiveness}}, \mathcal{T}_t$

1: $\mathcal{T}_t^{(s)} = \emptyset$ // Initialize tainted set for current stage

2: Truncate $\mathcal{G}_l^{(s)} \leftarrow \mathcal{G}_l^{(s)}[-l : \text{end}]$

3: $\text{Tainted}_{\text{downstream}} = \{r : (t, s', r) \in \mathcal{T}_t \text{ for some } s' > s\}$ // Workers tainted in downstream stages

4: **for** $r \in \{1, \dots, d\} \setminus B_s$ **do**

5:     **if** $r \in \text{Tainted}_{\text{downstream}}$ **then**

6:         $\mathcal{T}_t^{(s)} \leftarrow \mathcal{T}_t^{(s)} \cup \{(t, s, r)\}$ // Mark as tainted by downstream

7:     **else**

8:         Collect $\boldsymbol{g}_t^{(s,r)}$ from worker $r$

9:         Compute metrics: $\Gamma_t^{(s,r,i)} = \Omega_i(\boldsymbol{g}_t^{(s,r)}, \boldsymbol{m}_{t-1}^{(s)}(\boldsymbol{g}))$ for $i \in \mathcal{M}$

10:         **if** $\exists i \in \mathcal{M} : |\Gamma_t^{(s,r,i)} - q_2^{(s,i)}| \geq k_{\text{tukey}}^{(s,i)}(q_3^{(s,i)} - q_1^{(s,i)})$ **then**

11:             $v_r^{(s)} \leftarrow v_r^{(s)} + 1$ // Increment violation counter

12:             $\mathcal{T}_t^{(s)} \leftarrow \mathcal{T}_t^{(s)} \cup \{(t, s, r)\}$ // Mark as tainted

13:             **if** $v_r^{(s)} \geq c$ or $\Gamma_t^{(s,r,i)} \gg k_{\text{tukey}}^{(s,i)}(q_3^{(s,i)} - q_1^{(s,i)})$ **then**

14:                 $B_s \leftarrow B_s \cup \{r\}$ // Ban worker

15:                 Notify stages $s' < s$ to flag affected mini-batches

16:             **end if**

17:         **else**

18:             Add $\Gamma_t^{(s,r,i)}$ to $\mathcal{G}_l^{(s,i)} \forall i \in \mathcal{M}$

19:             $v_r^{(s)} \leftarrow \max(0, v_r^{(s)} - 1)$ if $T_{\text{forgiveness}}$ consecutive clean steps

20:         **end if**

21:     **end if**

22: **end for**

23: Update IQR statistics and adjust $k_{\text{tukey}}^{(s,i)} \forall i \in \mathcal{M}$ using Alg. 5

24: **if** $|\mathcal{T}_t^{(s)}| > 0.5 \cdot (d - |B_s|)$ **then**

25:     $\mathcal{T}_t^{(s)} \leftarrow \emptyset$ // Clear if more than 50% flagged (natural shift, see App. D.3)

26: **end if**

27: **return** $\mathcal{T}_t^{(s)}$ // Return tainted workers for this stage

---

**Algorithm 4** GRADIENTREPLACEMENT

**Require:** $s, t, \boldsymbol{m}_{t-1}^{(s)}(\boldsymbol{g}), \boldsymbol{g}_t^{(s)}, \mathcal{T}_t$

1: **for** $r \in \{1, \dots, d\}$ **do**

2:     **if** $(t, s, r) \in \mathcal{T}_t$ **then**

3:         $\boldsymbol{g}_t^{(s,r)} \leftarrow \boldsymbol{m}_{t-1}^{(s)}(\boldsymbol{g})$ // Replace gradient with momentum

4:     **end if**

5: **end for**

6: **return** $\boldsymbol{g}_t^{(s)}$ // Return updated gradients

---

**Algorithm 5** Adaptive IQR Threshold Adjustment

---

**Require:** History window $\mathcal{H}_l^{(s,i)}$ (or $\mathcal{G}_l^{(s,i)}$ for gradients) for stage $s$ and metric $i \in \mathcal{M}$, initial multiplier $k_0$, target false positive rate $\alpha$, growth factor $\gamma_g > 1$, shrink factor $\gamma_s < 1$, maximum iterations $N_{\max}$, minimum distance multipliers $\Lambda$

1: *// Calculate initial statistics*
2: $q_1, q_2, q_3 \leftarrow$ 25th, 50th, 75th percentiles of $\mathcal{H}_l^{(s,i)}$
3: $\text{IQR} \leftarrow \max(q_3 - q_1, \epsilon)$ *// Ensure non-zero IQR with small $\epsilon$*
4: $k \leftarrow k_0$ *// Initialize with previous multiplier value*
5: $\tau_{\text{lower}} \leftarrow q_2 - k \cdot \text{IQR}$
6: $\tau_{\text{upper}} \leftarrow q_2 + k \cdot \text{IQR}$
7: $\text{FP-rate} \leftarrow$ fraction of $\mathcal{H}_l^{(s,i)}$ outside $[\tau_{\text{lower}}, \tau_{\text{upper}}]$

8: *// Widen thresholds if false positive rate too high*
9: $\text{iter} \leftarrow 0$
10: **while** $\text{FP-rate} > \alpha$ and $\text{iter} < N_{\max}$ **do**
11: $\quad k \leftarrow k \cdot \gamma_g$ *// Grow multiplier*
12: $\quad \tau_{\text{lower}} \leftarrow q_2 - k \cdot \text{IQR}$
13: $\quad \tau_{\text{upper}} \leftarrow q_2 + k \cdot \text{IQR}$
14: $\quad \text{FP-rate} \leftarrow$ fraction of $\mathcal{H}_k^{(s,i)}$ outside $[\tau_{\text{lower}}, \tau_{\text{upper}}]$
15: $\quad \text{iter} \leftarrow \text{iter} + 1$
16: **end while**

17: *// Narrow thresholds if false positive rate too low*
18: $\text{iter} \leftarrow 0$
19: **while** $\text{FP-rate} \ll \alpha$ and $\text{iter} < N_{\max}$ **do**
20: $\quad k \leftarrow k \cdot \gamma_s$ *// Shrink multiplier*
21: $\quad \tau_{\text{lower}} \leftarrow q_2 - k \cdot \text{IQR}$
22: $\quad \tau_{\text{upper}} \leftarrow q_2 + k \cdot \text{IQR}$
23: $\quad \text{FP-rate} \leftarrow$ fraction of $\mathcal{H}_l^{(s,i)}$ outside $[\tau_{\text{lower}}, \tau_{\text{upper}}]$
24: $\quad \text{iter} \leftarrow \text{iter} + 1$
25: **end while**

26: *// Enforce minimum distance from median based on metric type (optional)*
27: $\lambda \leftarrow \Lambda[i]$ *// Get multiplier for current metric*
28: $d_{\min} \leftarrow |q_2| \cdot \lambda$ *// Minimum threshold distance*
29: $\tau_{\text{lower}} \leftarrow \min(\tau_{\text{lower}}, q_2 - d_{\min})$
30: $\tau_{\text{upper}} \leftarrow \max(\tau_{\text{upper}}, q_2 + d_{\min})$
31: **return** $\tau_{\text{lower}}, \tau_{\text{upper}}, k$

---

## D.2 On Handling the Cascading Effect

Pipeline parallelism exhibits distinct architectural characteristics compared to traditional federated learning (McMahan et al., 2017) approaches. One key challenge is what we term the "cascading effect" which occurs exclusively in pipeline parallelism. During forward propagation, a single node submitting malicious activations can contaminate all subsequent activations, potentially causing downstream verifier nodes to incorrectly flag benign nodes as malicious (see Fig. 5a). This phenomenon occurs similarly during backward propagation as depicted in Fig. 6b. The cascading effect could significantly increase false positive detection rates, making it critical to address this challenge.

To mitigate this issue, our approach (described in Sec. 3.1) implements inter-node communication protocols. Specifically, verifier nodes maintain a "tainted" list tracking upstream nodes identified as potentially malicious which they communicate with subsequent verifiers to prevent them from updating their EMAs and falsely flagging nodes affected by an attacker downstream. During backward propagation, all nodes sharing the same data parallel rank as the compromised node receive gradient momentum instead of actual gradients. Throughout this verification process, worker nodes continue processing data at a consistent pace, ensuring no node detects unusual behavioral patterns in the network.

The cascading effect manifests in both propagation directions, as malicious behavior can target either activation or gradient signals. We address this bidirectional vulnerability with two corresponding mitigation strategies:

1. When activations are compromised, all affected nodes receive gradient momentum (see Fig. 5);
2. When malicious behavior occurs during backward propagation, all downstream nodes switch to activation gradient (see Fig. 6).

While an alternative approach could involve sending zero vectors as gradients, this would effectively stall training in the affected pipe.[2] We leave exploration of appropriate gradient signals for the tainted segment to future work.

## D.3 On Natural Distribution Shift

Beyond malicious attacks, legitimate distribution shifts can occur naturally during training due to evolving data characteristics or model dynamics (Tian et al., 2023; Zhang et al., 2024). In such cases, multiple worker nodes at the same pipeline stage may simultaneously exhibit statistical deviations that would normally trigger malicious detection, despite all nodes behaving honestly.

To distinguish between natural distribution shifts and coordinated attacks, we implement a consensus-based approach at the verifier level. When more than 50% of nodes at a given pipeline stage are flagged as potentially malicious, the verifier attributes this to a natural distribution shift rather than malicious behavior. This threshold leverages the *honest majority assumption*: coordinated attacks involving more than half the nodes would violate our security model, making such scenarios indistinguishable from legitimate system-wide changes.

Upon detecting a natural distribution shift, the system responds as follows:

1. Training continues normally without malicious mitigation protocols.
2. The cascading effect mechanism described in App. D.2 is not activated.
3. Nodes update their EMA statistics to adapt to the new data distribution.

This consensus mechanism ensures that legitimate distributional changes do not trigger unnecessary verification overhead or training disruptions. However, refining this approach for asynchronous training environments (Ajanthan et al., 2025), where nodes may experience distribution shifts at different times, remains an important direction for future work.

---

[2]We will demonstrate that in our SWARM implementation in App. G, this choice does not have an impact on convergence as SWARM utilizes a stochastic routing.

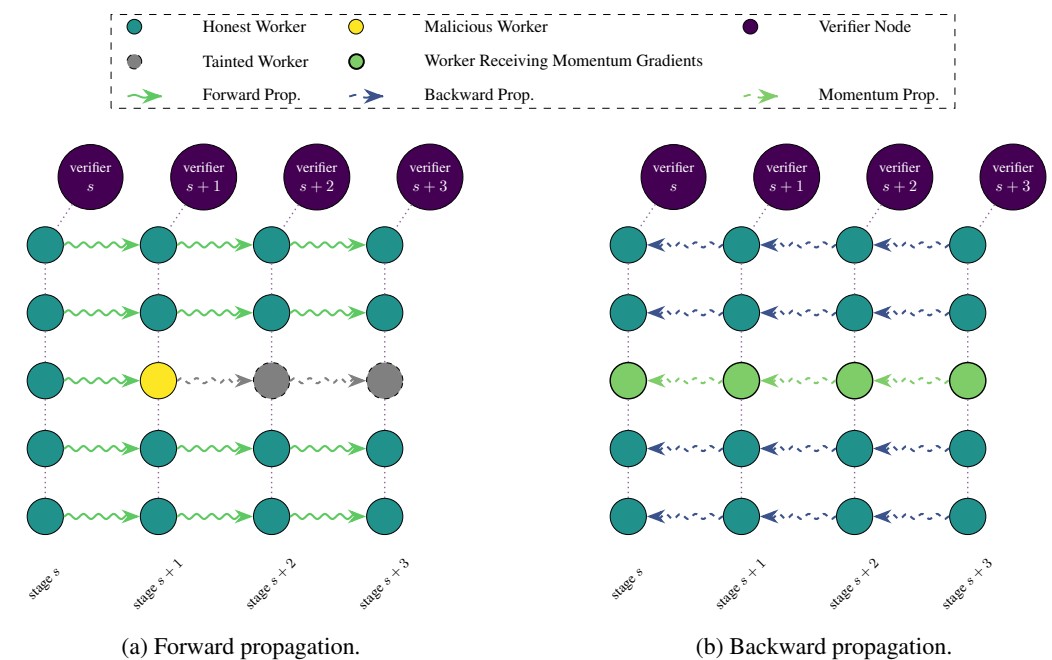

(a) Forward propagation.

(b) Backward propagation.

Figure 5: Verification protocol for handling compromised workers during distributed training. During forward propagation, a worker at stage $s+1$ is detected as potentially compromised (shown in yellow). The verifier nodes continue forwarding activations to subsequent stages without alerting downstream workers to avoid disrupting the pipeline. During backward propagation, instead of propagating gradients computed by the compromised worker, verifier nodes substitute gradient momentum values to maintain training stability. Communication flows through verifier nodes between consecutive pipeline stages, though direct worker-to-worker arrows are shown for visual clarity.

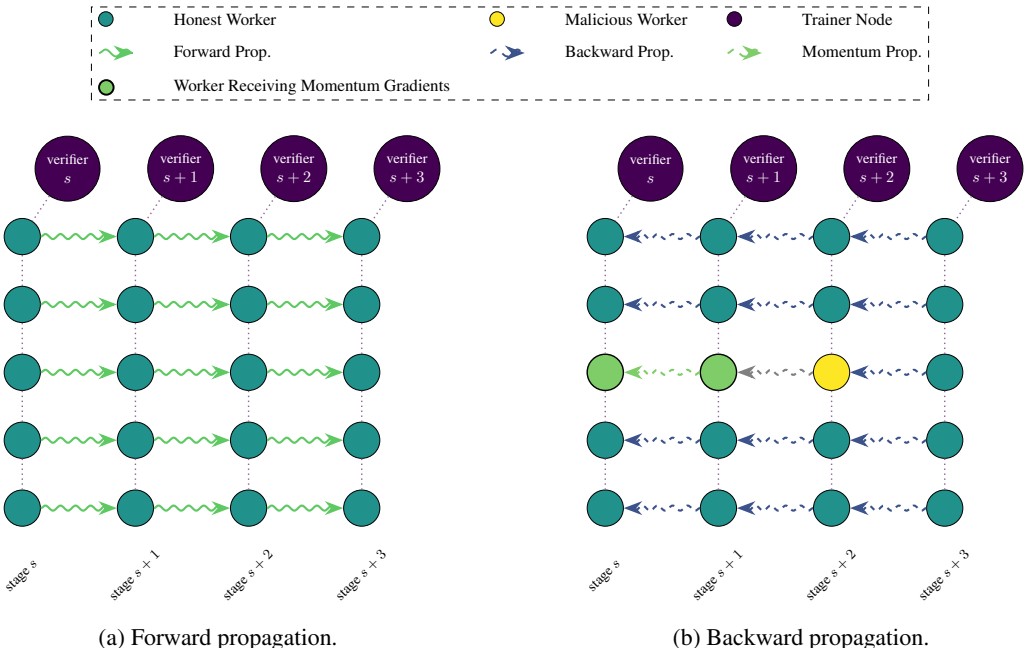

(a) Forward propagation.

(b) Backward propagation.

Figure 6: Verification protocol for handling compromised workers during gradient propagation. During backward propagation, a worker at stage $s+2$ is detected as potentially compromised (shown in yellow). To prevent propagation of tainted gradients, verifier nodes substitute gradient momentum values for all workers in preceding stages ($s+1$, $s$, etc.) instead of forwarding the corrupted gradients. This ensures training stability while maintaining the pipeline flow without alerting downstream workers to the compromise.

# E   THEORETICAL GUARANTEES AND THEIR PROOFS

## E.1   CONVERGENCE ANALYSIS

In this section, we present our full convergence analysis. Note that the bounds that we derive are in no means the tightest possible bounds. Instead, our aim is to establish a mathematical connection between our momentum-based verification and training dynamics.

### E.1.1   MALICIOUS DETECTION BOUNDS

Our first goal is to establish bounds on the maximum perturbation a malicious worker can introduce without being detected. We begin by analyzing how momentum smoothing affects the global deviation in activation vectors.[3]

**Lemma 2** (Momentum Smoothing Bounds the Global Deviation). *Let the activation vector momentum at stage $s$ and iteration $t$ be updated by*

$$\boldsymbol{m}_t^{(s)} = \beta_h \, \boldsymbol{m}_{t-1}^{(s)} + (1 - \beta_h) \left( \frac{1}{d} \sum_{r=1}^{d} \boldsymbol{h}_t^{(s,r)} \right), \qquad 0 \le \beta < 1, \tag{4}$$

*where $d$ represents the number of worker replicas at each stage. Assume:*

1. *A fraction $\gamma_s < \frac{1}{2}$ of the workers are malicious, with $B_s \subset \{1, 2, \dots, d\}$ denoting the subset of malicious workers.*

2. *A malicious worker adds a vector perturbation $\boldsymbol{\delta}_t^{(s,r)}$ satisfying*

$$\|\boldsymbol{\delta}_t^{(s,r)}\| \le \varepsilon. \tag{5}$$

*Then the deviation in the momentum caused by the malicious perturbations obeys*

$$\boxed{\|\Delta \boldsymbol{m}_t^{(s)}\| \le \gamma_s \, \varepsilon}, \tag{6}$$

*where $\Delta \boldsymbol{m}_t^{(s)}$ is the difference between the momentum computed with the malicious perturbations and the momentum computed using only the unperturbed (honest) activations.*

*Proof.* Let $H_s = \{1, 2, \dots, d\} \setminus B_s$ denote the set of honest workers at stage $s$. Since at most a fraction $\gamma_s$ of workers are malicious:

$$|B_s| = \gamma_s d, \qquad |H_s| = (1 - \gamma_s)d. \tag{7}$$

We can express each worker's activation vector as

$$\boldsymbol{h}_t^{(s,r)} = \boldsymbol{h}_{t,\text{nom}}^{(s,r)} + \boldsymbol{e}_t^{(s,r)}, \quad \text{where} \quad \boldsymbol{e}_t^{(s,r)} = \begin{cases} \boldsymbol{\delta}_t^{(s,r)}, & r \in B_s, \\ \boldsymbol{0}, & r \in H_s. \end{cases} \tag{8}$$

The nominal (unperturbed) average activation can be written as:

$$\bar{\boldsymbol{h}}_t^{(s)} = \frac{1}{d} \sum_{r=1}^{d} \boldsymbol{h}_{t,\text{nom}}^{(s,r)}. \tag{9}$$

For the observed (perturbed) average, we have:

$$\begin{aligned} \hat{\boldsymbol{h}}_t^{(s)} &= \frac{1}{d} \sum_{r=1}^{d} (\boldsymbol{h}_{t,\text{nom}}^{(s,r)} + \boldsymbol{e}_t^{(s,r)}) \\ &= \bar{\boldsymbol{h}}_t^{(s)} + \frac{1}{d} \sum_{r \in B_s} \boldsymbol{\delta}_t^{(s,r)}. \end{aligned} \tag{10}$$

---

[3]Even though we present our theory for the activation manipulation, our results are easily extendable to the gradient manipulation as well.

For the momentum terms, we can write:

$$\boldsymbol{m}_{t,\text{obs}}^{(s)} = \beta_h \boldsymbol{m}_{t-1,\text{obs}}^{(s)} + (1 - \beta_h)\hat{\boldsymbol{h}}_t^{(s)},$$
$$\boldsymbol{m}_{t,\text{nom}}^{(s)} = \beta_h \boldsymbol{m}_{t-1,\text{nom}}^{(s)} + (1 - \beta_h)\bar{\boldsymbol{h}}_t^{(s)}. \tag{11}$$

The deviation in the momentum at iteration $t$ is:

$$\begin{aligned}
\Delta \boldsymbol{m}_t^{(s)} &= \boldsymbol{m}_{t,\text{obs}}^{(s)} - \boldsymbol{m}_{t,\text{nom}}^{(s)} \\
&= \beta_h(\boldsymbol{m}_{t-1,\text{obs}}^{(s)} - \boldsymbol{m}_{t-1,\text{nom}}^{(s)}) + (1 - \beta_h)(\hat{\boldsymbol{h}}_t^{(s)} - \bar{\boldsymbol{h}}_t^{(s)}) \\
&= \beta_h \Delta \boldsymbol{m}_{t-1}^{(s)} + (1 - \beta_h)\frac{1}{d} \sum_{r \in B_s} \boldsymbol{\delta}_t^{(s,r)}.
\end{aligned} \tag{12}$$

Assuming $\Delta \boldsymbol{m}_0^{(s)} = \boldsymbol{0}$, we can solve this recurrence relation:

$$\Delta \boldsymbol{m}_t^{(s)} = (1 - \beta_h) \sum_{j=1}^{t} \beta_h^{t-j} \frac{1}{d} \sum_{r \in B_s} \boldsymbol{\delta}_j^{(s,r)}. \tag{13}$$

Taking the norm and applying the triangle inequality, we have:

$$\begin{aligned}
\|\Delta \boldsymbol{m}_t^{(s)}\| &\le (1 - \beta_h) \sum_{j=1}^{t} \beta_h^{t-j} \frac{1}{d} \sum_{r \in B_s} \|\boldsymbol{\delta}_j^{(s,r)}\| \\
&\le (1 - \beta_h) \sum_{j=1}^{t} \beta_h^{t-j} \frac{|B_s|}{d} \varepsilon \\
&= (1 - \beta_h) \gamma_s \varepsilon \sum_{j=1}^{t} \beta_h^{t-j} \\
&= (1 - \beta_h) \gamma_s \varepsilon \frac{1 - \beta_h^t}{1 - \beta_h} \\
&\le (1 - \beta_h) \gamma_s \varepsilon \frac{1}{1 - \beta_h} \\
&= \gamma_s \varepsilon.
\end{aligned} \tag{14}$$

which establishes the stated bound. For the case where we consider only the most recent iteration's effect (equivalent to initializing $\boldsymbol{m}_{t-1,\text{obs}}^{(s)} = \boldsymbol{m}_{t-1,\text{nom}}^{(s)}$), we have:

$$\begin{aligned}
\|\Delta \boldsymbol{m}_t^{(s)}\| &= \left\| (1 - \beta)\frac{1}{d} \sum_{r \in B_s} \boldsymbol{\delta}_t^{(s,r)} \right\| \\
&\le (1 - \beta)\frac{|B_s|}{d} \varepsilon \\
&= (1 - \beta)\gamma_s \varepsilon.
\end{aligned} \tag{15}$$

$$\square$$

Lemma 2 establishes a key property of momentum-based smoothing: it naturally attenuates the impact of malicious perturbations. This attenuation is proportional to the fraction of malicious workers $\gamma_s$, demonstrating that smaller malicious coalitions have less impact on the global state. This result is critical for understanding how effectively the system can contain malicious influence.

Building on this foundation, we now analyze how these bounded perturbations affect our detection statistics:

**Lemma 3** (Test Statistic Deviation). *For a metric function $\Omega$, assume that the detector computes*

$$\Gamma_t^{(s,r)} = \left\| \Omega(\boldsymbol{h}_t^{(s,r)}, \boldsymbol{m}_{t-1}^{(s)}) - \Omega_{\text{ref}}^{(s)} \right\| \tag{16}$$

*where $\Omega_{\text{ref}}^{(s)}$ is a reference statistic computed by the trusted trainer nodes (e.g., our median based reference statistic).*

*Assume for every worker replica $r \in \{1, 2, \ldots, d\}$:*

- *Activation perturbation: $\|\boldsymbol{\delta}_t^{(s,r)}\| \leq \varepsilon$.*
- *Momentum update: $\boldsymbol{m}_t^{(s)} = \beta_h \boldsymbol{m}_{t-1}^{(s)} + (1 - \beta_h) \frac{1}{d} \sum_{r=1}^{d} \boldsymbol{h}_t^{(s,r)}$ with at most a fraction $\gamma_s$ malicious workers (from Lemma 2)*

$$\|\Delta \boldsymbol{m}_t^{(s)}\| \leq \gamma_s \varepsilon. \tag{17}$$

- *Lipschitz continuity of $\Omega$: For any inputs $\boldsymbol{x}$, $\boldsymbol{y}$ and perturbations $\boldsymbol{\delta}_x$, $\boldsymbol{\delta}_y$:*

$$\left\| \Omega(\boldsymbol{x} + \boldsymbol{\delta}_x, \boldsymbol{y} + \boldsymbol{\delta}_y) - \Omega(\boldsymbol{x}, \boldsymbol{y}) \right\| \leq L_\Omega \left( \|\boldsymbol{\delta}_x\| + \|\boldsymbol{\delta}_y\| \right), \tag{18}$$

*where $L_\Omega$ is the Lipschitz constant of $\Omega$.*

*Define the (possibly known) baseline gap as:*

$$\delta^{\text{base}} = \Omega(\boldsymbol{h}_{t,\text{honest}}^{(s,r)}, \boldsymbol{m}_{t-1,\text{honest}}^{(s)}). \tag{19}$$

*Then, the test statistic satisfies:*

$$\boxed{\Gamma_t^{(s,r)} \leq \delta_{\text{avg}}^{\text{base}} + L_\Omega \varepsilon + L_\Omega \gamma_s \varepsilon} \tag{20}$$

*Proof.* For an activation vector with malicious perturbation $\boldsymbol{\delta}_t^{(s,r)}$, using the Lipschitz property of $\Omega$, we have:

$$\begin{aligned} \left\| \Omega(\boldsymbol{h}_t^{(s,r)}, \boldsymbol{m}_{t-1}^{(s)}) - \Omega(\boldsymbol{h}_{t,\text{honest}}^{(s,r)}, \boldsymbol{m}_{t-1}^{(s)}) \right\| &\leq L_\Omega \|\boldsymbol{\delta}_t^{(s,r)}\| \\ &\leq L_\Omega \varepsilon. \end{aligned} \tag{21}$$

From Lemma 2, we know that the momentum vector deviation is bounded by $\|\Delta \boldsymbol{m}_{t-1}^{(s)}\| \leq \gamma_s \varepsilon$. Thus, applying the Lipschitz property again:

$$\begin{aligned} \left\| \Omega(\boldsymbol{h}_{t,\text{honest}}^{(s,r)}, \boldsymbol{m}_{t-1}^{(s)}) - \Omega(\boldsymbol{h}_{t,\text{honest}}^{(s,r)}, \boldsymbol{m}_{t-1,\text{honest}}^{(s)}) \right\| &\leq L_\Omega \|\Delta \boldsymbol{m}_{t-1}^{(s)}\| \\ &\leq L_\Omega \gamma_s \varepsilon. \end{aligned} \tag{22}$$

We can now decompose the test statistic using the triangle inequality:

$$\begin{aligned} \Gamma_t^{(s,r)} &= \left\| \Omega(\boldsymbol{h}_t^{(s,r)}, \boldsymbol{m}_{t-1}^{(s)}) - \Omega_{\text{ref}}^{(s)} \right\| \\ &= \left\| \Omega(\boldsymbol{h}_t^{(s,r)}, \boldsymbol{m}_{t-1}^{(s)}) - \Omega(\boldsymbol{h}_{t,\text{honest}}^{(s,r)}, \boldsymbol{m}_{t-1}^{(s)}) \right. \\ &\quad + \Omega(\boldsymbol{h}_{t,\text{honest}}^{(s,r)}, \boldsymbol{m}_{t-1}^{(s)}) - \Omega(\boldsymbol{h}_{t,\text{honest}}^{(s,r)}, \boldsymbol{m}_{t-1,\text{honest}}^{(s)}) \\ &\quad \left. + \Omega(\boldsymbol{h}_{t,\text{honest}}^{(s,r)}, \boldsymbol{m}_{t-1,\text{honest}}^{(s)}) - \Omega_{\text{ref}}^{(s)} \right\| \\ &\leq \left\| \Omega(\boldsymbol{h}_t^{(s,r)}, \boldsymbol{m}_{t-1}^{(s)}) - \Omega(\boldsymbol{h}_{t,\text{honest}}^{(s,r)}, \boldsymbol{m}_{t-1}^{(s)}) \right\| \\ &\quad + \left\| \Omega(\boldsymbol{h}_{t,\text{honest}}^{(s,r)}, \boldsymbol{m}_{t-1}^{(s)}) - \Omega(\boldsymbol{h}_{t,\text{honest}}^{(s,r)}, \boldsymbol{m}_{t-1,\text{honest}}^{(s)}) \right\| \\ &\quad + \left\| \Omega(\boldsymbol{h}_{t,\text{honest}}^{(s,r)}, \boldsymbol{m}_{t-1,\text{honest}}^{(s)}) - \Omega_{\text{ref}}^{(s)} \right\| \\ &\leq L_\Omega \varepsilon + L_\Omega \gamma_s \varepsilon + \delta^{\text{base}} \end{aligned} \tag{23}$$

If the detector compensates for (or ignores) the baseline gap $\delta^{\text{base}}$ and raises an alarm when $\Gamma_t^{(s,r)} > \tau$, the additional deviation attributable only to malicious perturbations is:

$$\Gamma_{\text{pert}} \leq L_\Omega \varepsilon + L_\Omega \gamma_s \varepsilon = L_\Omega (1 + \gamma_s) \varepsilon, \tag{24}$$

so a malicious worker can remain undetected provided:

$$L_\Omega(1 + \gamma_s)\varepsilon \leq \tau, \quad \Longrightarrow \quad \boxed{\varepsilon \leq \frac{\tau}{L_\Omega(1 + \gamma_s)}} \tag{25}$$

which completes the proof. □

Lemma 3 provides a crucial bound on the test statistic deviation under malicious perturbations. The bound depends on two key factors: (1) the Lipschitz constant $L_\Omega$ of the test function and (2) the fraction of malicious workers $\gamma_s$. The practical implication is that a malicious worker can remain undetected only if its perturbation magnitude satisfies:

$$\varepsilon \leq \frac{\tau}{L_\Omega(1 + \gamma_s)} \tag{26}$$

This establishes a direct relationship between the detection threshold $\tau$ and the maximum undetectable perturbation magnitude. This equation demonstrates how tuning $\tau$ affects the security-performance tradeoff: with lower thresholds we can provide stronger security guarantees at the potential cost of increased false positives. This highlights the importance of setting an appropriate threshold for the test statistic.

### E.1.2 GRADIENT PERTURBATION ANALYSIS

Now that we have established bounds on undetectable activation perturbations, we analyze how these perturbations propagate through the network to affect parameter gradients. This analysis is critical for understanding the impact on training dynamics.

**Lemma 4** (Per–stage Lipschitz constants). *Assume replica $r$ of stage $s$ implements a map*

$$\boldsymbol{h}^{(s,r)} = f_s(\boldsymbol{h}^{(s-1,r)}; \boldsymbol{\theta}^{(s)})$$

*whose Jacobians satisfy*

$$\|\partial_{\boldsymbol{\theta}} f_s\| \leq L_{\boldsymbol{\theta}}^{(s)}, \qquad \|\partial_{\boldsymbol{h}} f_s\| \leq L_f^{(s)}.$$

*Then, the parameter gradient of stage $s$ obeys*

$$\|\nabla_{\boldsymbol{\theta}^{(s)}}^{agg} \mathcal{L}(\boldsymbol{\theta})\| \leq L_{\boldsymbol{\theta}}^{(s)} \frac{1}{d} \sum_{r=1}^{d} \|\boldsymbol{g}^{(s,r)}\|,$$

*where $\boldsymbol{g}^{(s+1,r)}$ is the gradient with respect to the activation $\boldsymbol{h}^{(s,r)}$.*

*Proof.* The loss $\mathcal{L}(\boldsymbol{\theta})$ depends on $\boldsymbol{\theta}^{(s)}$ only through the composition of stage maps:

$$\boldsymbol{h}^{(s,r)} = f_s(\boldsymbol{h}^{(s-1,r)}; \boldsymbol{\theta}^{(s)})$$
$$\boldsymbol{h}^{(s+1,r)} = f_{s+1}(\boldsymbol{h}^{(s,r)}; \boldsymbol{\theta}^{(s+1)})$$
$$\vdots \tag{27}$$
$$\boldsymbol{h}^{(p,r)} = f_p(\boldsymbol{h}^{(p-1,r)}; \boldsymbol{\theta}^{(p)})$$

followed by a readout $\mathcal{L}_{\text{head}}(\boldsymbol{h}^{(p,r)})$.

Applying the chain rule yields

$$\nabla_{\boldsymbol{\theta}^{(s,r)}} \mathcal{L}(\boldsymbol{\theta}) = \partial_{\boldsymbol{\theta}} f_s(\partial_{\boldsymbol{h}} f_{s+1}) \cdots (\partial_{\boldsymbol{h}} f_p) \nabla_{\boldsymbol{h}^{(p,r)}} \mathcal{L}_{\text{head}}. \tag{28}$$

Here, each $\partial_{\boldsymbol{\theta}} f_s$ is evaluated at $(\boldsymbol{h}^{(s-1,r)}, \boldsymbol{\theta}^{(s)})$ and each $\partial_{\boldsymbol{h}} f_j$ at $(\boldsymbol{h}^{(j-1,r)}, \boldsymbol{\theta}^{(j)})$. Define

$$\boldsymbol{g}^{(s+1,r)} := (\partial_{\boldsymbol{h}} f_{s+1}) \cdots (\partial_{\boldsymbol{h}} f_p) \nabla_{\boldsymbol{h}^{(p,r)}} \mathcal{L}_{\text{head}}, \tag{29}$$

so that $\boldsymbol{g}^{(s+1,r)}$ is precisely the activation gradient that enters replica $r$ of stage $s$ during back-propagation. The all-reduce operation aggregates all gradients from the $d$ replicas of stage $s$

before applying them, i.e.,

$$\nabla_{\boldsymbol{\theta}^{(s)}}^{\mathrm{agg}} \mathcal{L}(\boldsymbol{\theta}) = \frac{1}{d} \sum_{r=1}^{d} \nabla_{\boldsymbol{\theta}^{(s,r)}} \mathcal{L}(\boldsymbol{\theta}) \tag{30}$$

Taking Euclidean norms, applying the triangle inequality, and using sub-multiplicativity of the operator norm yields

$$\|\nabla_{\boldsymbol{\theta}^{(s)}}^{\mathrm{agg}} \mathcal{L}(\boldsymbol{\theta})\| \leq \frac{1}{d} \sum_{r=1}^{d} \|\nabla_{\boldsymbol{\theta}^{(s,r)}} \mathcal{L}(\boldsymbol{\theta})\|$$

$$\leq \frac{1}{d} \sum_{r=1}^{d} \|\partial_{\boldsymbol{\theta}} f_s\| \, \|\boldsymbol{g}^{(s+1,r)}\| \tag{31}$$

$$\leq L_{\boldsymbol{\theta}}^{(s)} \frac{1}{d} \sum_{r=1}^{d} \|\boldsymbol{g}^{(s+1,r)}\|.$$

Note that the step in Eq. (30) follow the fact that the Lipschitz constant assumptions are uniform over the data distribution. □

Lemma 4 characterizes how strongly the parameter gradients at each stage depend on activation perturbations. The Lipschitz constants $L_{\boldsymbol{\theta}}^{(s)}$ and $L_f^{(s)}$ quantify this relationship, providing a foundation for understanding gradient sensitivity. These stage-specific Lipschitz constants are important because they reveal which stages of the model are most vulnerable to malicious manipulation. Stages with larger constants amplify perturbations more strongly, making them prime targets for attackers and priority areas for enhanced monitoring.

Building on these Lipschitz properties, we now quantify exactly how activation perturbations translate to gradient perturbations:

**Lemma 5** (Sensitivity of Parameter Gradient to Activation Perturbation). *Let an expected honest replica in stage $s$ be activation $\boldsymbol{h}^{(s,r)}$. Assume that a malicious worker replaces it by $\boldsymbol{h}^{(s,r)} + \boldsymbol{\delta}$. If the activation perturbations are small such that changing the input activation by $\boldsymbol{\delta}$ perturbs $\boldsymbol{g}^{(s+1,r)}$ through the local Jacobian only, then the change in the* aggregated *stage $s$ parameter gradient satisfies*

$$\|\Delta \nabla_{\boldsymbol{\theta}^{(s)}}^{agg} \mathcal{L}(\boldsymbol{\theta})\| := \|\nabla_{\boldsymbol{\theta}^{(s)}}^{agg} \mathcal{L}(\boldsymbol{\theta} \mid \boldsymbol{h}^{(s,r)} + \boldsymbol{\delta}) - \nabla_{\boldsymbol{\theta}^{(s)}}^{agg} \mathcal{L}(\boldsymbol{\theta} \mid \boldsymbol{h}^{(s,r)})\| \leq \frac{G_s}{d} \|\boldsymbol{\delta}\|,$$

*where $G_s := L_{\boldsymbol{\theta}}^{(s)} \left( \prod_{j \geq s+1} L_f^{(j)} \right)$. More generally, if a set $B_s$ of $|B_s| = \gamma_s \cdot d$ malicious replicas each injects a perturbation of norm at most $\|\boldsymbol{\delta}\|$, then*

$$\boxed{\|\Delta \nabla_{\boldsymbol{\theta}^{(s)}}^{agg}\| \leq \gamma_s \cdot G_s \cdot \varepsilon.} \tag{32}$$

*Proof.* Let us first consider a single replica $r$ at stage $s$. For this replica, using Lemma 4 we can write:

$$\nabla_{\boldsymbol{\theta}^{(s,r)}} \mathcal{L}(\boldsymbol{\theta}) = \partial_{\boldsymbol{\theta}} f_s \, \boldsymbol{g}^{(s+1,r)}, \quad \text{s.t. } \boldsymbol{g}^{(s+1,r)} = \left( \prod_{j=s+1}^{p} \partial_{\boldsymbol{h}} f_j \right) \nabla_{\boldsymbol{h}^{(p,r)}} \mathcal{L}_{\mathrm{head}}.$$

Thus, assuming a linearization of the change in gradient signal under small input perturbation, we can write:

$$\delta \boldsymbol{g}^{(s+1,r)} = \left( \prod_{j=s+1}^{p} \partial_{\boldsymbol{h}} f_j \right) \boldsymbol{\delta}.$$

Hence, for replica $r$ we can write the sensitivity of the parameter gradient as:

$$\Delta \nabla_{\boldsymbol{\theta}^{(s,r)}} \mathcal{L}(\boldsymbol{\theta}) = (\partial_{\boldsymbol{\theta}} f_s) \, \delta \boldsymbol{g}^{(s+1,r)} = \partial_{\boldsymbol{\theta}} f_s \left( \prod_{j=s+1}^{p} \partial_{\boldsymbol{h}} f_j \right) \boldsymbol{\delta}.$$

Using sub-multiplicativity and the Lipschitz bounds, we have

$$
\begin{aligned}
\|\Delta \nabla_{\boldsymbol{\theta}^{(s)}} \mathcal{L}(\boldsymbol{\theta})\| &\leq \|\partial_{\boldsymbol{\theta}} f_s\| \Big( \prod_{j=s+1}^{p} \|\partial_{\boldsymbol{h}} f_j\| \Big) \|\boldsymbol{\delta}\| \\
&\leq L_{\boldsymbol{\theta}}^{(s)} \Big( \prod_{j \geq s+1} L_f^{(j)} \Big) \|\boldsymbol{\delta}\| \\
&= G_s \|\boldsymbol{\delta}\|
\end{aligned}
\tag{33}
$$

Since the stage update uses the aggregate gradients, we can write

$$
\Delta \nabla_{\boldsymbol{\theta}^{(s)}}^{\mathrm{agg}} \mathcal{L}(\boldsymbol{\theta}) = \frac{1}{d} \Delta \nabla_{\boldsymbol{\theta}^{(s,r)}} \mathcal{L}(\boldsymbol{\theta}).
$$

Hence, $\|\Delta \nabla_{\boldsymbol{\theta}^{(s)}}^{\mathrm{agg}} \mathcal{L}(\boldsymbol{\theta})\| \leq \frac{G_s}{d} \|\boldsymbol{\delta}\|$. If $|B_s|$ replicas are corrupted, we would have

$$
\begin{aligned}
\|\Delta \nabla_{\boldsymbol{\theta}^{(s)}}^{\mathrm{agg}} \mathcal{L}(\boldsymbol{\theta})\| &\leq \frac{1}{d} \sum_{r \in B_s} \|\Delta \nabla_{\boldsymbol{\theta}^{(s,r)}} \mathcal{L}(\boldsymbol{\theta})\| \\
&\leq \frac{|B_s|}{d} \cdot G_s \cdot \|\boldsymbol{\delta}\| \\
&= \gamma_s \cdot G_s \cdot \|\boldsymbol{\delta}\|,
\end{aligned}
\tag{34}
$$

and the proof is complete. $\square$

Lemma 5 provides the crucial link between activation perturbations and their impact on parameter gradients. The amplification factor $G_s$ represents how perturbations at stage $s$ propagate through the network during backpropagation. This factor depends on both the local parameter gradient sensitivity ($L_{\boldsymbol{\theta}}^{(s)}$) and the product of activation gradient sensitivities in subsequent stages ($\prod_{j \geq s+1} L_f^{(j)}$). This result has important implications for robustness against malicious workers in pipeline-parallel training:

1. Earlier stages (lower $s$) typically have larger amplification factors because perturbations must propagate through more subsequent stages.
2. Stages with larger parameter counts or complex activation patterns may have higher individual Lipschitz constants.
3. The fractional impact of malicious workers is reduced by the averaging effect of the all-reduce operation, as captured by the $\gamma_s$ factor.

Combined with our detection bounds, we can now establish the maximum parameter gradient perturbation that can be induced by undetected malicious workers:

$$
\|\Delta \nabla_{\boldsymbol{\theta}^{(s)}}^{\mathrm{agg}} \mathcal{L}(\boldsymbol{\theta})\| \leq \gamma_s \cdot G_s \cdot \frac{\tau}{L_{\Omega}(1 + \gamma_s)} := \zeta
\tag{35}
$$

This bound directly links detection thresholds to gradient perturbations, which will be essential for our convergence analysis.

### E.1.3 CONVERGENCE UNDER PERTURBED GRADIENTS

Having established bounds on gradient perturbations, we now analyze how these perturbations affect the convergence properties of momentum-SGD. We consider general non-convex loss functions, but our results can be easily extended to the strongly convex case.

**Theorem 2** (Convergence of Momentum SGD under Smoothness for Convex and Non-convex Cases with Perturbation and Noise)**.** *Consider the balanced momentum update:*

$$
\begin{aligned}
\boldsymbol{v}_{t+1} &= \beta \boldsymbol{v}_t + (1 - \beta) \boldsymbol{g}_t, \\
\boldsymbol{\theta}_{t+1} &= \boldsymbol{\theta}_t - \eta \boldsymbol{v}_{t+1}
\end{aligned}
\tag{36}
$$

*where $\boldsymbol{g}_t = \nabla \mathcal{L}(\boldsymbol{\theta}_t) + \boldsymbol{\zeta}_t + \boldsymbol{\xi}_t$, with a Lyapunov function $\Psi_t = \mathcal{L}(\boldsymbol{\theta}_t) + c\|\boldsymbol{v}_t\|^2$ for some constant $c > 0$.*
*Assume:*

1. $\mathcal{L}$ is L-smooth but potentially non-convex
2. $\mathcal{L}$ is bounded below by $\mathcal{L}^*$
3. $\beta \in [0, 1)$ is the momentum parameter
4. $\eta > 0$ is the learning rate
5. $\boldsymbol{\zeta}_t$ is a deterministic perturbation with maximum perturbation norm $\|\boldsymbol{\zeta}_t\| \leq \zeta$, and $\boldsymbol{\xi}_t$ is zero-mean noise with $\mathbb{E}[\|\boldsymbol{\xi}_t\|^2] \leq \sigma^2$

*For any positive constants $\varepsilon_1, \varepsilon_2, \varepsilon_3$ conditioned on the past $\mathcal{F}_t$, we have:*

$$\mathbb{E}[\Psi_{t+1}|\mathcal{F}_t] \leq \Psi_t - \alpha\|\nabla\mathcal{L}(\boldsymbol{\theta}_t)\|^2 + C_1\|\boldsymbol{v}_t\|^2 + C_2\|\boldsymbol{\zeta}_t\|^2 + D\sigma^2 \tag{37}$$

*Where the constants are given by:*

$$\alpha = \eta(1-\beta)\left(1 - \varepsilon_2 - \frac{\beta}{4\varepsilon_1(1-\beta)} - \frac{2}{\eta}\left(\frac{\eta^2 L}{2} + c\right)\left(1 - \beta + \frac{\beta}{4\varepsilon_1}\right)\right)$$

$$C_1 = \left(\eta\beta\varepsilon_1 + \left(\frac{\eta^2 L}{2} + c\right)\beta\left(\beta + 2(1-\beta)(\varepsilon_1 + \varepsilon_3)\right) - c\right)$$

$$C_2 = \left(\frac{\eta(1-\beta)}{4\varepsilon_2} + 2\left(\frac{\eta^2 L}{2} + c\right)(1-\beta)\left(1 - \beta - \frac{\beta}{4\varepsilon_3}\right)\right) \tag{38}$$

$$D = \left(\frac{\eta^2 L}{2} + c\right)(1-\beta)^2.$$

*If we choose appropriate values for $\varepsilon_1, \varepsilon_2, \varepsilon_3$ such that $\alpha > 0$ and $C_1 < 0$, and assume $\boldsymbol{v}_0 = \boldsymbol{0}$, then the algorithm converges in expectation to a neighborhood of a stationary point:*

$$\boxed{\frac{1}{T}\sum_{t=0}^{T-1}\mathbb{E}[\|\nabla\mathcal{L}(\boldsymbol{\theta}_t)\|^2] \leq \frac{\mathcal{L}_0 - \mathcal{L}^*}{\alpha T} + \frac{C_2\zeta^2 + D\sigma^2}{\alpha},} \tag{39}$$

*where $\mathcal{L}_0 := \mathcal{L}(\boldsymbol{\theta}_0)$ is our loss value at initialization.*

*Proof.* We begin by analyzing one-step progress with the Lyapunov potential function $\Psi_t = \mathcal{L}(\boldsymbol{\theta}_t) + c\|\boldsymbol{v}_t\|^2$ inspired by (Liu et al., 2020; Mai & Johansson, 2020).

**Evolution of the Loss Term.** By the $L$-smoothness of $\mathcal{L}$, we have:

$$\mathcal{L}(\boldsymbol{\theta}_{t+1}) \leq \mathcal{L}(\boldsymbol{\theta}_t) + \langle\nabla\mathcal{L}(\boldsymbol{\theta}_t), \boldsymbol{\theta}_{t+1} - \boldsymbol{\theta}_t\rangle + \frac{L}{2}\|\boldsymbol{\theta}_{t+1} - \boldsymbol{\theta}_t\|^2$$
$$= \mathcal{L}(\boldsymbol{\theta}_t) - \eta\langle\nabla\mathcal{L}(\boldsymbol{\theta}_t), \boldsymbol{v}_{t+1}\rangle + \frac{\eta^2 L}{2}\|\boldsymbol{v}_{t+1}\|^2 \tag{40}$$

Now, expanding $\Psi_{t+1} - \Psi_t = [\mathcal{L}(\boldsymbol{\theta}_{t+1}) - \mathcal{L}(\boldsymbol{\theta}_t)] + c[\|\boldsymbol{v}_{t+1}\|^2 - \|\boldsymbol{v}_t\|^2]$, we have:

$$\Psi_{t+1} - \Psi_t = [\mathcal{L}(\boldsymbol{\theta}_{t+1}) - \mathcal{L}(\boldsymbol{\theta}_t)] + c[\|\boldsymbol{v}_{t+1}\|^2 - \|\boldsymbol{v}_t\|^2]$$
$$= -\eta\langle\nabla\mathcal{L}(\boldsymbol{\theta}_t), \boldsymbol{v}_{t+1}\rangle + \frac{\eta^2 L}{2}\|\boldsymbol{v}_{t+1}\|^2 + c\|\boldsymbol{v}_{t+1}\|^2 - c\|\boldsymbol{v}_t\|^2 \tag{41}$$
$$= -\eta\langle\nabla\mathcal{L}(\boldsymbol{\theta}_t), \boldsymbol{v}_{t+1}\rangle + \left(\frac{\eta^2 L}{2} + c\right)\|\boldsymbol{v}_{t+1}\|^2 - c\|\boldsymbol{v}_t\|^2.$$

Substitute $\boldsymbol{v}_{t+1} = \beta\boldsymbol{v}_t + (1-\beta)\boldsymbol{g}_t$ (Polyak, 1964), then we have:

$$\Psi_{t+1} - \Psi_t \leq -\eta\langle\nabla\mathcal{L}(\boldsymbol{\theta}_t), \boldsymbol{v}_{t+1}\rangle + \left(\frac{\eta^2 L}{2} + c\right)\|\boldsymbol{v}_{t+1}\|^2 - c\|\boldsymbol{v}_t\|^2$$
$$= -\eta\langle\nabla\mathcal{L}(\boldsymbol{\theta}_t), \beta\boldsymbol{v}_t + (1-\beta)\boldsymbol{g}_t\rangle + \left(\frac{\eta^2 L}{2} + c\right)\|\beta\boldsymbol{v}_t + (1-\beta)\boldsymbol{g}_t\|^2 - c\|\boldsymbol{v}_t\|^2$$
$$= -\eta\beta\langle\nabla\mathcal{L}(\boldsymbol{\theta}_t), \boldsymbol{v}_t\rangle - \eta(1-\beta)\langle\nabla\mathcal{L}(\boldsymbol{\theta}_t), \boldsymbol{g}_t\rangle \tag{42}$$
$$+ \left(\frac{\eta^2 L}{2} + c\right)\left(\beta^2\|\boldsymbol{v}_t\|^2 + (1-\beta)^2\|\boldsymbol{g}_t\|^2 + 2\beta(1-\beta)\langle\boldsymbol{v}_t, \boldsymbol{g}_t\rangle\right) - c\|\boldsymbol{v}_t\|^2$$

Next, we bound individual terms.

**Bounding $-\eta\beta\langle\nabla\mathcal{L}(\boldsymbol{\theta}_t),\boldsymbol{v}_t\rangle$.** Using Young's inequality with parameter $\varepsilon_1 > 0$, we write:

$$-\eta\beta\langle\boldsymbol{v}_t,\nabla\mathcal{L}(\boldsymbol{\theta}_t)\rangle \leq \eta\beta\varepsilon_1\|\boldsymbol{v}_t\|^2 + \frac{\eta\beta}{4\varepsilon_1}\|\nabla\mathcal{L}(\boldsymbol{\theta}_t)\|^2. \tag{43}$$

**Bounding $-\eta(1-\beta)\langle\nabla\mathcal{L}(\boldsymbol{\theta}_t),\boldsymbol{g}_t\rangle$.** Since $\boldsymbol{g}_t = \nabla\mathcal{L}(\boldsymbol{\theta}_t) + \boldsymbol{\zeta}_t + \boldsymbol{\xi}_t$, we can write:

$$\begin{aligned}
-\eta(1-\beta)\langle\nabla\mathcal{L}(\boldsymbol{\theta}_t),\boldsymbol{g}_t\rangle &= -\eta(1-\beta)\langle\nabla\mathcal{L}(\boldsymbol{\theta}_t),\nabla\mathcal{L}(\boldsymbol{\theta}_t)+\boldsymbol{\zeta}_t+\boldsymbol{\xi}_t\rangle \\
&\leq -\eta(1-\beta)\|\nabla\mathcal{L}(\boldsymbol{\theta}_t)\|^2 - \eta(1-\beta)\langle\nabla\mathcal{L}(\boldsymbol{\theta}_t),\boldsymbol{\zeta}_t\rangle \\
&\quad - \eta(1-\beta)\langle\nabla\mathcal{L}(\boldsymbol{\theta}_t),\boldsymbol{\xi}_t\rangle \\
&\leq -\eta(1-\beta)\|\nabla\mathcal{L}(\boldsymbol{\theta}_t)\|^2 + \eta(1-\beta)\varepsilon_2\|\nabla\mathcal{L}(\boldsymbol{\theta}_t)\|^2 + \frac{\eta(1-\beta)}{4\varepsilon_2}\|\boldsymbol{\zeta}_t\|^2 \\
&\quad - \eta(1-\beta)\langle\nabla\mathcal{L}(\boldsymbol{\theta}_t),\boldsymbol{\xi}_t\rangle
\end{aligned} \tag{44}$$

**Bounding $\|\boldsymbol{g}_t\|^2$.** For this term, we write:

$$\begin{aligned}
\|\boldsymbol{g}_t\|^2 &= \|\nabla\mathcal{L}(\boldsymbol{\theta}_t)+\boldsymbol{\zeta}_t+\boldsymbol{\xi}_t\|^2 \\
&= \|\nabla\mathcal{L}(\boldsymbol{\theta}_t)+\boldsymbol{\zeta}_t\|^2 + \|\boldsymbol{\xi}_t\|^2 + 2\langle\nabla\mathcal{L}(\boldsymbol{\theta}_t)+\boldsymbol{\zeta}_t,\boldsymbol{\xi}_t\rangle \\
&\leq 2\|\nabla\mathcal{L}(\boldsymbol{\theta}_t)\|^2 + 2\|\boldsymbol{\zeta}_t\|^2 + \|\boldsymbol{\xi}_t\|^2 + 2\langle\nabla\mathcal{L}(\boldsymbol{\theta}_t)+\boldsymbol{\zeta}_t,\boldsymbol{\xi}_t\rangle,
\end{aligned} \tag{45}$$

**Bounding $\langle\boldsymbol{v}_t,\boldsymbol{g}_t\rangle$.** Expanding $\boldsymbol{g}_t = \nabla\mathcal{L}(\boldsymbol{\theta}_t) + \boldsymbol{\zeta}_t + \boldsymbol{\xi}_t$, we have:

$$\begin{aligned}
\langle\boldsymbol{v}_t,\boldsymbol{g}_t\rangle &= \langle\boldsymbol{v}_t,\nabla\mathcal{L}(\boldsymbol{\theta}_t)+\boldsymbol{\zeta}_t+\boldsymbol{\xi}_t\rangle \\
&= \langle\boldsymbol{v}_t,\nabla\mathcal{L}(\boldsymbol{\theta}_t)\rangle + \langle\boldsymbol{v}_t,\boldsymbol{\zeta}_t\rangle + \langle\boldsymbol{v}_t,\boldsymbol{\xi}_t\rangle \\
&\leq \varepsilon_1\|\boldsymbol{v}_t\|^2 + \frac{1}{4\varepsilon_1}\|\nabla\mathcal{L}(\boldsymbol{\theta}_t)\|^2 + \varepsilon_3\|\boldsymbol{v}_t\|^2 + \frac{1}{4\varepsilon_3}\|\boldsymbol{\zeta}_t\|^2 + \langle\boldsymbol{v}_t,\boldsymbol{\xi}_t\rangle
\end{aligned} \tag{46}$$

**Combining Terms.** Taking conditional expectation from Eq. (42) and substituting the previous bounds, we have

$$\begin{aligned}
\mathbb{E}[\Psi_{t+1}|\mathcal{F}_t] - \Psi_t &\leq \eta\beta\varepsilon_1\|\boldsymbol{v}_t\|^2 + \frac{\eta\beta}{4\varepsilon_1}\|\nabla\mathcal{L}(\boldsymbol{\theta}_t)\|^2 \\
&\quad - \eta(1-\beta)\|\nabla\mathcal{L}(\boldsymbol{\theta}_t)\|^2 + \eta(1-\beta)\varepsilon_2\|\nabla\mathcal{L}(\boldsymbol{\theta}_t)\|^2 + \frac{\eta(1-\beta)}{4\varepsilon_2}\|\boldsymbol{\zeta}_t\|^2 \\
&\quad + \left(\frac{\eta^2 L}{2}+c\right)\left(\beta^2\|\boldsymbol{v}_t\|^2 + 2(1-\beta)^2\|\nabla\mathcal{L}(\boldsymbol{\theta}_t)\|^2 + 2(1-\beta)^2\|\boldsymbol{\zeta}_t\|^2 + (1-\beta)^2\sigma^2\right) \\
&\quad 2\left(\frac{\eta^2 L}{2}+c\right)\beta(1-\beta)\left((\varepsilon_1+\varepsilon_3)\|\boldsymbol{v}_t\|^2 + \frac{1}{4\varepsilon_1}\|\nabla\mathcal{L}(\boldsymbol{\theta}_t)\|^2 + \frac{1}{4\varepsilon_3}\|\boldsymbol{\zeta}_t\|^2\right) - c\|\boldsymbol{v}_t\|^2
\end{aligned} \tag{47}$$

**Collecting Terms and Setting Bounds.** After substituting all bounds and collecting terms, we have:

$$\mathbb{E}[\Psi_{t+1}|\mathcal{F}_t] - \Psi_t \leq$$

$$- \eta(1-\beta)\left(1 - \varepsilon_2 - \frac{\eta\beta}{4\varepsilon_1\eta(1-\beta)} - \frac{2}{\eta}\left(\frac{\eta^2 L}{2} + c\right)\left(1 - \beta + \frac{\beta}{4\varepsilon_1}\right)\right)\|\nabla\mathcal{L}(\boldsymbol{\theta}_t)\|^2$$

$$+ \left(\eta\beta\varepsilon_1 + \left(\frac{\eta^2 L}{2} + c\right)\beta\left(\beta + 2(1-\beta)(\varepsilon_1 + \varepsilon_3)\right) - c\right)\|\boldsymbol{v}_t\|^2$$

$$+ \left(\frac{\eta(1-\beta)}{4\varepsilon_2} + 2\left(\frac{\eta^2 L}{2} + c\right)(1-\beta)\left(1 - \beta - \frac{\beta}{4\varepsilon_3}\right)\right)\|\boldsymbol{\zeta}_t\|^2$$

$$+ \left(\frac{\eta^2 L}{2} + c\right)(1-\beta)^2\sigma^2$$

$$(48)$$

Define the following constants:

$$\alpha = \eta(1-\beta)\left(1 - \varepsilon_2 - \frac{\beta}{4\varepsilon_1(1-\beta)} - \frac{2}{\eta}\left(\frac{\eta^2 L}{2} + c\right)\left(1 - \beta + \frac{\beta}{4\varepsilon_1}\right)\right)$$

$$C_1 = \left(\eta\beta\varepsilon_1 + \left(\frac{\eta^2 L}{2} + c\right)\beta\left(\beta + 2(1-\beta)(\varepsilon_1 + \varepsilon_3)\right) - c\right)$$

$$C_2 = \left(\frac{\eta(1-\beta)}{4\varepsilon_2} + 2\left(\frac{\eta^2 L}{2} + c\right)(1-\beta)\left(1 - \beta - \frac{\beta}{4\varepsilon_3}\right)\right)$$

$$D = \left(\frac{\eta^2 L}{2} + c\right)(1-\beta)^2$$

$$(49)$$

**Establishing Convergence.** For convergence, we can set the variables such that $\alpha > 0$ and $C_1 < 0$. The one-step progress in expectation becomes:

$$\mathbb{E}[\Psi_{t+1}|\mathcal{F}_t] \leq \Psi_t - \alpha\|\nabla\mathcal{L}(\boldsymbol{\theta}_t)\|^2 + C_1\|\boldsymbol{v}_t\|^2 + C_2\|\boldsymbol{\zeta}_t\|^2 + D\sigma^2 \qquad (50)$$

When $C_1 < 0$, the term with $\|\boldsymbol{v}_t\|^2$ helps convergence. Taking the full expectation and summing from $t = 0$ to $T - 1$:

$$\sum_{t=0}^{T-1}\mathbb{E}[\alpha\|\nabla\mathcal{L}(\boldsymbol{\theta}_t)\|^2] \leq \mathbb{E}[\Psi_0] - \mathbb{E}[\Psi_T] + \sum_{t=0}^{T-1}(C_2\|\boldsymbol{\zeta}_t\|^2 + D\sigma^2])$$

$$\leq \mathbb{E}[\Psi_0] - \mathcal{L}^* + T(C_2\zeta^2 + D\sigma^2)$$

$$(51)$$

Where we used $\mathcal{L}^* \leq \mathcal{L}(\boldsymbol{\theta}_t)$ and dropped the negative term with $C_1 < 0$. Substituting $\Psi_0 = \mathcal{L}_0 + c\|v_0\|^2 = \mathcal{L}_0$ and dividing by $\alpha T$ we have:

$$\frac{1}{T}\sum_{t=0}^{T-1}\mathbb{E}[\|\nabla\mathcal{L}(\boldsymbol{\theta}_t)\|^2] \leq \frac{\mathcal{L}_0 - \mathcal{L}^*}{\alpha T} + \frac{C_2\zeta^2 + D\sigma^2}{\alpha}. \qquad (52)$$

Thus, the average squared gradient norm converges to a neighborhood determined by the perturbation magnitude $\zeta^2$ and noise variance $\sigma^2$. This result shows that momentum SGD with perturbation and noise converges to a neighborhood of a stationary point in the non-convex smooth case. $\qquad\square$

Theorem 2 provides convergence guarantees for momentum-SGD in the non-convex setting, which is particularly relevant for deep learning applications like LLMs. Instead of convergence to a neighborhood of the optimum, we provide guarantees on the average gradient norm, a standard measure for non-convex optimization. The bound depends directly on the perturbation magnitude, establishing that even in non-convex settings, controlling malicious perturbations through effective detection mechanisms is crucial for ensuring convergence to stationary points.

### E.1.4 Unified Analysis: Detection-Convergence Relationship

We now unify our results to establish a comprehensive relationship between detection thresholds and convergence guarantees. This unified perspective provides us with clear guidance on the security-performance tradeoff.

**Theorem 3** (Convergence Guarantees for Distributed Training with Malicious Workers). *Let a distributed training system with $p$ stages, each having $d$ worker replicas, where a fraction $\gamma_s < \frac{1}{2}$ of workers at stage $s$ are malicious. Using the momentum-based verification with a test statistic detector with threshold $\tau$, let us assume we will use balanced momentum SGD with parameter $\beta \in [0, 1)$ and learning rate $\eta > 0$ for optimization.*
*Assume:*

1. *The test statistic function $\Omega$ is Lipschitz continuous with constant $L_\Omega$,*
2. *Each stage $s$ implements a map $\boldsymbol{h}^{(s,r)} = f_s(\boldsymbol{h}^{(s-1,r)}; \boldsymbol{\theta}^{(s)})$ with Jacobian bounds $\|\partial_{\boldsymbol{\theta}} f_s\| \leq L_{\boldsymbol{\theta}}^{(s)}$ and $\|\partial_{\boldsymbol{h}} f_s\| \leq L_f^{(s)}$,*
3. *The loss function $\mathcal{L}$ is $L$-smooth and bounded below by $\mathcal{L}^*$,*
4. *The stochastic gradient includes zero-mean noise with variance bounded by $\sigma^2$.*

*Then the following results hold:*

- **Detection Evasion Bound.** *For a malicious worker to remain undetected by the test statistic detector:*
$$\varepsilon \leq \frac{\tau}{L_\Omega(1 + \gamma_s)}. \tag{53}$$

- **Parameter Gradient Perturbation.** *The maximum parameter gradient perturbation that can be induced by undetected malicous workers at stage $s$ is:*
$$\|\Delta \nabla_{\boldsymbol{\theta}^{(s)}}^{agg}\| \leq \gamma_s \cdot G_s \cdot \frac{\tau}{L_\Omega(1 + \gamma_s)} \tag{54}$$
*where $G_s = L_{\boldsymbol{\theta}}^{(s)} \left( \prod_{j \geq s+1} L_f^{(j)} \right)$ represents the amplification factor for perturbations.*

- **Convergence Bounds.** *Under the maximum undetected perturbation and assuming $\zeta := \gamma_s \cdot G_s \cdot \frac{\tau}{L_\Omega(1+\gamma_s)}$, for non-convex loss we have:*
$$\boxed{\frac{1}{T} \sum_{t=0}^{T-1} \mathbb{E}[\|\nabla \mathcal{L}(\boldsymbol{\theta}_t)\|^2] \leq \frac{\mathcal{L}_0 - \mathcal{L}^*}{\alpha T} + \frac{C_2 \zeta^2 + D\sigma^2}{\alpha}} \tag{55}$$

*where constants are as defined in Theorem 2.*

*Proof.* We prove the theorem by connecting the results from Lemmas 2 and 5 and Theorem 2. $\square$

### E.1.5 Recovering Well-known Lower-bounds for SGD Convergence from Theorem 3

To evaluate our convergence theorem's validity, we examine whether it generalizes to common non-convex optimization bounds. Consider vanilla SGD without malicious perturbations: setting $\beta = 0$ (relaxing momentum SGD to SGD), $\zeta = 0$ (no malicious perturbation), and substituting into our coefficients from Eq. (38) with $c \to 0^+$:
$$\begin{aligned} \alpha &\approx \mathcal{O}(\eta) \\ D &\approx \mathcal{O}(\eta^2 L), \end{aligned} \tag{56}$$
where $\eta$ is our learning rate (see Eq. (36)).

Substituting these coefficients into our convergence theorem, we have:

$$\frac{1}{T}\sum_{t=0}^{T-1}\mathbb{E}[\|\nabla\mathcal{L}(\theta_t)\|^2] \leq \frac{\mathcal{L}_0 - \mathcal{L}^*}{\alpha T} + \frac{C_2\zeta^2 + D\sigma^2}{\alpha} \tag{57}$$

$$\overset{(a)}{\leq} \frac{\mathcal{L}_0 - \mathcal{L}^*}{\alpha T} + \frac{D\sigma^2}{\alpha} \tag{58}$$

$$\overset{(b)}{\leq} \mathcal{O}\left(\frac{\mathcal{L}_0 - \mathcal{L}^*}{\eta T} + \eta L\sigma^2\right) \tag{59}$$

where (a) assumes no perturbation, and (b) uses the derived constants.

This matches the classical SGD bound from Koloskova et al. (2024):

> **"SGD, Ex. 3.1** Since $\sigma^2 \leq \tau\sigma_{\text{SGD}}^2$ (see Table 2), and using that $\tau = \Theta(1/L\gamma)$ the convergence rate in Theorem 5.1 converts to
>
> $$\frac{1}{T}\sum_{t=0}^{T}\mathbb{E}\|\nabla f(x_t)\|^2 \leq \mathcal{O}\left(\frac{F_0}{\gamma T} + L\gamma\sigma_{\text{SGD}}^2\right),$$
>
> with $\gamma \leq \frac{1}{8\sqrt{3}L}$ [and where $F_0 = f(\mathbf{x}_0) - f^*$]. This recovers classical convergence rate of SGD for non-convex functions (up to constants)."

This bound exactly matches what we derive from our convergence theorem.

### E.1.6 THEORY IMPLICATIONS

Our unified analysis reveals several important implications for designing malicious-tolerant SWARM verification:

- **Security-Convergence Tradeoff**: The detection threshold $\tau$ directly impacts the convergence guarantees through its effect on the maximum undetected perturbation. Lower thresholds provide stronger security guarantees but may increase false positives and potentially slow convergence due to unnecessary worker exclusion.
- **Byzantine Fraction Impact**: As shown in Eq. (54), the maximum parameter deviation is proportional to $\frac{\gamma_s}{1+\gamma_s}$, where $\gamma_s$ represents the fraction of malicious workers at stage $s$. This monotonically increasing function implies that larger malicious fractions allow more severe parameter deviations, highlighting the importance of maintaining an honest majority.
- **Detector Sensitivity**: A detector $\Omega$ with larger Lipschitz constant $L_\Omega$ reduces the parameter gradient deviation. In practical terms, employing a more sensitive detection function constrains the potential impact of malicious workers by allowing them less room for undetected perturbation.
- **Stage Vulnerability**: Stages with higher values of the amplification factor $G_s$ are more vulnerable to malicious perturbations. In typical neural network architectures, this often means that earlier layers (which affect all subsequent computation) have greater vulnerability. This suggests that security resources should be prioritized to monitor these critical stages more closely.
- **Momentum as Robustness against Malicious Behavior**: Higher momentum values $\beta$ naturally reduce the impact of per-iteration perturbations by placing more weight on the historical gradient estimates. This provides an inherent form of robustness that complements explicit detection mechanisms. The optimal momentum value therefore depends not only on optimization dynamics but also on security considerations.
- **Adaptive Detection Thresholds**: Our analysis suggests that detection thresholds could be optimally set differently for each stage based on their amplification factors $G_s$. Stages with higher amplification factors should use stricter thresholds to maintain consistent convergence guarantees across the model. We leave this for future work.

### E.2 PROOF OF HONEST MAJORITY GUARANTEE

This section provides the complete proof of Lemma 1 from Sec. 3.2.

**Lemma 1** (Honest Majority Guarantee). *Consider our distributed training system with $p$ pipeline stages, each replicated across $d$ worker nodes. Let $b$ be the total number of malicious workers, and $\epsilon \in (0,1)$ be a small positive constant. If workers are assigned to each stage randomly and*

$$b \le \frac{dp}{2} - p\sqrt{\frac{d}{2} \ln\left(\frac{p}{\epsilon}\right)}, \tag{60}$$

*then with probability at least $1 - \epsilon$ every pipeline stage has strictly fewer than $d/2$ malicious workers.*

In other words, Lemma 1 makes a connection between the initial pool of malicious workers and the conditions under which a "random worker assignment" to each stage would preserve the per-stage honest majority assumption needed by SENTINEL. This is important as usually we do not know apriori if a worker is Byzantine or not, and since usually a random worker assignment is used to allocate workers, we need to ensure that the honest majority assumption is preserved.

*Proof.* Let $B_s$ denote the set of malicious worker nodes at stage $s$. We model the assignment of malicious nodes as follows: each worker node has probability $q = b/n = b/(dp)$ of being malicious, independently of other nodes.

For any stage $s$, when stages are randomly assigned to workers, the number of malicious nodes $|B_s|$ follows a binomial distribution with parameters $d$ and $q$:

$$|B_s| \sim \text{Binomial}(d, q), \quad \mathbb{E}[|B_s|] = qd. \tag{61}$$

Our goal is to ensure that, with high probability, every stage $s$ has $|B_s| < d/2$. Using Hoeffding's inequality (Hoeffding, 1994) for sums of independent Bernoulli random variables, we have

$$\Pr\left[|B_s| - \mathbb{E}[|B_s|] \ge t\right] \le \exp\left(-\frac{2t^2}{d}\right). \tag{62}$$

Setting $t = d/2 - qd = d(1/2 - q)$, we obtain

$$\Pr\left[|B_s| \ge d/2\right] \le \exp\left(-2d\left(\frac{1}{2} - q\right)^2\right). \tag{63}$$

Applying the union bound across all $p$ stages, the probability that at least one stage has a majority of malicious workers is bounded by

$$\Pr\left[\exists s : |B_s| \ge d/2\right] \le p \cdot \exp\left(-2d\left(\frac{1}{2} - q\right)^2\right). \tag{64}$$

For this probability to be at most $\epsilon$, we require

$$p \cdot \exp\left(-2d\left(\frac{1}{2} - q\right)^2\right) \le \epsilon. \tag{65}$$

Taking the natural logarithm of both sides and solving for $q$, we get

$$\frac{1}{2} - q \ge \sqrt{\frac{\ln(p/\epsilon)}{2d}}. \tag{66}$$

Substituting $q = b/n = b/(dp)$ and solving for $b$, we obtain the maximum allowable number of malicious nodes:

$$b_{\max} = \left(\frac{1}{2} - \sqrt{\frac{\ln(p/\epsilon)}{2d}}\right) dp = \frac{dp}{2} - p\sqrt{\frac{d}{2}\ln\left(\frac{p}{\epsilon}\right)}. \tag{67}$$

Therefore, if the total number of malicious nodes $b$ is at most $b_{\max}$, then with probability at least $1 - \epsilon$, all pipeline stages will have strictly fewer than $d/2$ malicious worker nodes. $\square$

## F    EXTENDED EXPERIMENTAL RESULTS

This section provides comprehensive experimental details supporting our main findings. We first describe the experimental setup and hyper-parameters, followed by an extended version of our results.

### F.1    DETAILED EXPERIMENTAL SETTINGS

**Experimental Infrastructure.**    To simulate a heterogeneous distributed environment, we developed our experiments using the `TorchTitan` (Liang et al., 2025) framework built on `PyTorch` (Paszke et al., 2019).[4] Our setup employed pipeline parallelism where each transformer layer corresponds to a pipeline stage, with data parallel replicas serving as workers within each stage. We conducted experiments across three scales using 1-3 compute nodes, each equipped with 8 `NVIDIA A100-SXM4-40GB` GPUs, resulting in configurations with 64, 128, and 256 total workers. Ablation studies were performed using 8 `NVIDIA A100-SXM4-80GB` GPUs. For our large scale experiment with Llama-3-4B, we used 8 `NVIDIA H100-HBM3-80GB` GPUs with activation checkpointing.

**Model and Training Configuration.**    We evaluated decoder-only Llama-3 (Dubey et al., 2024) models with varying architectural configurations. Complete training hyper-parameters are provided in Tab. 10. Additionally, we also evaluate the performance of our approach on NanoGPT (Karpathy, 2022) as a representative GPT2 (Radford et al., 2019) architecture, and Llama-4-0.4B (MetaAI, 2025) and DeepSeek-V3-1B (DeepSeek-AI, 2024) as representative Mixture-of-Expert models. Unless stated otherwise, we train all model configurations for 5000 steps.

**Verification Settings.**    SENTINEL deploys dedicated verifier nodes that intercept inter-stage communications and monitor both forward activations and backward activation gradients for anomalous behavior. We permanently ban workers after $c = 5$ violations. Moreover, we implement an adaptive IQR thresholding mechanism to detect outliers, with parameters detailed in Tab. 11.[5] These detection thresholds were calibrated empirically through preliminary experiments on each configuration, though additional hyper-parameter tuning may yield further improvements.

As pointed out in the paper, we assume that the first layer and final two layers of each model are operated by honest nodes. This assumption is cruicial for two reasons:

1. these layers control the primary data and gradient flow during forward and backward propagation, making them essential for training stability, and
2. they represent key attack surfaces for adversaries seeking to compromise model integrity through data poisoning/backdoor attacks (Li et al., 2024) (via the input layer) or label manipulation (Biggio et al., 2012; Fung et al., 2020) (via the output layers).

Without this honest-node assumption, malicious actors could easily circumvent intermediate verification by corrupting inputs or outputs directly.

**Datasets.**    We conduct experiments on three large-scale text corpora: CommonCrawl (C4) (Raffel et al., 2020), FineWeb (FW) (Penedo et al., 2024), and OpenWebText (OW) (Gokaslan et al., 2019), all obtained through Hugging Face Datasets Streaming API (Lhoest et al., 2021). To enable model evaluation, we construct a held-out validation set comprising 100k samples that remains separate from training data throughout the process. During validation phases, we sample batches of 160 samples per data-parallel worker from this validation set for evaluation.

---

[4]We integrate the NanoGPT following the implementation of Karpathy (2022) into `TorchTitan`.
[5]For our experiments on MoE-based models, we use the same settings as the Llama-3-0.6B model.

Table 10: Model architectures and training configuration.

| Model Configuration | NanoGPT-0.25B | Llama-3-0.6B | Llama-3-1.2B | Llama-3-4B | Llama-4-0.4B | DeepSeek-V3-1B |
|---|---|---|---|---|---|---|
| **Model Architecture** | | | | | | |
| Parameters | 278,364,672 | 574,391,296 | 1,224,247,296 | 3,984,989,184 | 443,957,760 | 967,989,760 |
| Hidden Dimensions | 768 | 1024 | 2048 | 3072 | 512 | 1024 |
| Number of Layers (Stages) | 12 | 16 | 8 | 22 | 16 | 15 |
| Attention Heads | 12 | 32 | 32 | 32 | 32 | 16 |
| Key-Value Heads | – | 8 | 8 | 8 | 8 | – |
| Number of Experts | – | – | – | – | 8 | 8 |
| Number of Shared Experts | – | – | – | – | 1 | 2 |
| Top-K (for MoE) | – | – | – | – | 1 | 6 |
| RoPE Theta | – | 500000 | 500000 | 500000 | 500000 | |
| FFN Dimension Multiplier | – | – | 1.3 | 1.3 | 1.3 | |
| Multiple Of | – | – | 1024 | 1024 | 1024 | |
| RoPE Theta | – | 500000 | 500000 | 500000 | 500000 | |
| **Distributed Setup** | | | | | | |
| Data Parallel Dimension | 8 | 8 (16 for 16×16 mesh) | 8 | 8 | 8 | 8 |
| Pipeline Parallel Dimension | 12 | 16 | 8 | 22 | 16 | 15 |
| Total Workers | 96 | 128 (256 for 16×16 mesh) | 64 | 176 | 128 | 90 |
| **Optimizer** | | | | | | |
| Type | AdamW | AdamW | AdamW | AdamW | AdamW | AdamW |
| Learning Rate | 6e-4 | 6e-4 | 6e-4 | 6e-4 | 4e-3 | 2.2e-4 |
| Epsilon | 1e-8 | 1e-8 | 1e-8 | 1e-8 | 1e-15 | 1e-8 |
| **Learning Rate Scheduler** | | | | | | |
| Warmup Steps | 100 | 100 | 100 | 500 | 600 | 600 |
| Decay Ratio | 0.8 | 0.8 | 0.8 | 0.8 | – | 0.8 |
| Decay Type | Linear | Linear | Linear | Linear | Linear | Cosine |
| Minimum LR | 0.0 | 0.0 | 0.0 | 0.0 | 0.1 | 0.1 |
| **Training** | | | | | | |
| Worker Batch Size | 8 | 12 | 10 | 12 | 16 | 16 |
| Global Batch Size | 64 | 96 (192 for 16×16 mesh) | 80 | 96 | 128 | 128 |
| Sequence Length | 1024 | 1024 | 1024 | 1024 | 1024 | 1024 |
| Gradient Clipping | 1.0 | 1.0 | 1.0 | 1.0 | 1.0 | 1.0 |

Table 11: Hyper-parameters of the proposed verification approach (Alg. 1).

| PARAMETER | NANOGPT | | | | LLAMA-3 | | | |
|---|---|---|---|---|---|---|---|---|
| | 8×12 MESH, 0.25B | | 8×16 MESH, 0.6B | | 16×16 MESH, 0.6B | | 8×8 MESH, 1.2B | |
| | ACTIVATION | GRADIENT | ACTIVATION | GRADIENT | ACTIVATION | GRADIENT | ACTIVATION | GRADIENT |
| Momentum Beta ($\beta$) | 0.90 | 0.80 | 0.90 | 0.80 | 0.90 | 0.80 | 0.99 | 0.85 |
| Initial IQR Multiplier ($k_0$) | 1.5 | 3.0 | 1.5 | 3.0 | 1.5 | 3.0 | 3.0 | 3.0 |
| Target FP-Rate ($\alpha$) | 0.01 | 0.001 | 0.0001 | 0.001 | 0.01 | 0.01 | 0.001 | 0.05 |
| Adaptive Max Iterations ($N_{max}$) | 10 | 10 | 10 | 10 | 10 | 10 | 10 | 10 |
| Adaptive Grow Factor ($\gamma_g$) | 1.1 | 1.01 | 1.1 | 1.01 | 1.1 | 1.01 | 1.25 | 1.01 |
| Adaptive Shrink Factor ($\gamma_s$) | 0.9 | 0.99 | 0.9 | 0.99 | 0.9 | 0.99 | 0.95 | 0.99 |
| Adaptive Min Distance Multiplier ($\Lambda$) | 0.35 | 0.15 | 0.15 | 0.05 | 0.25 | 0.05 | 0.2 | 0.05 |
| Adaptive Epsilon ($\varepsilon$) | 0.01 | 0.001 | 0.001 | 0.00005 | 0.0001 | 0.0001 | 0.01 | 0.0001 |

## F.2 DETAILED EXPERIMENTAL RESULTS

**Activation vs. Gradient Attack Analysis in Pipeline Parallelism.** Building on the subset of attacks presented in Sec. 5, we now provide comprehensive verification results across all activation and gradient attacks from Sec. 2 to fully characterize their behavior. Tab. 12 presents complete results on the C4 dataset, while Tabs. 13 and 14 demonstrate performance on FineWeb and OpenWebText datasets, respectively.

Our comprehensive evaluation reveals several important insights. First, activation manipulation poses an equally significant threat as gradient manipulation in distributed, pipeline parallel-based training, confirming that both attack vectors require careful consideration in Byzantine-tolerant systems. Second, when attacks evade detection, their deviation from baseline vanilla training remains negligible, directly supporting our theoretical analysis presented in Theorem 1. This consistency holds across all three datasets, demonstrating the robustness and versatility of our approach across diverse training scenarios.

Particularly noteworthy are attacks detected with a detection speed of 1.0, indicating that despite our forgiveness strategy introduced in Sec. 3.1, these attacks produce sufficiently substantial deviations to warrant immediate exclusion of the malicious worker. The training and validation loss curves in Figs. 7 to 10 further illustrate how our EMA verification approach effectively controls malicious behavior, maintaining performance close to vanilla baselines throughout training. These results collectively validate the effectiveness and generalizability of our proposed verification framework.

Table 12: Attack detection performance for Llama-3-0.6B on C4 dataset. Metrics shown include precision, recall, F1 score (all as percentages), average detection speed (in iterations), and validation loss.

| MODE | ATTACK | SENTINEL (OURS) | | | | | NO VERIF. |
|---|---|---|---|---|---|---|---|
| | | PR. (%) ↑ | RE. (%) ↑ | F1 (%) ↑ | DET. SPEED ↓ | VAL. LOSS ↓ | VAL. LOSS ↓ |
| - | None (Vanilla) | 100.0 | 100.0 | 100.0 | N/A | 3.819 | 3.821 |
| ACTIVATION MANIPULATION | Constant (Zeros) | 100.0 | 100.0 | 100.0 | 6.5 | 3.809 | 11.761 |
| | Constant (Ones) | 100.0 | 100.0 | 100.0 | 6.33 | 3.817 | 7.778 |
| | Random Value | 100.0 | 100.0 | 100.0 | 6.48 | 3.827 | 7.778 |
| | Scaling ($\alpha = -1$) | 100.0 | 100.0 | 100.0 | 6.38 | 3.824 | 4.109 |
| | Random Sign (1%) | 100.0 | 100.0 | 100.0 | 6.33 | 3.825 | 4.670 |
| | Random Sign (10%) | 100.0 | 100.0 | 100.0 | 6.52 | 3.822 | 4.619 |
| | Random Sign (30%) | 88.9 | 100.0 | 94.1 | 70.91 | 3.841 | 4.567 |
| | Delay (100-steps) | 88.9 | 100.0 | 94.1 | 13.21 | 3.841 | 7.675 |
| | Bias Addition | 84.6 | 91.7 | 88.0 | 14.57 | 3.830 | 3.892 |
| | Invisible Noise (90%) | 100.0 | 100.0 | 100.0 | 6.48 | 3.836 | 7.675 |
| | Invisible Noise (95%) | 100.0 | 100.0 | 100.0 | 6.52 | 3.823 | 7.677 |
| | Invisible Noise (99%) | 100.0 | 100.0 | 100.0 | 6.48 | 3.826 | 7.682 |
| GRADIENT MANIPULATION | Constant (Zeros) | 100.0 | 100.0 | 100.0 | 6.42 | 3.829 | 3.942 |
| | Constant (Ones) | 88.9 | 100.0 | 94.1 | 1.0 | 3.816 | 10.630 |
| | Random Value | 100.0 | 100.0 | 100.0 | 1.0 | 3.818 | 9.595 |
| | Scaling ($\alpha = -1$) | 0.0 | 0.0 | 0.0 | N/A | 3.893 | 3.893 |
| | Random Sign (1%) | 0.0 | 0.0 | 0.0 | N/A | 3.990 | 3.982 |
| | Random Sign (10%) | 0.0 | 0.0 | 0.0 | N/A | 3.982 | 3.994 |
| | Random Sign (30%) | 0.0 | 0.0 | 0.0 | N/A | 3.944 | 3.933 |
| | Delay (100-steps) | 100.0 | 100.0 | 100.0 | 7.33 | 3.826 | 10.157 |
| | Bias Addition | 100.0 | 100.0 | 100.0 | 1.0 | 3.828 | 10.813 |
| | Invisible Noise (90%) | 100.0 | 75.0 | 85.7 | 101.89 | 3.968 | 4.218 |
| | Invisible Noise (95%) | 100.0 | 79.2 | 88.4 | 229.68 | 3.954 | 4.174 |
| | Invisible Noise (99%) | 100.0 | 79.2 | 88.4 | 211.0 | 3.943 | 4.176 |

Table 13: Attack detection performance for Llama-3-0.6B on FineWeb dataset. Metrics shown include precision, recall, F1 score (all as percentages), average detection speed (in iterations), and validation loss.

| MODE | ATTACK | SENTINEL (OURS) | | | | | NO VERIF. |
|---|---|---|---|---|---|---|---|
| | | PR. (%) ↑ | RE. (%) ↑ | F1 (%) ↑ | DET. SPEED ↓ | VAL. LOSS ↓ | VAL. LOSS ↓ |
| - | None (Vanilla) | 100.0 | 100.0 | 100.0 | N/A | 3.818 | 3.840 |
| ACTIVATION MANIPULATION | Constant (Zeros) | 100.0 | 100.0 | 100.0 | 6.43 | 3.819 | 11.761 |
| | Constant (Ones) | 100.0 | 100.0 | 100.0 | 6.61 | 3.814 | 7.793 |
| | Random Value | 96.0 | 100.0 | 98.0 | 6.46 | 3.831 | 7.793 |
| | Scaling ($\alpha = -1$) | 100.0 | 100.0 | 100.0 | 6.29 | 3.827 | 4.121 |
| | Random Sign (1%) | 100.0 | 100.0 | 100.0 | 6.42 | 3.825 | 4.693 |
| | Random Sign (10%) | 51.1 | 100.0 | 67.6 | 3.58 | 3.829 | 4.716 |
| | Random Sign (30%) | 92.3 | 100.0 | 96.0 | 7.5 | 3.826 | 4.564 |
| | Delay (100-steps) | 85.7 | 100.0 | 92.3 | 14.83 | 3.832 | 7.692 |
| | Bias Addition | 86.4 | 79.2 | 82.6 | 10.68 | 3.828 | 3.898 |
| | Invisible Noise (90%) | 100.0 | 100.0 | 100.0 | 6.38 | 3.824 | 7.709 |
| | Invisible Noise (95%) | 100.0 | 100.0 | 100.0 | 6.29 | 3.824 | 7.713 |
| | Invisible Noise (99%) | 100.0 | 100.0 | 100.0 | 6.33 | 3.829 | 7.712 |
| GRADIENT MANIPULATION | Constant (Zeros) | 100.0 | 100.0 | 100.0 | 6.33 | 3.815 | 3.949 |
| | Constant (Ones) | 92.3 | 100.0 | 96.0 | 1.14 | 3.824 | 10.726 |
| | Random Value | 100.0 | 100.0 | 100.0 | 1.38 | 3.831 | 9.359 |
| | Scaling ($\alpha = -1$) | 0.0 | 0.0 | 0.0 | N/A | 3.888 | 3.901 |
| | Random Sign (1%) | 0.0 | 0.0 | 0.0 | N/A | 3.989 | 3.999 |
| | Random Sign (10%) | 0.0 | 0.0 | 0.0 | N/A | 3.990 | 4.004 |
| | Random Sign (30%) | 0.0 | 0.0 | 0.0 | N/A | 3.941 | 3.966 |
| | Delay (100-steps) | 100.0 | 100.0 | 100.0 | 7.29 | 3.817 | 10.017 |
| | Bias Addition | 100.0 | 100.0 | 100.0 | 1.14 | 3.828 | 10.573 |
| | Invisible Noise (90%) | 100.0 | 75.0 | 85.7 | 52.94 | 3.954 | 4.212 |
| | Invisible Noise (95%) | 100.0 | 75.0 | 85.7 | 140.89 | 3.959 | 4.217 |
| | Invisible Noise (99%) | 100.0 | 75.0 | 85.7 | 209.56 | 3.949 | 4.197 |

Table 14: Attack detection performance for Llama-3-0.6B on OpenWebText dataset. Metrics shown include precision, recall, F1 score (all as percentages), average detection speed (in iterations), and validation loss.

| MODE | ATTACK | SENTINEL (OURS) | | | | | NO VERIF. |
|---|---|---|---|---|---|---|---|
| | | PR. (%) ↑ | RE. (%) ↑ | F1 (%) ↑ | DET. SPEED ↓ | VAL. LOSS ↓ | VAL. LOSS ↓ |
| - | None (Vanilla) | 100.0 | 100.0 | 100.0 | N/A | 3.773 | 3.778 |
| ACTIVATION MANIPULATION | Constant (Zeros) | 100.0 | 100.0 | 100.0 | 6.33 | 3.773 | 11.761 |
| | Constant (Ones) | 100.0 | 100.0 | 100.0 | 6.29 | 3.779 | 7.820 |
| | Random Value | 100.0 | 100.0 | 100.0 | 6.29 | 3.777 | 7.821 |
| | Scaling ($\alpha = -1$) | 100.0 | 100.0 | 100.0 | 6.29 | 3.776 | 4.016 |
| | Random Sign (1%) | 100.0 | 100.0 | 100.0 | 6.21 | 3.774 | 4.614 |
| | Random Sign (10%) | 100.0 | 100.0 | 100.0 | 6.29 | 3.779 | 4.578 |
| | Random Sign (30%) | 96.0 | 100.0 | 98.0 | 25.62 | 3.779 | 4.524 |
| | Delay (100-steps) | 88.9 | 100.0 | 94.1 | 10.74 | 3.790 | 7.701 |
| | Bias Addition | 91.7 | 91.7 | 91.7 | 14.0 | 3.782 | 3.843 |
| | Invisible Noise (90%) | 100.0 | 100.0 | 100.0 | 6.25 | 3.777 | 7.677 |
| | Invisible Noise (95%) | 100.0 | 100.0 | 100.0 | 6.5 | 3.783 | 7.674 |
| | Invisible Noise (99%) | 100.0 | 100.0 | 100.0 | 6.29 | 3.780 | 7.678 |
| GRADIENT MANIPULATION | Constant (Zeros) | 100.0 | 100.0 | 100.0 | 5.7 | 3.775 | 3.908 |
| | Constant (Ones) | 100.0 | 100.0 | 100.0 | 1.0 | 3.774 | 10.722 |
| | Random Value | 100.0 | 100.0 | 100.0 | 1.0 | 3.780 | 9.611 |
| | Scaling ($\alpha = -1$) | 0.0 | 0.0 | 0.0 | N/A | 3.847 | 3.860 |
| | Random Sign (1%) | 12.1 | 16.7 | 14.0 | 1516.33 | 3.990 | 3.957 |
| | Random Sign (10%) | 5.3 | 4.2 | 4.7 | 2646.0 | 3.967 | 3.952 |
| | Random Sign (30%) | 0.0 | 0.0 | 0.0 | N/A | 3.901 | 3.906 |
| | Delay (100-steps) | 100.0 | 100.0 | 100.0 | 7.09 | 3.771 | 11.508 |
| | Bias Addition | 100.0 | 100.0 | 100.0 | 1.0 | 3.777 | 10.947 |
| | Invisible Noise (90%) | 100.0 | 87.5 | 93.3 | 92.95 | 3.877 | 4.178 |
| | Invisible Noise (95%) | 100.0 | 91.7 | 95.7 | 136.42 | 3.828 | 4.098 |
| | Invisible Noise (99%) | 100.0 | 87.5 | 93.3 | 64.42 | 3.858 | 4.123 |

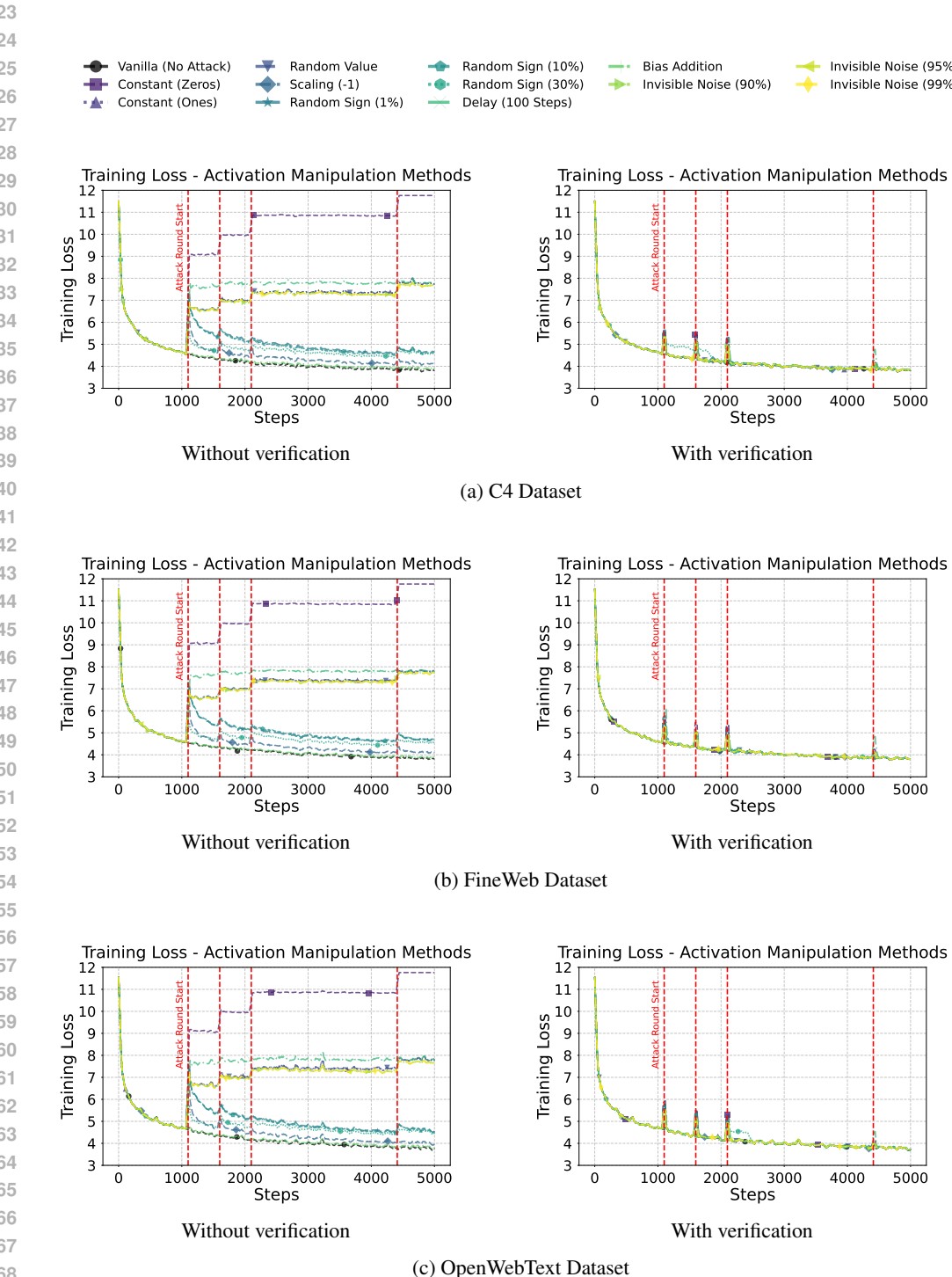

Figure 7: **Training loss** comparing our verification mechanism against baseline vanilla training under **activation manipulation attacks**. We evaluate on Llama-3-0.6B using three datasets (C4, FineWeb, and OpenWebText). Dotted vertical lines indicate attack initiation points where 6 randomly selected nodes begin submitting adversarial activations in coordinated Byzantine attacks. Our verification approach maintains stable convergence while the baseline suffers significant degradation under attack.

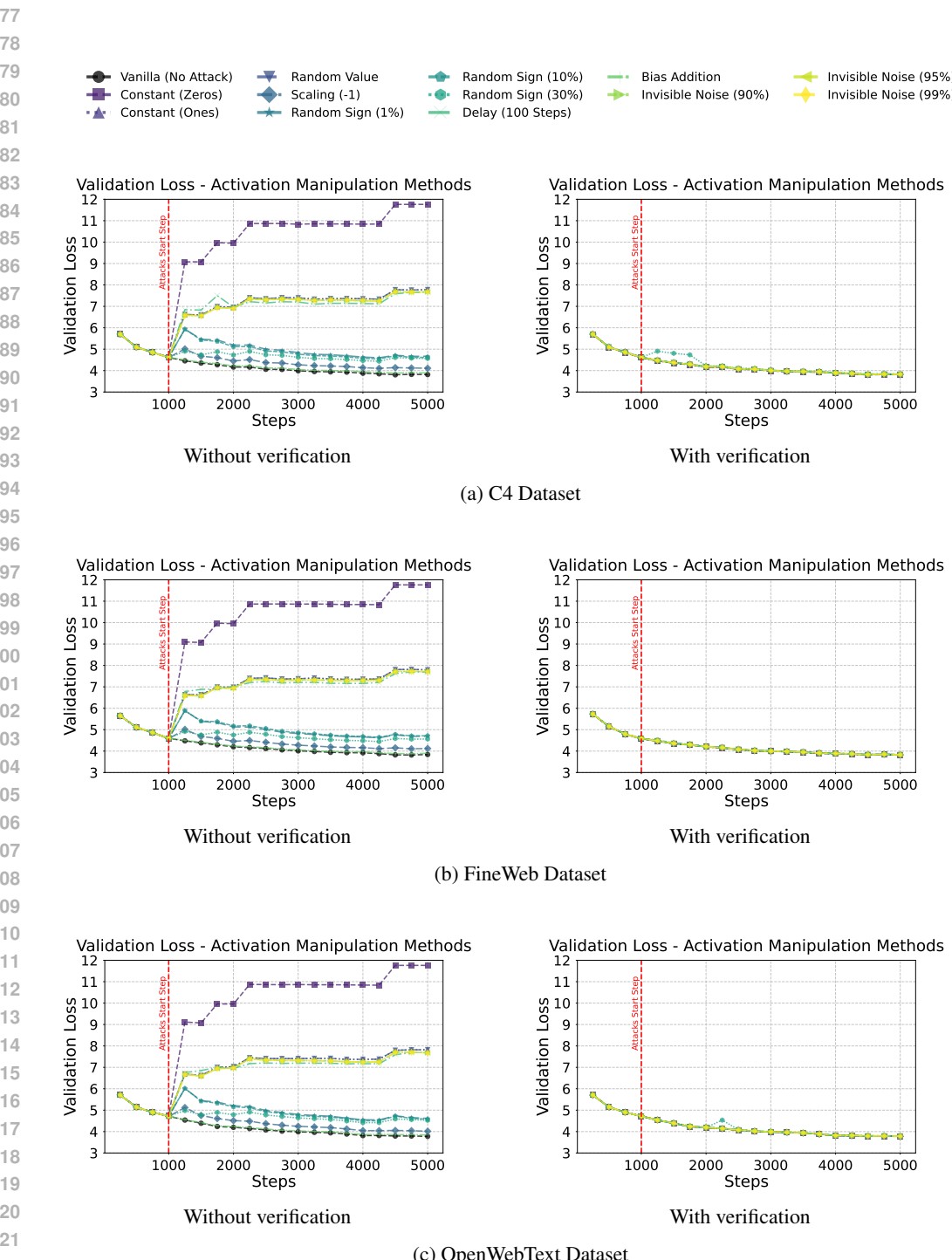

Figure 8: **Validation loss** comparing our verification mechanism against baseline vanilla training under **activation manipulation attacks**. We evaluate Llama-3-0.6B across three datasets (C4, FineWeb, and OpenWebText). The dotted red line marks the transition from warm-up to the attack phase, where Byzantine nodes begin submitting adversarial activations. Our verification approach maintains stable validation performance while without it, the baseline shows significant degradation post-attack.

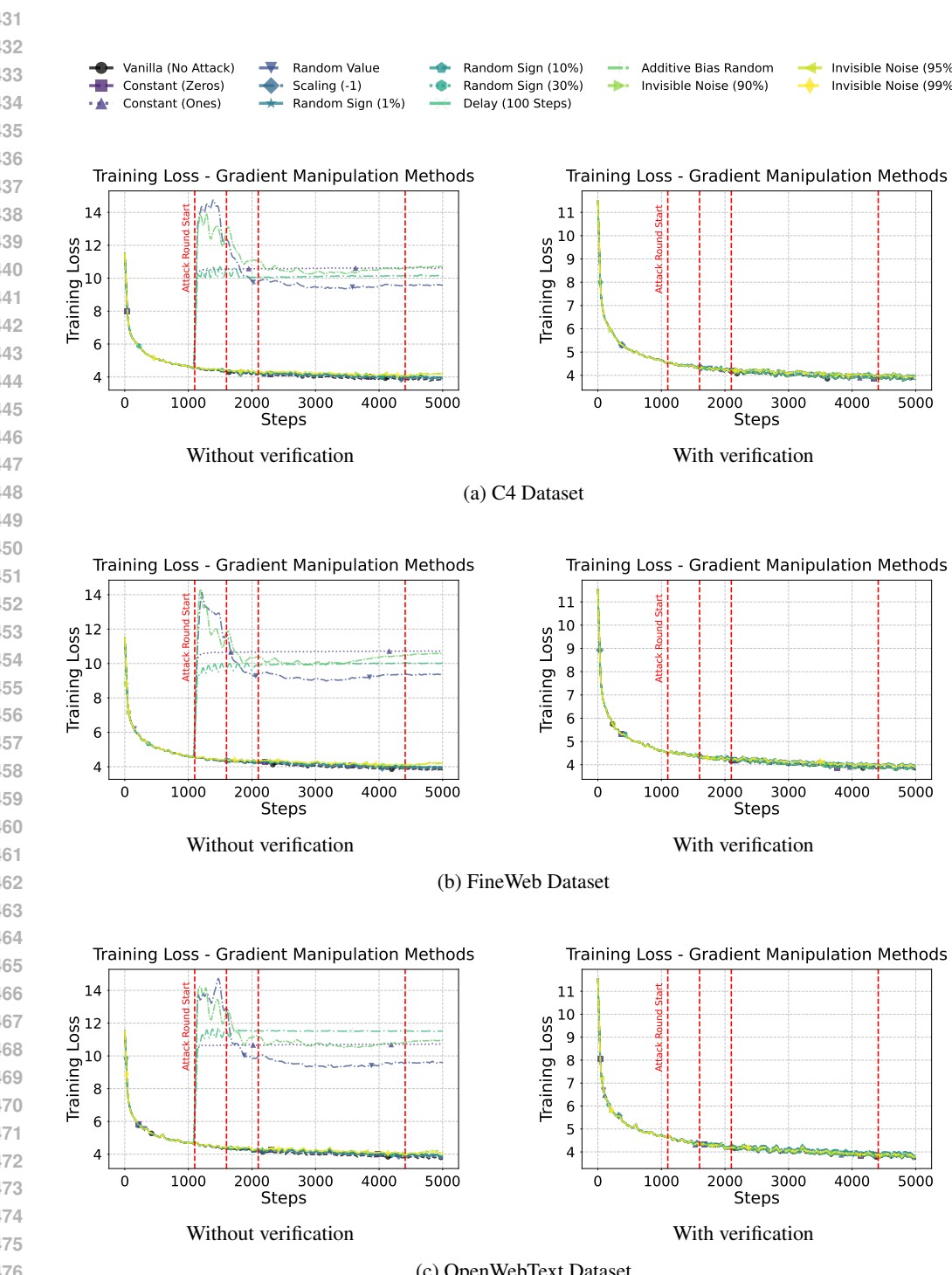

Figure 9: **Training loss** comparing our verification mechanism against baseline vanilla training under **gradient manipulation attacks**. We evaluate on Llama-3-0.6B using three datasets (C4, FineWeb, and OpenWebText). Dotted vertical lines indicate attack initiation points where 6 randomly selected nodes begin submitting adversarial gradients in coordinated Byzantine attacks. Our verification approach maintains stable convergence while the baseline suffers significant degradation under attack.

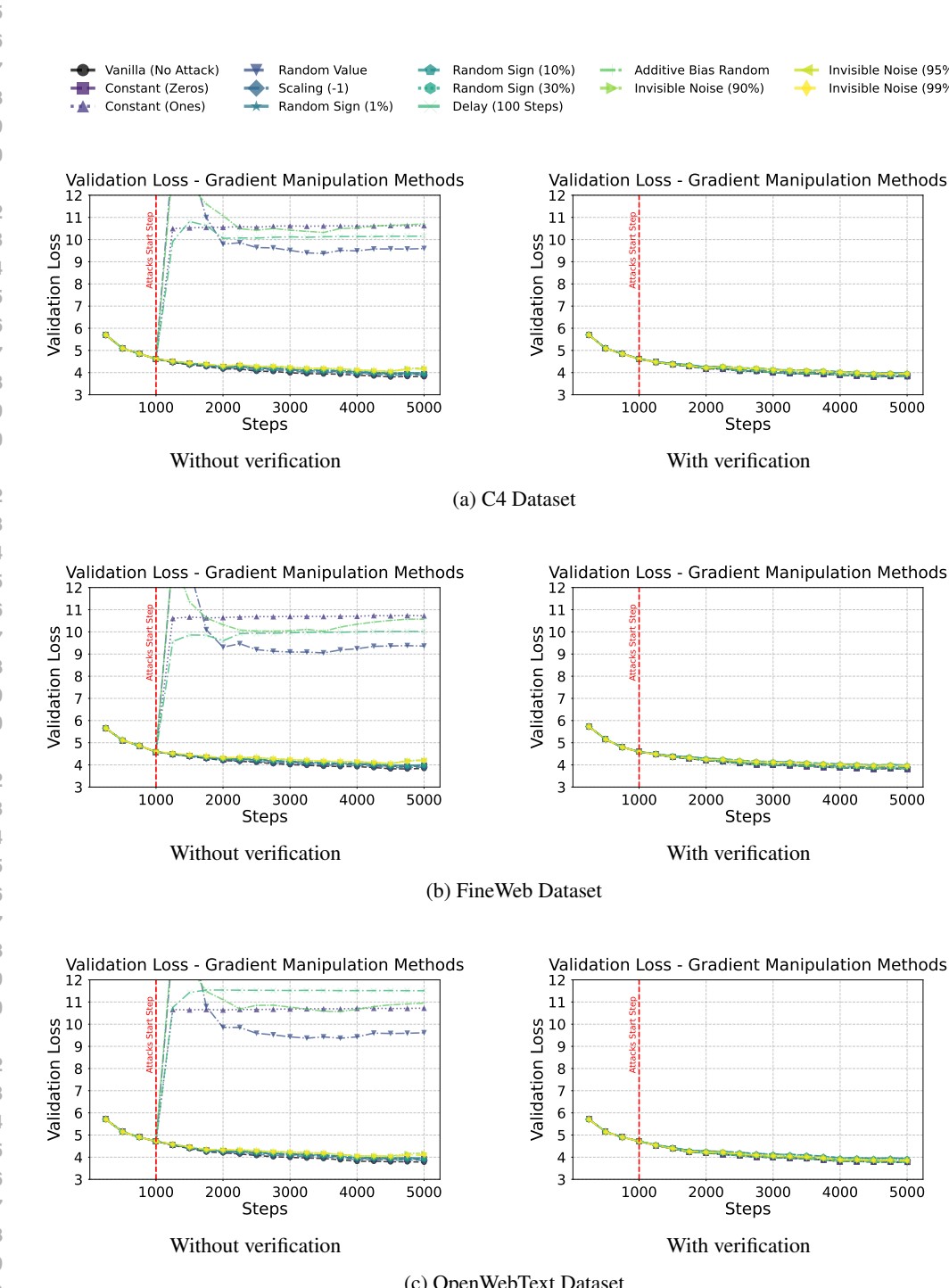

(a) C4 Dataset

(b) FineWeb Dataset

(c) OpenWebText Dataset

Figure 10: **Validation loss** comparing our verification mechanism against baseline vanilla training under **gradient manipulation attacks**. We evaluate Llama-3-0.6B across three datasets (C4, FineWeb, and OpenWebText). The dotted red line marks the transition from warm-up to the attack phase, where Byzantine nodes begin submitting adversarial gradients. Our verification approach maintains stable validation performance while without it, the baseline shows significant degradation post-attack.

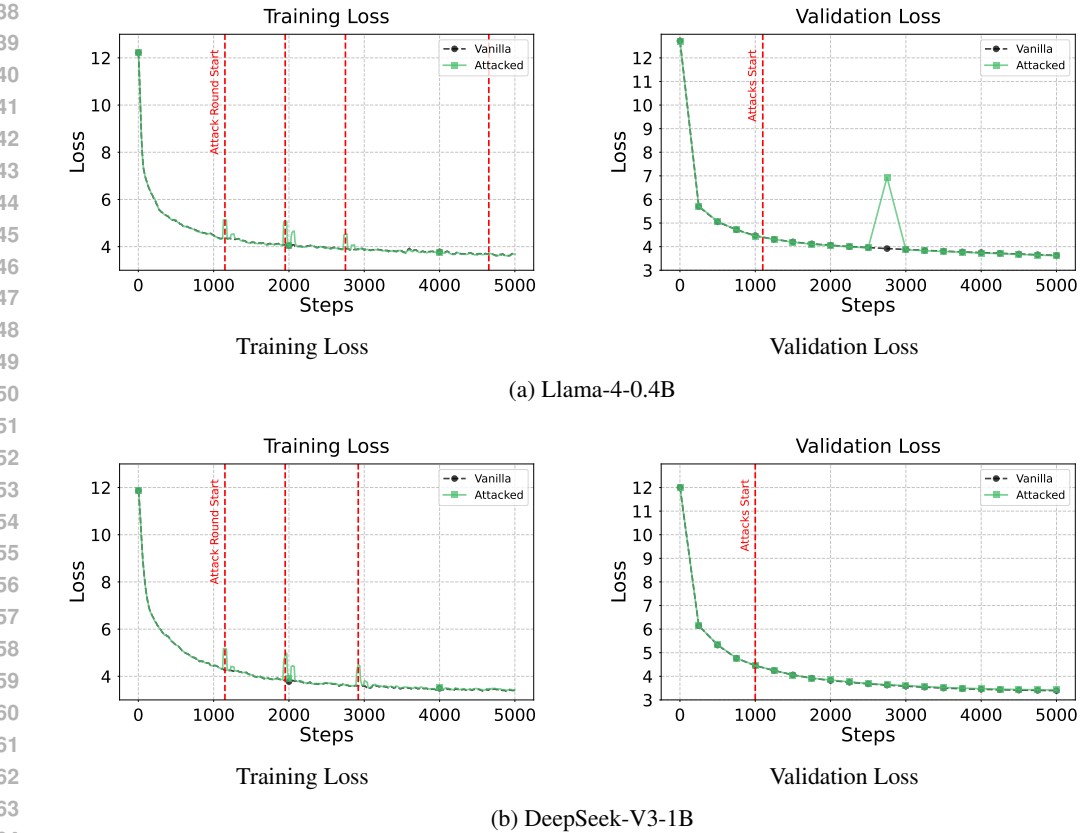

Training Loss

Validation Loss

(a) Llama-4-0.4B

Training Loss

Validation Loss

(b) DeepSeek-V3-1B

Figure 11: Training and validation loss evolution for MoE-based models trained on FineWeb-EDU dataset over 5k iterations. The mixed activation attack scenario assumes 37.5% Byzantine nodes at each pipeline stage, with 30% randomly selected nodes performing a randomly chosen attack at a randomly sampled iteration. Each node has a different activation manipulation method (as outlined in Sec. 2.1).

**Training and Validation Loss Evolution for MoE-based Models.**    In Sec. 5, we discussed how SENTINEL extends to MoE-based models such as Llama-4 and DeepSeek-V3. Here, we show the training and validation loss evolution throughout training. As seen from Fig. 11, training loss starts to display higher values after a round of attacks start. However, the validation loss keeps going down similar to the vanilla baseline and match its performance.

**Evolution of Adaptive Deviation Bounds.**    Our approach employs an adaptive IQR-based thresholding mechanism, outline in Alg. 5, that dynamically adjusts acceptable deviation bounds for each monitored metric. To demonstrate the effectiveness of this approach, we present the evolution of these adaptive thresholds alongside the corresponding deviations recorded for each worker across two representative layers of our Llama-3-0.6B model. Fig. 12a shows results under gradient delay attacks, while Fig. 12b illustrates behavior during activation random sign attacks. The results demonstrate that our adaptive bounds effectively encapsulate the normal operational behavior of honest workers during benign training phases. Critically, when a malicious worker initiates an attack, the adaptive bounds enable verifier nodes to immediately detect and flag the anomalous behavior, providing robust protection against adversarial interference in the distributed training process.

**Large-scale Experimental Results.**    In Sec. 5, we discussed scaling our experiments to two additional settings: (1) a $16 \times 16$ mesh topology with 256 workers, and (2) a 1.2B parameter model on an $8 \times 8$ mesh with 64 workers. Here we present the complete results for both configurations across all attack types and malicious ratios tested. Tab. 15 shows detailed performance metrics including validation loss, precision, recall, and F1-score for each experimental condition. Our comprehensive results corroborate the main findings regarding the effectiveness of our proposed verification approach in detecting malicious actors that attempt to disrupt distributed training.

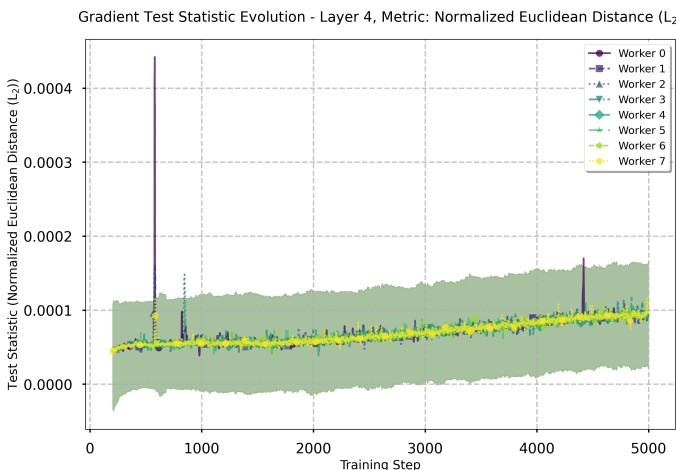

(a) Normalized $L_2$ distance evolution of gradients at layer 4 under gradient delay attack. Worker 0 initiates attack at iteration 4411 and is immediately flagged.

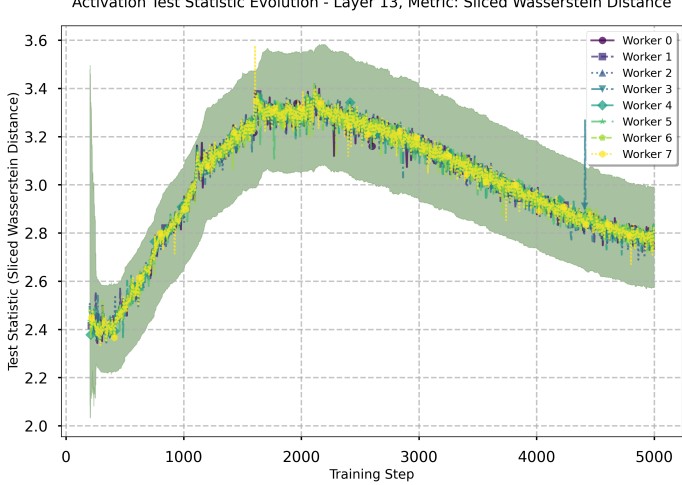

(b) Sliced Wasserstein distance evolution of activations at layer 13 under random sign attack (1%). Workers 7 and 3 initiate attacks at iterations 1600 and 4411, respectively, and are immediately flagged.

Figure 12: Evolution of adaptive deviation bounds and worker statistics. The proposed thresholding mechanism adapts to natural distribution shifts while detecting Byzantine behavior.

Additionally, we demonstrated in Tab. 4 that we can scale the model size easily and SENTINEL still successfully work for these settings. Fig. 13 shows the training and validation loss behavior throughout training. It demonstrates that our approach mitigates the malicious workers that aim to lead the model into divergence and allows the training to continue with a similar behavior to the vanilla training.

**Longer Training & Alternative Architectures.** Two natural questions arise from our approach: whether our verification method remains stable under longer training regimes, and whether it generalizes to alternative transformer architectures beyond our initial experiments. To address these concerns, we conduct extended evaluations on two different models: NanoGPT-0.25B and Llama-3-0.6B, training each for 30,000 iterations. This training duration corresponds to approximately 2B tokens for NanoGPT-0.25B ($7\times$ the parameter count) and 3B tokens for Llama-3-0.6B ($5\times$ the parameter count), following established scaling laws in LLM training (Hoffmann et al., 2022).

Table 15: Attack detection performance for large-scale Llama-3 training on C4 dataset. Metrics shown include precision, recall, F1 score (all as percentages), average detection speed (in iterations), and validation loss. For all experiments, we assume a 37.5% Byzantine workers per stage (thus, for $16 \times 16$ mesh we have 6 malicious vs. 10 honest workers per stage, while for $8 \times 8$ mesh their ratio is 3:5.)

| SETUP | MODE | ATTACK | SENTINEL (OURS) | | | | |
| --- | --- | --- | --- | --- | --- | --- | --- |
| | | | PR. (%) ↑ | RE. (%) ↑ | F1 (%) ↑ | DET. SPEED ↓ | VAL. LOSS ↓ |
| 0.6B ON 16 × 16 MESH | ACTIVATION | Random Value | 100.0 | 100.0 | 100.0 | 7.96 | 3.900 |
| | | Delay (100-steps) | 85.7 | 100.0 | 92.3 | 14.39 | 3.945 |
| | | Bias Addition | 100.0 | 25.6 | 40.8 | 65.15 | 3.981 |
| | | Invisible Noise (99%) | 100.0 | 100.0 | 100.0 | 7.96 | 3.898 |
| | GRADIENT | Random Value | 100.0 | 100.0 | 100.0 | 124.08 | 3.895 |
| | | Delay (100-steps) | 100.0 | 100.0 | 100.0 | 9.05 | 3.890 |
| | | Bias Addition | 100.0 | 100.0 | 100.0 | 1.69 | 3.894 |
| | | Invisible Noise (99%) | 98.7 | 93.6 | 96.0 | 14.27 | 3.915 |
| 1.2B ON 8 × 8 MESH | - | None (Vanilla) | 100.0 | 100.0 | 100.0 | N/A | 3.723 |
| | ACTIVATION | Random Value | 100.0 | 100.0 | 100.0 | 4.33 | 3.723 |
| | | Delay (100-steps) | 37.5 | 100.0 | 54.5 | 67.0 | 3.774 |
| | | Bias Addition | 0.0 | 0.0 | 0.0 | N/A | 3.738 |
| | | Invisible Noise (99%) | 100.0 | 100.0 | 100.0 | 4.33 | 3.727 |
| | GRADIENT | Random Value | 100.0 | 100.0 | 100.0 | 1.0 | 3.726 |
| | | Delay (100-steps) | 100.0 | 100.0 | 100.0 | 1.0 | 3.722 |
| | | Bias Addition | 0.0 | 0.0 | 0.0 | N/A | 3.805 |
| | | Invisible Noise (99%) | 100.0 | 100.0 | 100.0 | 9.2 | 3.725 |

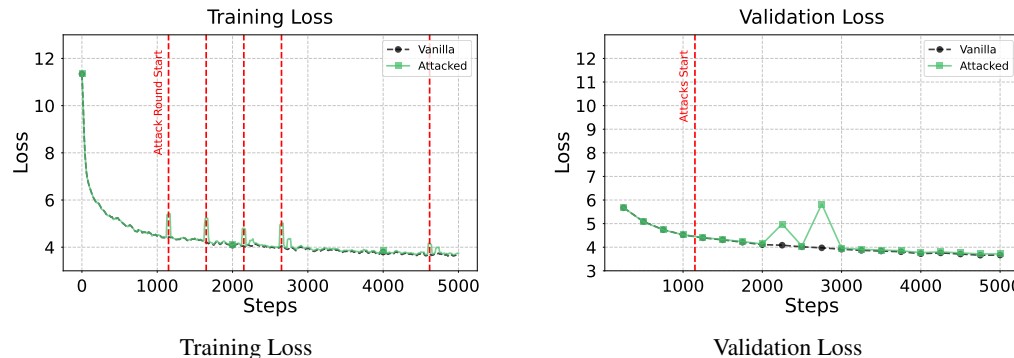

Training Loss                          Validation Loss

Figure 13: Training and validation loss evolution for Llama-3-4B trained on FineWeb dataset over 5k iterations. The mixed activation attack scenario assumes 37.5% Byzantine nodes at each pipeline stage, with 20% randomly selected nodes performing a randomly chosen attack at a randomly sampled iteration. Each node has a different activation manipulation method (as outlined in Sec. 2.1).

We evaluate our method's robustness by simulating a challenging adversarial environment where 50% of nodes are malicious at each transformer stage. At each attack round, one randomly selected malicious node performs a randomly chosen attack from Tab. 12 under a "no collusion" assumption. Tab. 16 summarizes our detection performance across this extended training period for both architectures.

Our results demonstrate consistent stability across both model architectures and all three datasets, achieving high F1-scores ($> 81\%$) for attack detection. Importantly, we observe that the median detection speed across all successfully detected attack types is 5.0 iterations, which aligns precisely with our acceptable number of violations threshold. This indicates that we can detect the majority of attacks with significant training impact within our predefined tolerance window. Even when some attacks remain undetected, they exhibit negligible impact on training convergence, as illustrated in Figs. 15 and 16. The validation loss under our verification method closely tracks the vanilla baseline throughout the entire training duration for both models, corroborating our theoretical analysis from Theorem 1 and demonstrating the method's architectural flexibility and long-term stability.

Table 16: Detection performance for training NanoGPT-0.25B and Llama-3-0.6B against mixed attacks for 30k iterations.

| MODEL | DATASET | SENTINEL (OURS) | | | | | | VANILLA |
|---|---|---|---|---|---|---|---|---|
| | | PR. (%) ↑ | RE. (%) ↑ | F1 (%) ↑ | MED. SPEED ↓ | AVG. SPEED ↓ | VAL. LOSS ↓ | VAL. LOSS ↓ |
| NanoGPT-0.25B | C4 | 91.2 | 86.1 | 88.6 | 5.0 | 44.07 | 3.747 | 3.650 |
| | FW | 91.2 | 86.1 | 88.6 | 5.0 | 62.13 | 3.752 | 3.731 |
| | OW | 73.3 | 91.7 | 81.5 | 5.0 | 320.43 | 3.571 | 3.531 |
| LLAMA-3-0.6B | C4 | 76.0 | 86.4 | 80.9 | 5.0 | 263.51 | 3.357 | 3.347 |
| | FW | 88.6 | 88.6 | 88.6 | 5.0 | 108.73 | 3.465 | 3.459 |
| | OW | 78.0 | 88.6 | 83.0 | 5.0 | 44.06 | 3.316 | 3.313 |

**Complementary Nature of DP Defense Methods to SENTINEL.** Throughout the paper, we explained in detail how the defense methods in the data parallel domain proposed by prior Byzantine-tolerant literature (e.g., please see (Mhamdi et al., 2018; Gorbunov et al., 2022; Malinovsky et al., 2024)) are complementary to SENTINEL that secures the pipeline parallel dimension by verifying the activations and activation gradients transmitted during forward and backward pass. To demonstrate this complementary nature, we combine our method with two well-know robust aggregation techniques used to defend against parameter gradient attacks. In particular, we use Krum (Blanchard et al., 2017) and Bulyan (Mhamdi et al., 2018) instead of vanilla gradient averaging. Our goal is to show that these methods would not impact the operation of SENTINEL in a negative way as they are operating in an orthogonal dimension. Fig. 14 depicts the training and validation loss for a Llama-3-0.6B model trained against mixed attacks. As seen, **presence of robust aggregation methods does not interfere with how SENTINEL works**, and our method can ensures a convergence rate that follows the vanilla, non-attacked baseline.

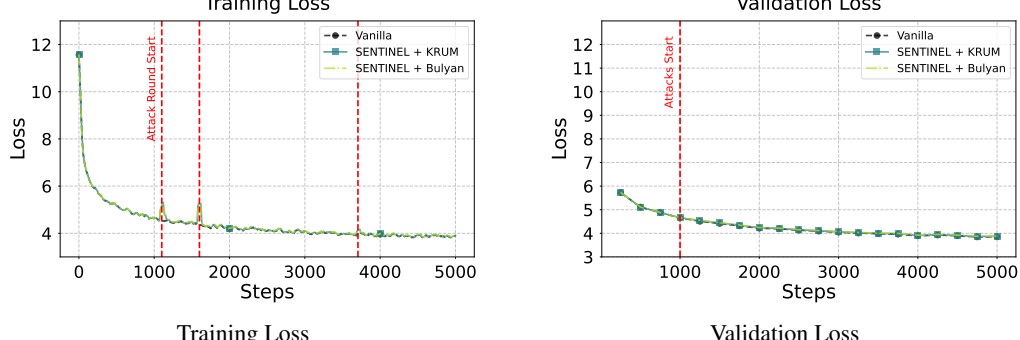

Training Loss

Validation Loss

Figure 14: Training and validation loss evolution for Llama-3-0.6B trained on FineWeb dataset over 5k iterations **in the presence of robust aggregation methods during gradient averaging**. The mixed attack scenario assumes 37.5% Byzantine nodes at each pipeline stage, with 33% randomly selected nodes performing a randomly chosen attack at a randomly sampled iteration. Each node has a different manipulation method (as outlined in Sec. 2.1).

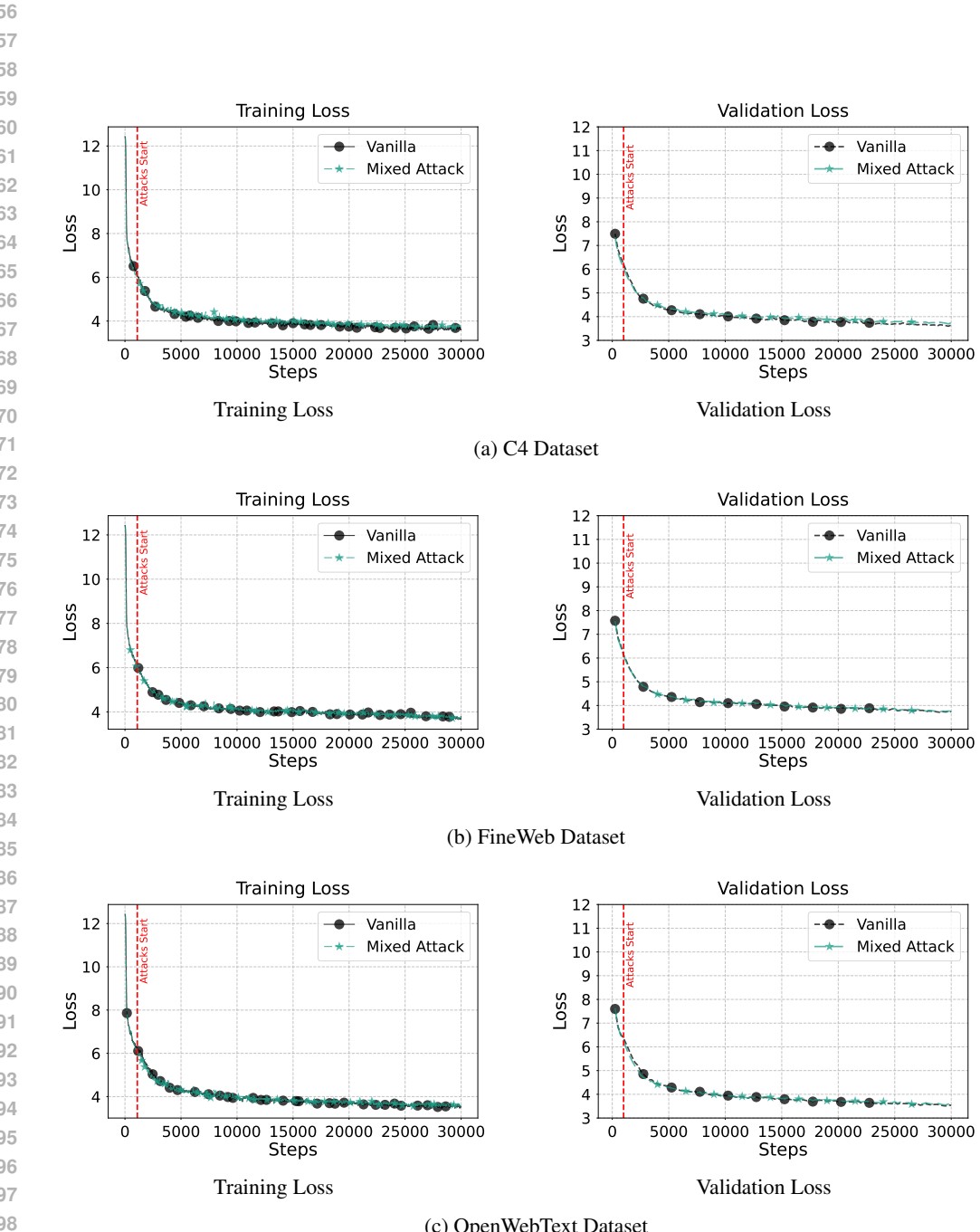

(a) C4 Dataset

(b) FineWeb Dataset

(c) OpenWebText Dataset

Figure 15: Training and validation loss evolution for **NanoGPT-0.25B** model on C4, FineWeb, and OpenWebText datasets over 30k iterations. The mixed attack scenario assumes 50% Byzantine nodes at each pipeline stage, with one randomly selected node performing a randomly chosen attack at a randomly sampled iteration. Each node has a different mode (activation vs. gradient) and manipulation method (as outlined in Sec. 2.1).

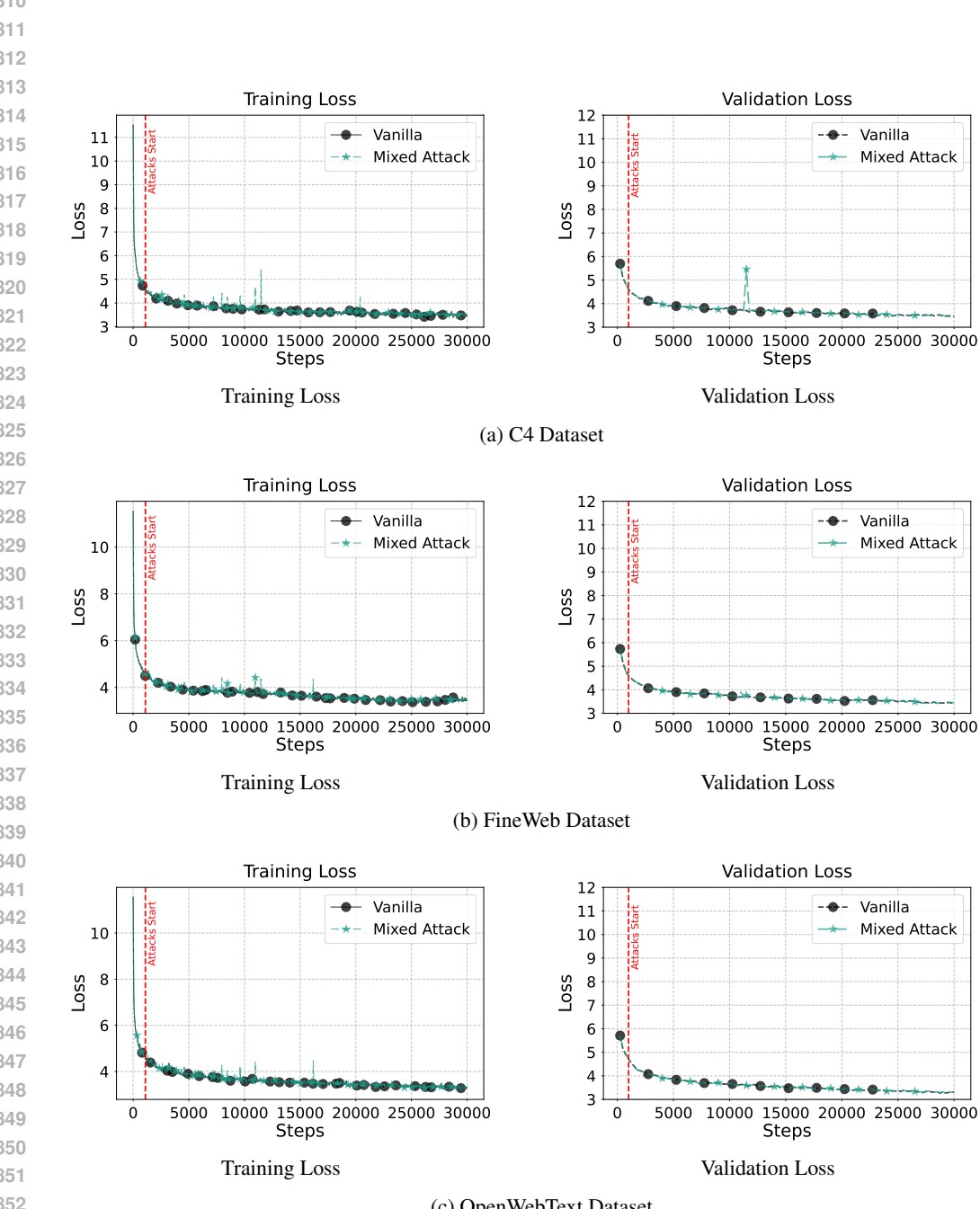

(a) C4 Dataset

(b) FineWeb Dataset

(c) OpenWebText Dataset

Figure 16: Training and validation loss evolution for **Llama-3-0.6B** model on C4, FineWeb, and OpenWebText datasets over 30k iterations. The mixed attack scenario assumes 50% Byzantine nodes at each pipeline stage, with one randomly selected node performing a randomly chosen attack at a randomly sampled iteration. Each node has a different mode (activation vs. gradient) and manipulation method (as outlined in Sec. 2.1).

## F.3 ADDITIONAL ABLATION STUDIES

In this section, we present additional ablation studies on the impact of various components used in SENTINEL.

**Impact of EMA in Verification.** To demonstrate how the temporal training dynamics captured within the EMA affects the detection of malicious workers, we compare our approach against a naïve version that simply compares submitted signals with the average. For this experiment, we start training a Llama-3-0.6B model while malicous workers employ activation delay attacks. To cancel out the EMA, we set $\beta = 0$ which essentially means that we would compare against the instantaneous activation average. Our results are show in Tab. 17. As seen, comparison with the instantaneous mean is not enough to protect against malicious workers. This corroborates the importance of temporal patterns in detecting pipeline parallel attacks. Additionally, we measure sensitivity of SENTINEL to the EMA decay rate by modifying $\beta_h = \beta_g$ to 0.6 and 0.99. As shown in Tab. 17, the sensitivity to the decay rates is not significant.

Table 17: Ablation study on the impact of EMA in SENTINEL.

| REFERENCE POINT | METRICS | | | |
|---|---|---|---|---|
| | PR. (%) ↑ | RE. (%) ↑ | F1 (%) ↑ | VAL. LOSS ↓ |
| AVERAGE ($\beta_h = \beta_g = 0$) | 23.08 | 100.0 | 37.5 | 6.248 |
| SENTINEL ($\beta_h = 0.9, \ \beta_g = 0.8$) | **100.0** | **100.0** | **100.0** | **3.826** |
| SENTINEL ($\beta_h = \beta_g = 0.60$) | 94.7 | **100.0** | 97.3 | 3.894 |
| SENTINEL ($\beta_h = \beta_g = 0.99$) | 90.0 | **100.0** | 94.7 | 3.875 |

**Impact of Distance Metrics.** The sensitivity of different attacks to various distance metrics varies between activation and activation gradient attacks, which is why we require multiple distance metrics. As discussed in Sec. 3.1, other optimal distance metric choices could provide a unified solution (e.g., neural network classifiers), which we defer to future work. Here, to evaluate the impact of various distance metrics, we conducted an ablation study using the mixed attack setting from Tab. 2 for training a Llama-3-0.6B on the C4 dataset, employing only one distance metric at a time for detection against mixture attacks. Our results are shown in Tab. 18. As seen, combining all metrics yields optimal performance against the mixture of all activation and activation gradient attacks.

Table 18: Ablation study on distance metrics against mixed attacks.

| DISTANCE METRIC | METRICS | | | |
|---|---|---|---|---|
| | PR. (%) ↑ | RE. (%) ↑ | F1 (%) ↑ | VAL. LOSS ↓ |
| SFR | 42.9 | 83.3 | 56.6 | 8.882 |
| SWD | 40.0 | **94.4** | 56.2 | 6.332 |
| NORMALIZED $L_2$ | 75.0 | 75.0 | 75.0 | 10.274 |
| ABSOLUTE DEVIATION $L_1$ | 71.1 | 88.9 | 79.0 | 3.883 |
| ALL (SENTINEL) | **83.7** | 92.3 | **87.8** | **3.831** |

**Impact of Random Seeds.** Due to limited computational resources and substantial experimental costs, we have not reported error bars throughout the paper. To demonstrate the statistical integrity of our approach, we computed error bars for two of the most challenging activation attacks: delay (100-steps) and invisible noise (99%). We randomly selected 5 seeds and repeated the experiments from Tab. 1, randomizing both network initialization and malicious worker selection. The results provided in Tab. 19 indicate the statistical significance of our findings.

## G SENTINEL IN THE WILD: VERIFICATION FOR DECENTRALIZED LLM TRAINING USING SWARM PARALLELISM

In this section, we detail the adaptation of SENTINEL to SWARM parallelism (Ryabinin et al., 2023) and showcase its capabilities for decentralized training of LLMs. We first provide a high-level overview of SWARM's operational dynamics. Then, we demonstrate the compatibility of our

Table 19: Statistical significance analysis with error bars for the performance of SENTINEL against activation delay and invisible noise attack.

| ATTACK | METRICS | | | |
|---|---|---|---|---|
| | PR. (%) ↑ | RE. (%) ↑ | F1 (%) ↑ | VAL. LOSS ↓ |
| DELAY (100-steps) | $99.2 \pm 1.6$ | $100.0 \pm 0.0$ | $99.6 \pm 0.8$ | $3.843 \pm 0.008$ |
| INVISIBLE NOISE (99%) | $100.0 \pm 0.0$ | $100.0 \pm 0.0$ | $100.0 \pm 0.0$ | $3.832 \pm 0.004$ |

verification mechanism with communication-efficient compression techniques employed in distributed SWARM training. We describe how SENTINEL integrates with SWARM's existing infrastructure by leveraging its trainer node architecture. Additionally, we analyze the critical role of verification in the presence of SWARM's stochastic wiring mechanism, which introduces additional failure modes beyond traditional PP. Finally, we present comprehensive experimental settings and detailed results from our SWARM experiments.

### G.1 SWARM PARALLELISM OVERVIEW

As briefly discussed in Sec. 5.1, SWARM parallelism can be viewed as a stochastic DP/PP training approach. At a high level, each worker gets assigned a random layer/stage of the model to process (PP-axis) while other workers process different micro-batches for the same stage (DP-axis). Unlike traditional fixed meshes used in distributed training frameworks such as `torch.distributed` (Paszke et al., 2019; Li et al., 2020), SWARM parallelism operates as pools of workers available at each stage that process data for subsequent stages.

Coordination of training in such a stochastic environment is handled by so-called trainer nodes. Each trainer node is responsible for processing a single micro-batch of data end-to-end. In particular, each trainer node receives a disjoint micro-batch of data and sends it to a worker at stage 0. Once the worker processes the data, the trainer receives the result and forwards it to the next stage. The trainer continues this process until the micro-batch has passed through all layers in the forward pass. Then, the trainer begins the backward pass by traversing the stages in reverse order. With this architecture, workers do not need to save the intermediate activations for backward since the trainer maintains all necessary information and can send it to different workers during the backward pass. The workers then accumulate parameter gradients locally and update their parameters after an all-reduce operation with all existing workers of the same stage.

Trainer nodes use a Distributed Hash Table (DHT) to route their micro-batches. Since devices at each stage process data at different speeds, trainers maintain worker load and throughput information in the DHT so that all other trainers know when to send signals and which worker at each stage processes data most efficiently. This mechanism is at the heart of the *stochastic wiring* that occurs within SWARM and distinguishes it from fixed mesh DP/PP approaches (Ryabinin et al., 2023). As demonstrated, a SWARM system can accommodate many trainer nodes since each processes a single micro-batch, and typically multiple devices at each stage are capable of processing data batches. For a visual representation of SWARM, please refer to Fig. 17a. A pseudo-code of how the trainer nodes work in SWARM is also provided in Alg. 6.

### G.2 SUBSPACE COMPRESSION

Regular SWARM parallelism enables decentralized training, but suffers from slow speeds due to bandwidth requirements (Ryabinin et al., 2023). To relax this requirement and enable faster training, recent work by Ramasinghe et al. (2025) has shown that training with speeds as low as 60 Mbps is plausible using lossless compression. In particular, Ramasinghe et al. (2025) observes that most pretrained transformers already exhibit low-rank properties in their feedforward and attention projection layers. They utilize this observation to construct a lossless compression mechanism using a shared subspace that enables faster transfer of activations and gradients passed between layers in PP.

In simple terms, given a signal $h^{(s)} = f_s(h^{(s-1)})$ that has been processed by a worker at stage $s$, instead of communicating $h^{(s)} \in \mathbb{R}^{b \times n \times d}$, they compress it using a subspace compression matrix

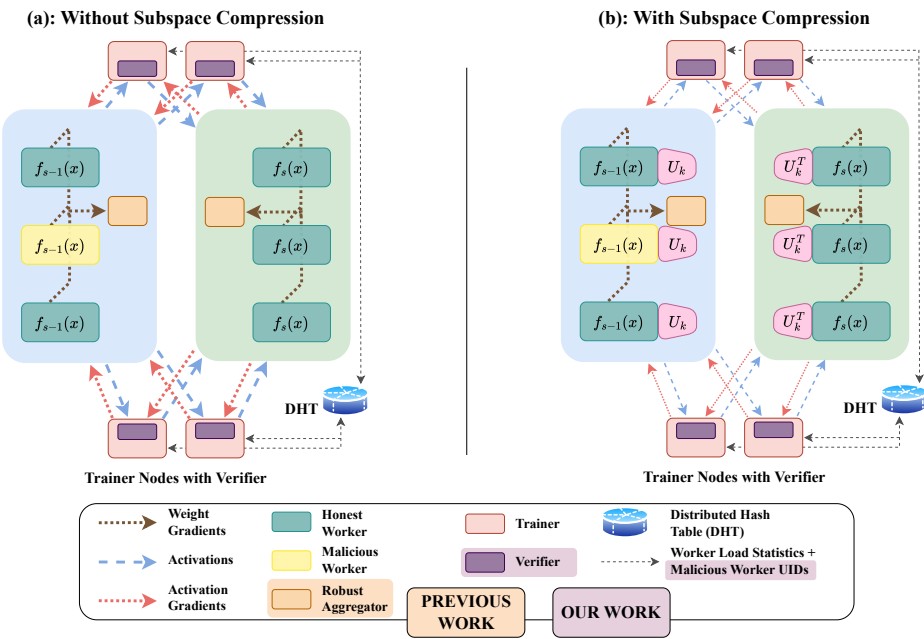

Figure 17: (a) SWARM (Ryabinin et al., 2023) parallelism utilizes pipeline and data parallelism to train neural networks. Compared to traditional DP/PP methods, SWARM utilizes trainer nodes to stochastically route micro-batches of input data through the model layers. Trainer nodes leverage a Distributed Hash Table (DHT) to communicate worker load/throughput to enable a better device utilization within the SWARM. (b) Subspace compression (Ramasinghe et al., 2025) complements SWARM by adding lossless compression at the boundary of worker devices for faster communication of activations and activation gradients from/to trainer nodes. SENTINEL can be adapted to both settings and enable worker verification. It sits naturally within the trainer nodes that are already coordinating signal transmission among workers.

$U_k$ that projects it to a $k$-dimensional subspace where $k \ll d$ (Ramasinghe et al., 2025):

$$h_{\text{compressed}}^{(s)} = (h^{(s)} - \text{PE} - \mathbf{T}_{\text{fixed}}[t_{1:n}, :])U_k. \tag{68}$$

Here, PE denotes the positional embeddings and $\mathbf{T}_{\text{fixed}}[t_{1:n}, :])$ are the fixed token embeddings. At layer $s + 1$, the receiver decompresses this using (Ramasinghe et al., 2025):

$$h_{\text{recovered}}^{(s)} = h_{\text{compressed}}^{(s)}U_k^T + \text{PE} + \mathbf{T}_{\text{fixed}}[t_{1:n}, :] = h^s \tag{69}$$

This compression scheme, tailored for training transformer-based LLMs, can enable training across four geographical regions with bandwidth as low as 60 Mbps. Therefore, we will incorporate it as part of our realistic SWARM integration. For an overview of SWARM with subspace compression, please refer to Fig. 17b.

### G.2.1 SENTINEL COMPATIBILITY WITH SUBSPACE COMPRESSION

To demonstrate the compatibility of our proposed method with subspace compression, we integrate this compression algorithm into our fixed mesh `TorchTitan` implementation. We assume that malicious workers would manipulate the compressed signals since these are what is being sent to subsequent workers.

In particular, we use the first $k$ dimensions of the transmitted signal $h_{\text{compressed}}^{(s)} \in \mathbb{R}^{b \times n \times k + 1}$, as the last dimension corresponds to the fixed token embeddings which may vary significantly between different batches. Specifically, SENTINEL uses $h_{\text{compressed}}^{(s)}[:, :, :-1]$ for activation verification using EMA (and similarly $g_{\text{compressed}}^{(s)}[:, :, :-1]$ for gradients). The rest of the algorithm remains unchanged.

---

**Algorithm 6** SWARM Trainer Node (Ryabinin et al., 2023)

---

```python
class TrainerNode:
    def __init__(self, dht, stage_uids):
        self.dht = dht
        self.stages = [RemoteExpert(uid, dht) for uid in stage_uids]

    def forward(self, input_batch):
        """Process microbatch through all pipeline stages"""
        hidden = input_batch
        for stage in self.stages:
            hidden = stage.process(hidden)
        return hidden

class RemoteExpert:
    def __init__(self, stage_uid, dht):
        self.stage_uid = stage_uid
        self.dht = dht

    def process(self, data):
        """Route data to available worker at this stage"""
        worker = self.dht.find_available_worker(self.stage_uid)
        result = worker.forward(data)
        return result
```

---

Using these assumptions, we train our 16-layer Llama-3-0.6B on the FineWeb-EDU dataset using the settings given in Tab. 10. The only difference is that for these experiments we use a local batch-size 8 with 8 gradient accumulation steps to reach a target batch-size of 512 per optimizer step. For subspace compression, we use a compression factor of 25 for 96% compression. As demonstrated by our results in Tab. 20, SENTINEL can easily be adapted to this setting and demonstrates efficient detection capability against various kinds of attacks. We also plot the validation loss for all these attacks in Fig. 18, demonstrating uninterrupted training in all cases.

Table 20: Attack detection performance across different attack modes for training Llama-3-0.6B with subspace compression (Ramasinghe et al., 2025). Metrics shown include precision, recall, F1 score (all as percentages), and validation loss at 5000 steps. In all scenarios, each stage has 3:5 malicious to honest ratio.

| MODE | ATTACK | DETECTION PERFORMANCE | | | TRAINING |
|---|---|---|---|---|---|
| | | PR. (%) ↑ | RE. (%) ↑ | F1 (%) ↑ | VAL. LOSS ↓ |
| ACTIVATION | Constant (Zeros) | 100.00 | 100.00 | 100.00 | 4.2490 |
| | Constant (Ones) | 100.00 | 100.00 | 100.00 | 4.2486 |
| | Random Value | 100.00 | 100.00 | 100.00 | 4.2999 |
| | Bias Addition (Constant) | 100.00 | 75.00 | 85.71 | 4.2484 |
| | Bias Addition (Random) | 0.00 | 0.00 | 0.00 | 4.3248 |
| | Delay (100-steps) | 100.00 | 100.00 | 100.00 | 4.2677 |
| GRADIENT | Constant (Zeros) | 100.00 | 100.00 | 100.00 | 4.2470 |
| | Constant (Ones) | 100.00 | 100.00 | 100.00 | 4.2639 |
| | Random Value | 100.00 | 100.00 | 100.00 | 4.2726 |
| | Bias Addition (Constant) | 80.00 | 100.00 | 88.89 | 4.2751 |
| | Bias Addition (Random) | 100.00 | 100.00 | 100.00 | 4.2608 |
| | Delay (100-steps) | 100.00 | 100.00 | 100.00 | 4.2567 |
| **MIXED MODE** | | 96.97 | 88.89 | 92.75 | 4.2779 |

## G.3 SENTINEL INTEGRATION WITH SWARM

Now that we have laid out the background on SWARM and how subspace compression acts as a complementary factor that enables decentralized training under restricted bandwidths, let us detail how SENTINEL can be integrated into this realistic, production-ready decentralized training ecosystem.

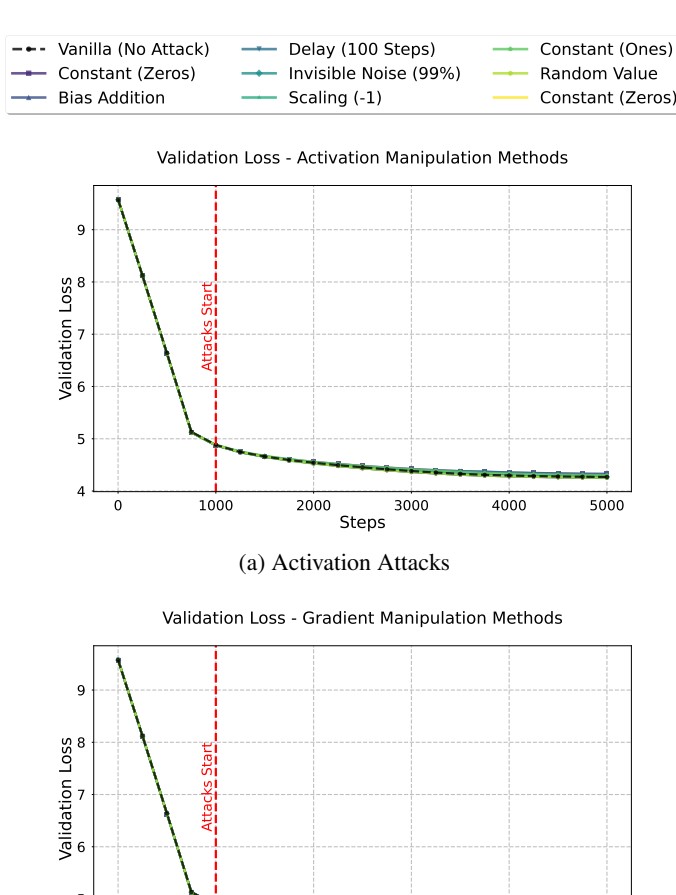

(a) Activation Attacks

(b) Gradient Attacks

Figure 18: Validation loss evolution for training Llama-3-0.6B models that employ subspace compression (Ramasinghe et al., 2025) under various activation/gradient manipulation attacks. Despite the compression in the activation/gradients, SENTINEL can successfully prevent divergence from vanilla training loss.

As discussed, trainer nodes are in a natural position to take on the verifier role. As mentioned in App. A Q3, trainer nodes only perform CPU-based operations and are readily available at a fraction of worker node costs. Thus, it is not far-fetched to operate such nodes for training coordination. Additionally, data integrity heavily depends on the trustworthiness of trainer nodes, and if this role were delegated to untrusted parties, other major issues regarding backdoor and privacy attacks would arise. Therefore, it is both economically feasible and logically crucial to have trusted trainer nodes.

To add SENTINEL verification to trainer nodes, we employ a mechanism for trainers to maintain an EMA of the signals they distribute across different stages. In particular, each trainer stores the EMA of all layer outputs since it operates on them end-to-end. Specifically, trainer $i$ maintains:

$$\{\boldsymbol{m}_{t,i}^{(s)} \mid 1 \leq s \leq p\} \tag{70}$$

where

$$\boldsymbol{m}_{t,i}^{(s)}(\boldsymbol{h}) = \beta_h \boldsymbol{m}_{t-1,i}^{(s)}(\boldsymbol{h}) + (1 - \beta_h)\boldsymbol{h}_{t,i}^{(s,r)}, \tag{71}$$

is the momentum at step $t$ and stage $s$. A similar set is also kept for gradient EMAs, but we omit it for brevity. Using these EMA states, each trainer can run SENTINEL verification as signals are being processed by the workers.

---

**Algorithm 7** SWARM Trainer Node with SENTINEL Verification

---

```python
class SENTINELTrainerNode:
    def __init__(self, dht, stage_uids):
        self.dht = dht
        self.stages = [RemoteExpert(uid, dht) for uid in stage_uids]
        self.ema_detector = EMADetector()
        self.banned_workers = set()

    def forward(self, input_batch):
        """Process microbatch with Sentinel verification"""
        hidden = input_batch
        for i, stage in enumerate(self.stages):
            # Process data at current stage
            hidden = stage.process(hidden)

            # Sentinel verification
            is_suspicious = self.ema_detector.update_and_detect(i, hidden)

            if is_suspicious:
                worker_uid = stage.get_last_worker_uid()
                self._report_violation(worker_uid)

        return hidden

    def _report_violation(self, worker_uid):
        """Report violation to DHT for global coordination"""
        violations = self.dht.get(f"violations_{worker_uid}", default=0)
        self.dht.store(f"violations_{worker_uid}", violations + 1)

        if violations + 1 > MAX_VIOLATIONS:
            self.dht.store(f"banned_{worker_uid}", True)
            self.banned_workers.add(worker_uid)
```

---

The challenging aspect of implementing SENTINEL in SWARM is communicating malicious behavior among trainers. Since each trainer is responsible for maintaining a separate EMA and verifying their randomly chosen workers independently, this might increase the malicious impact of bad actors. To address this issue, we utilize the DHT for lightweight communication between trainers about malicious workers. Instead of each trainer independently tracking violation counters or worker bans, they cooperate through the DHT to increment violation counters or ban workers collectively. In simple terms, we track the number of violations for each worker through their unique identifiers (UIDs) in the DHT, and if they surpass the allowed number of violations, the first trainer that observes this bans them. Similarly, we periodically check whether workers are demonstrating good behavior after transient violations and decrease their violation counts through our forgiveness strategy discussed in SENTINEL. The only slight modification that makes SENTINEL work better in SWARM is replacing tainted gradients with zero tensors, which we observed makes training more stable.

A pseudo-code of SENTINEL integration with SWARM is given in Alg. 7 (backward pass is omitted for brevity).

### G.3.1 IMPORTANCE OF VERIFICATION UNDER STOCHASTIC WIRING

An interesting observation that we made through implementing SENTINEL in SWARM is the importance of highly calibrated detection thresholds in removing abundant false positives due to the interplay between "cascading effect" in PP and "stochastic wiring" in SWARM. As discussed in Ryabinin et al. (2023), stochastic wiring helps the SWARM to move towards maximum device utilization and less idle time. However, this behavior can have an adverse effect considering adversarial actors. Specifically, if a trainer fails to detect/flag a bad actor, it not only updates its own EMA with corrupted signals, but other trainers would also be in danger since they would eventually use that malicious worker while routing their micro-batches due to SWARM's stochastic wiring. Thus, if a malicious worker goes undetected, it can corrupt the EMA of all trainers and they could flag all

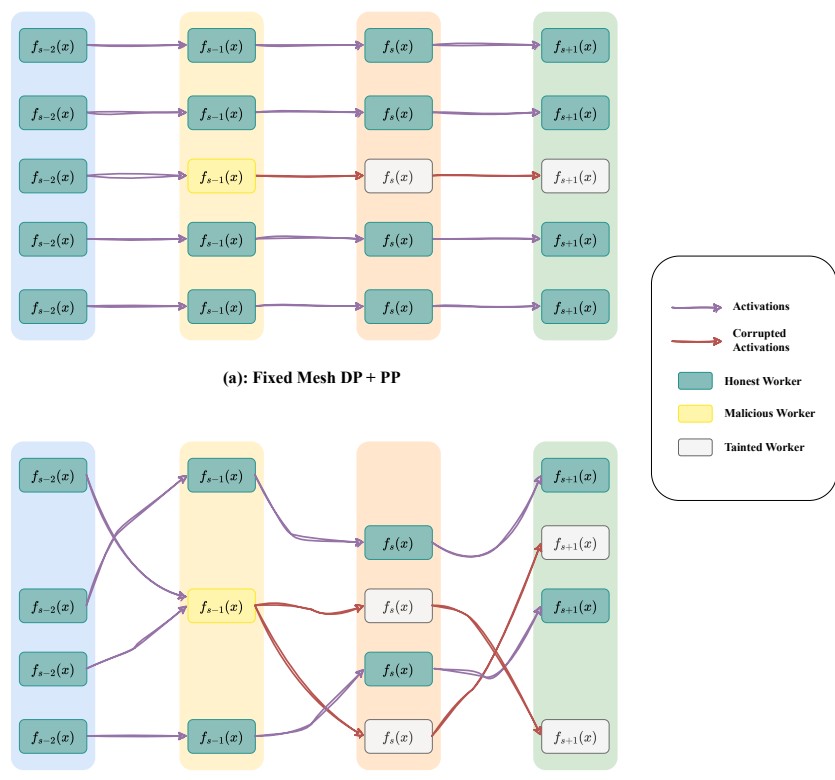

(a): Fixed Mesh DP + PP

(b): Stochastic DP + PP (SWARM with Stochastic Wiring)

Figure 19: Visual comparison of (a) fixed mesh DP + PP vs. (b) SWARM (Ryabinin et al., 2023). Unlike fixed mesh DP + PP where each worker sends their data to a specific worker in the next stage, SWARM utilizes stochastic wiring to route the data batches between stages. This is to ensure that workers would reach their maximum utilization and have minimum idle time. This behavior, however, is a double-edge sword as by stochastically routing data batches between workers, we are effectively increasing the reachability of malicious workers that can pollute "any" worker in the subsequent stage if they go undetected. This highlights the importance of highly calibrated verification mechanisms in the real-world. Note that the signals transmitted between workers in SWARM all pass through trainer nodes, and hence, there is no peer-to-peer communication between workers. We just directly connected the workers for a better visual comparision with fixed mesh.

honest workers in that stage as malicious. In the fixed mesh (shown in Fig. 19a), in the worst case we would wrongly flag all workers from a single pipe due to the "cascading effect" that we discussed in App. D.2. In SWARM, however, we are in danger of falsely accusing all workers of the same stage due to "stochastic wiring" which when considered together with "cascading", could mean a high false positive rate for the detection algorithm. We defer further investigation into this interesting inter-play to future work. This phenomenon has been elaborated in Fig. 19b.

## G.4 DETAILED EXPERIMENTAL RESULTS

To evaluate our SENTINEL integration with SWARM under realistic distributed training conditions including subspace compression, we conduct comprehensive experiments that test both the detection capabilities and the system's robustness under various attack scenarios. Our experimental setup is designed to validate the effectiveness of the proposed method under realistic conditions.

### G.4.1 EXPERIMENTAL SETTINGS

**Distributed Architecture.** We use a Llama-3-0.6B model with 16 transformer layers, partitioned into 16 PP stages for SWARM. Each stage employs 8 parallel workers to process micro-batches concurrently, resulting in a total of 128 worker nodes. To simulate realistic distributed environments,

Table 21: Detection performance of SENTINEL in a distributed SWARM with 128 workers. There are 37.5% malicious workers that are submitting randomly chosen mixed gradient and activation attacks. Note that the low recall is attributed to some nodes employing weak attacks that are not disruptive to training, hence they do not get flagged as malicious but training continues without disruption. This is in line with our observation in Fig. 1 and intuition from Theorem 1 that weak attackers may survive detection but they are no harm to the training.

| ATTACK MODE | METRICS | | |
|---|---|---|---|
| | PR. (%) ↑ | RE. (%) ↑ | F1 (%) ↑ |
| MIXED GRADIENT | 100.0 | 75.0 | 85.7 |
| MIXED ACTIVATION/GRADIENT | 100.0 | 66.7 | 80.0 |

each worker is deployed on a separate AWS instance with no direct interconnection to other nodes. We use heterogeneous instance types based on computational requirements: `g5.2xlarge` instances with NVIDIA A10G GPUs for the embedding and first transformer layer, `g5.4xlarge` instances with NVIDIA A10G GPUs for the final transformer layer and projection head, and `g4dn.2xlarge` instances with NVIDIA T4 GPUs for all intermediate transformer layers.

The trainer infrastructure consists of 32 parallel trainers distributed across 4 `c6a.8xlarge` CPU instances, with 8 trainers running simultaneously per instance. Each trainer processes a disjoint data shard streamed from FineWeb-EDU (Penedo et al., 2024) via HuggingFace (Lhoest et al., 2021), ensuring no data overlap during training. This configuration allows us to test the scalability of our DHT-based violation reporting system under realistic deployment conditions.

**Attack Scenarios.** We evaluate SENTINEL's detection performance under two distinct scenarios:

1. **Gradient-Only Attacks**: A heterogeneous mixture comprising constant attacks (zeros and ones), bias addition, random value injection, and scaling attacks.
2. **Combined Gradient and Activation Attacks**: An expanded attack suite including a mixture of gradient and activation attacks such as constant (zeros and ones), bias addition, random value, scale, delay, and random sign attacks. This more sophisticated threat model evaluates SENTINEL's comprehensive detection capabilities across the entire forward and backward pass.

For both scenarios, we simulate a challenging environment with a 3:5 malicious-to-honest worker ratio (37.5% malicious workers). In the gradient-only setting, we assume attackers operate independently without coordination, initiating attacks at random intervals. For the combined attack scenario, we model partial coordination where 15% of malicious workers launch synchronized attacks at random intervals, simulating coordinated adversarial behavior while maintaining realistic assumptions about attacker capabilities.

**Training Configuration.** We use micro-batches of size 4 per worker with a total target batch-size of 512 per optimizer step. For subspace compression, we use a compression factor of 25. All other hyper-parameters and training settings are the same as provided in Tab. 10. Given the costs associated with running 128 SWARM workers, we chose to train for 2500 steps only. Nevertheless, we made sure to squeeze all the attack start times within those steps.

### G.4.2 RESULTS

In this section, we present our results. From Fig. 20, we can see that in the absence of a viable verification mechanism while training across an untrusted distributed environment, training can easily be disrupted. In Tab. 21 we also present SENTINEL's detection performance where we achieve greater than 80% F1-score. Note that lower recall in mixed activation/gradient attacks is due to some nodes employing weak attacks that are not disruptive to training, hence they do not get flagged as malicious but training continues without disruption. This is in line with our observation in Fig. 1 and our intuition from Theorem 1 that weak attackers may survive detection but do not harm the training. This can be seen in Fig. 20 that training continues without divergence in both cases. These results prove the versatility of SENTINEL in real-world applications involving distributed training.

**EMA Variance across Trainers.** When employing SENTINEL in SWARM, we discussed how each trainer accumulate their own version of EMAs. Thus, these signals can vary from one trainer

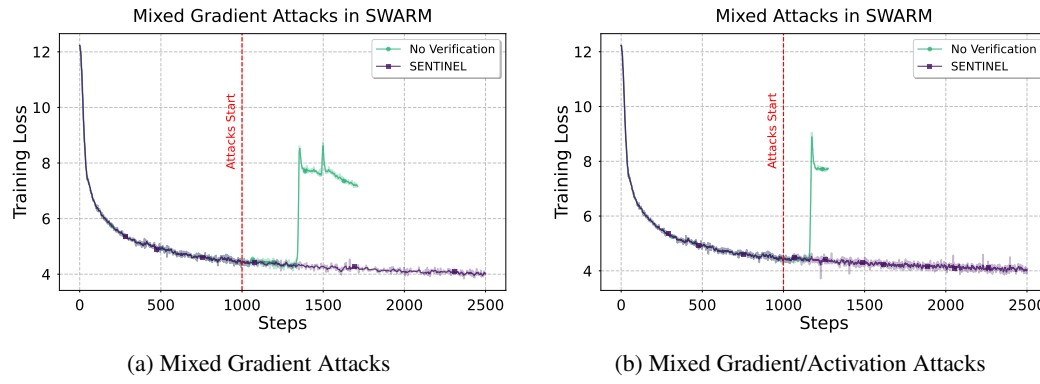

(a) Mixed Gradient Attacks          (b) Mixed Gradient/Activation Attacks

Figure 20: Loss when training Llama-3-0.6B models with subspace compression (Ramasinghe et al., 2025) in a distributed SWARM (Ryabinin et al., 2023) of 128 workers. Workers employ various activation/gradient manipulation attacks to disrupt training. While in the absence of verification training gets disrupted, SENTINEL can successfully protect training from divergence by detecting and banning malicious workers.

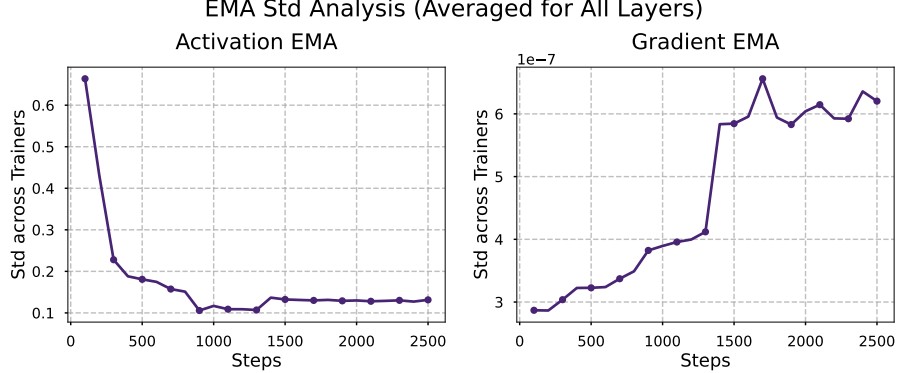

Figure 21: Standard deviation of EMA signals (activation/gradient) between the 32 trainers used in training our Llama-3-0.6B model. Since each trainer keeps the EMAs of all layers, here we report the average across all layers. As seen, activation EMA variance decreases as training progresses while gradient EMA variance increases among trainer. Despite this increase, note that the scale of gradient EMA variance is very close to zero ($10^{-7}$).

to another. If this variance is large, it may cause verification disruption: a worker that appears honest to one trainer could be flagged as malicious solely due to EMA variance between trainers. To demonstrate that EMAs among trainers have minimum divergence, we plot the standard deviation of the activation and activation gradient EMA among all our 32 trainers used for training in SWARM. As seen in Fig. 21, both activation and activation gradient display a controlled amount of variance across trainers which is a testament to the fact that EMAs in different trainers evolve similarly.

**Test Statistic Evolution.** Finally, we examine how SENTINEL is utilized by disjoint trainer nodes in SWARM. To this end, we track the test statistics for a particular worker (worker number 7 at layer 12) when it processes micro-batches from different trainers. We also track the lower and upper thresholds determined using our adaptive IQR mechanism from Alg. 5. Fig. 22 shows the evolution of these test statistics over time. From this figure, we conclude that despite no direct EMA or threshold communication between trainer nodes, all 32 trainers exhibit similar evolutionary patterns. This consistency is crucial because each trainer routes its micro-batches through different workers, requiring their detection criteria to remain aligned. Without this alignment, the system could suffer from false positives or false negatives that would disrupt training stability. The key benefit of using EMA within SENTINEL is its ability to efficiently track historical patterns, enabling this coordinated behavior across distributed trainers.

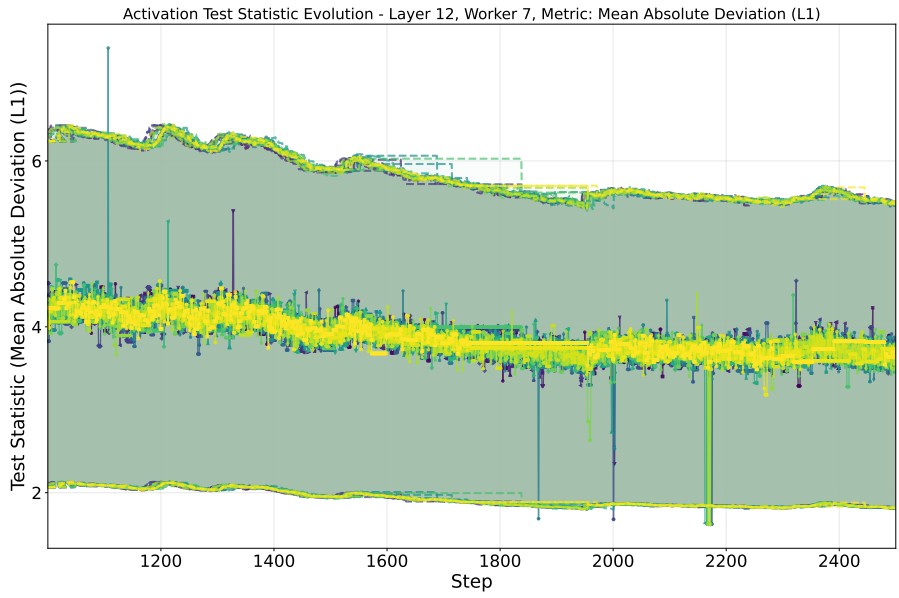

Figure 22: Evolution of adaptive deviation bounds and worker statistics for a worker processing layer 12 in our 16 layer Llama-3-0.6B. Each color represents the test statistics recorded by a different trainer. The upper and lower thresholds determined by our adaptive IQR mechanism are also shown as the top and bottom lines for each trainer. Despite no direct EMA or threshold communication between the trainers, they usually have a similar threshold. The worker starts their attack around step 2150 after which the trainers flag and ban the worker.

