# OpenReview forum: "SENTINEL: StagewisE iNtegriTy verification for pIpeliNe parallEL decentralized training"
_ICLR.cc/2026/Conference — Submitted to ICLR 2026_

### Official Review · Reviewer_ux6h · 2025-10-29

**Soundness:** 3
**Presentation:** 3
**Contribution:** 2
**Rating:** 6
**Confidence:** 4

**Summary:**

The paper addresses an underexplored yet relevant threat model unique to PP training, with clear formulation and empirical validation (Tabs. 1–4). The proposed SENTINEL method demonstrates strong detection performance and low overhead, supported by theoretical reasoning (Sec. 3; Sec. 3.2 and Appendix). Theoretical analysis regarding convergence under undetected bounded corruption supports the approach (Thm. 1). However, several aspects require clarification or extension: limited empirical baselines against existing PP security methods (No direct evidence found in the manuscript), potentially strong reliance on trusted verifiers (Sec 2.1), and incomplete exploration of complex adaptive or stealthy attacks (adaptive attack limited to single scenario). The paper would benefit from deeper discussion on system overhead, scalability trade-offs, and interactions with DP-level Byzantine defenses.

**Strengths:**

- Well-motivated identification of a critical security gap in PP decentralized training:
>> Clear distinction between DP and PP vulnerabilities, including cascading activation corruption (Sec. 2; Fig. 2).
>> Importance supported by large-scale LLM training requirements (Sec. 1).

- Comprehensive attack taxonomy and evaluation:
>> Seven activation/gradient attack types, including delay and invisible-noise attacks (Sec. 2.1).
>> Mixed-attack experiments better match real-world adversaries (Table 2).
>> Detection performance measured via precision, recall, F1, detection speed, and final validation loss (Tabs. 1–3).

- Theoretical support with assumptions stated explicitly:
>> Convergence guaranteed to a bounded neighborhood under undetected bounded perturbations (Sec. 3.2; Thm. 1).
>> Honest majority conditions formally quantified (Lemma 1).

- Large-scale distributed experiments and SWARM deployment:
>> Demonstrates end-to-end robustness in real decentralized environment (Fig. 4).
>> Ablation studies explore warm-up duration, collusion, and gradient delay impact (Fig. 3).

Clear articulation of limitations and threat boundaries:
>> Notes vulnerability to other ML attack types, e.g., backdoor, privacy attacks (Conclusion).
>> No assumption that >50% malicious workers can be tolerated (Lemma 1).

**Weaknesses:**

- Limited baseline comparison with Byzantine robust methods
>> Despite claims of incompatibility, no empirical or conceptual comparison with adapted DP defenses (Sec. 1), which would negatively impact the novelty claim and related practical justification.

- Insufficient quantification of false positives and their impact
>> Precision degradation noted in collusion experiments (Fig. 3b), but consequences such as reduced worker availability are not analyzed.
>> Validation loss alone may not reveal long-term optimization harm (Tabs. 2–3).

- Partial mathematical clarity and missing definitions
>> Distance measures in Eq. (2) was referenced in App. D, but no summary or concrete examples in main text.
>> Threshold adaptation method references App. Alg. 5 without high-level stability discussion (Sec. 3).
>> Overall, it feels that the lack of clarity weakens interpretability and reproducibility directly from the paper.

- Threat model assumptions not fully explored
>> Assumes no collusion between malicious workers yet collusion only tested up to 60% among attackers (Fig. 3b), without theoretical support.
>> First and last stage assumed to be honest (footnote 1), limiting generality for end-to-end secure decentralization.
>> Broader adversarial coordination strategies remain unexplored.

**Questions:**

In Section 3.2, Theorem 1 states that the convergence neighborhood size is proportional to ( \tau ). Could the authors provide a more precise relationship (e.g., a constant factor or specific bound) between ( \tau ) and the convergence error? This would help in tuning ( \tau ) for desired performance.

In Section 5.1, when integrating with SWARM, the paper mentions "32 trainer nodes with verification capability" but doesn't detail how the EMAs are synchronized across trainers. Could the authors clarify the synchronization mechanism and its impact on detection accuracy?

The paper uses multiple distance metrics (Appdix) but doesn't clarify how they are combined (e.g., majority vote, weighted average). Could the authors specify the combination strategy and its impact on detection performance? This would improve reproducibility and understanding of the method's robustness.

The ablation studies in Fig. 3 focus on warm-up steps, collusion, and delay, but do not explore the impact of varying EMA decay rates ((\beta_{h}) and (\beta_{g})). Could the authors provide additional experiments varying these hyperparameters to understand their effect on detection accuracy?

**Details Of Ethics Concerns:**

Nil

---

> ### Author Response · Authors · 2025-11-21
> **Response to Reviewer ux6h**
>
> We sincerely thank the reviewer for providing feedback on our work. We are encouraged that they found the setting of our work "underexplored yet relevant" with "clear formulation and empirical validation." We are also glad that they found our attack taxonomy "comprehensive" and their appreciation for providing theoretical guarantees. Below, we try to provide clarification on points made by the reviewer.
>
> -----------------------------------------------------------------------
>
>
> > **(W1) Despite claims of incompatibility, no empirical or conceptual comparison with adapted DP defenses (Sec. 1), which would negatively impact the novelty claim and related practical justification.**
>
> We would like to provide some clarification. During DP+PP based training (which is the most commonly used form of parallelization beyond a single device, please see our response to W1 from reviewer Byyi), three inter-device communications happen: forward pass for sending activations, backward pass for sending activation gradients, and parameter gradient sync/averaging between DP workers. Prior Byzantine-tolerant literature considers the case that the entire model is hosted on a single device, and only parameter gradients are shared. Hence, they try to defend against parameter gradient attacks using robust aggregation techniques such as Krum (Blanchard et al. 2017) and Bulyan (El-Mhamdi et al. 2018). In our work, however, we consider the pipeline parallel axis (attacks during forward and backward pass) and try to protect against them using SENTINEL. As seen, these two axes are complementary, and in this work we focus on the unexplored pipeline parallel axis. We discuss these differences in the introduction, Figure 2, and also Q0 in Appendix A.
>
> That being said, we provide detailed explanations on combining methods that prevent DP attacks with our method. **In particular, we combine our method with two robust aggregation methods and show that these two axes (DP and PP) are complementary in nature**, and our method can be added to any existing defense against gradient averaging attacks.
>
> | **Method**                 | **Pr. (%) ↑** | **Re. (%) ↑** | **F1 (%) ↑** | **Val. Loss (Attacked) ↓** | **Val. Loss (Vanilla) ↓** |
> |---------------------------|---------------|---------------|--------------|-----------------------------|-----------------------------|
> | **SENTINEL + Krum**       | 93.6          | 80.6          | 86.6         | 3.873                       | 3.855                       |
> | **SENTINEL + Bulyan**     | 85.3          | 80.6          | 82.9         | 3.883                       | 3.855                       |
>
> As shown in the table, the difference in validation loss between the attacked and vanilla settings is negligible (<0.5%), confirming that our method remains effective when combined with robust aggregation. For details on this experiment, please see our response to W5 from reviewer gMeg.
>
> -----------------------------------------------------------------------
>
> > **(W2) consequences such as reduced worker availability are not analyzed.**
>
> As pointed out in Appendix G, real-world use-cases of SENTINEL involve integration with SWARM which is different from fixed mesh data + pipeline parallel (see Figure 19 in Appendix G.3.1.) In SWARM (Ryabinin et al. 2023), workers get their data from trainer nodes which send activations and activation gradients between workers. **These trainer nodes use "stochastic wiring," where instead of sending their data to only a single worker for processing, they send it to whichever worker is not busy**. As seen, this is a different scenario compared to fixed mesh settings, and workers join and leave the SWARM as they wish. As such, there is no reason to believe that this would have an impact on stochastic eco-systems such as SWARM where multiple workers are available per stage. For more details, please see Appendix G. However, we acknowledge that if workers keep getting kicked and no new workers join the SWARM, this could impact throughput and the TPS. We have added this point to Q10 in Appendix A.

---

> > ### Author Response · Authors · 2025-11-21
> > **Response to Reviewer ux6h (cont.)**
> >
> > -----------------------------------------------------------------------
> >
> > > **(W3) Validation loss alone may not reveal long-term optimization harm (Tabs. 2–3).**
> >
> > We appreciate this comment, but there are two points to consider in relation to this:
> >
> > 1. We have a long-term run that extends towards Chinchilla optimality, and as can be seen from Figures 15 and 16 in Appendix F.2, our method is still matching the baseline even for longer runs.
> > 2. As **the LLM literature has pointed out in Mayilvahanan et al. 2025, all major baselines used to evaluate the LLMs not only have a positive correlation with lower validation loss, but their relationship is almost linear**. Based on these findings (see Figures 1 and 2 in Mayilvahanan et al. 2025), we can safely conclude there is no reason to believe that there are long-term harms beyond the validation loss, as the validation loss has a linear relationship with all major baselines.
> >
> > -----------------------------------------------------------------------
> >
> >
> > > **(W4) Distance measures in Eq. (2) was referenced in App. D, but no summary or concrete examples in main text. Threshold adaptation method references App. Alg. 5 without high-level stability discussion (Sec. 3). Overall, it feels that the lack of clarity weakens interpretability and reproducibility directly from the paper.**
> >
> > We really appreciate this comment. We have added additional discussions and pointers in the main paper (see Section 3, lines 233-253) to increase the clarity. Please let us know if any further improvements are needed in that front.
> >
> > -----------------------------------------------------------------------
> >
> >
> > >  **(W5) First and last stage assumed to be honest (footnote 1), limiting generality for end-to-end secure decentralization.**
> >
> > As noted in the paper, we made these assumptions as there are no guarantees that the first and last layer would not tamper with the original data and the labels. For now, we assume that those two layers are operated either centrally or by workers that have gained enough credibility through the SWARM network to assume that the first and last layers are honest. Economically, this still makes sense as those layers constitute only a fraction of the model, especially considering LLMs that may have many intermediate layers. For more discussion on this point, please see Q4 in Appendix A.
> >
> > -----------------------------------------------------------------------
> >
> >
> > > **(W6) Broader adversarial coordination strategies remain unexplored.**
> >
> > **We have considered coordinated attacks as the simplest form of collusion** (see lines 436-439 and Figure 3.2 in Section 5). **We have also considered adaptive attackers that are aware of the EMA detection strategy** (see lines 461-478 and Table 6 in Section 5). We believe broader adversarial intent and coordination requires substantial research which is interesting but beyond the scope of our current work. We have written our thoughts on this in Appendix B of the paper, which we highlighted for your convenience.
> >
> > -----------------------------------------------------------------------
> >
> > > **(Q1) Could the authors provide a more precise relationship (e.g., a constant factor or specific bound) between ( \tau ) and the convergence error? This would help in tuning ( \tau ) for desired performance.**
> >
> > We would like to clarify that in non-convex convergence guarantees, we usually measure the bound not in terms of actual error distance but in the form of an upper-bound on the gradient magnitude ( $\frac{1}{T} \sum_{t=0}^{T} \mathbb{E}\|\|\nabla f(x_t)\|\|^2$), for instance, please see Koloskova et al. 2024. We have provided the full proof alongside the relationship between $\tau$ and this upper-bound in Equation (55) in Appendix E.1.4. Please let us know if further clarification is needed.

---

> ### Author Response · Authors · 2025-11-21
> **Response to Reviewer ux6h (cont.)**
>
> -----------------------------------------------------------------------
>
> > **(Q2) In Section 5.1, when integrating with SWARM, the paper mentions "32 trainer nodes with verification capability" but doesn't detail how the EMAs are synchronized across trainers. Could the authors clarify the synchronization mechanism and its impact on detection accuracy?**
>
> We thank the reviewer for this important question. Due to space limit, we deferred the full technical details of our SENTINEL integration in SWARM to Appendix G. As pointed out in lines 501-507 of the main paper and also Algorithm 7 in Appendix G.3, when integrating SENTINEL with SWARM there is no need for trainers to synchronize their EMAs. This is because each trainer would send their job to various workers due to stochastic wiring, and thus, different trainers should have similar EMAs despite not synchronising them.
>
> In our SWARM experiment, EMA synchronization happens **only once at the end of the warm-up** period where trainers average their EMAs. **Except for this single synchronization, trainers use their own EMA throughout training**. As we demonstrate in Figure 21 in Appendix G.4.2, the standard deviation of EMAs between different trainers is small, echoing our point about them having similar EMAs despite not being constantly synced.
>
> Please let us know if further clarification is needed.
>
> -----------------------------------------------------------------------
>
> > **(Q3) The paper uses multiple distance metrics (Appdix) but doesn't clarify how they are combined (e.g., majority vote, weighted average). Could the authors specify the combination strategy and its impact on detection performance? This would improve reproducibility and understanding of the method's robustness.**
>
> As outlined in line 249-251 of the paper in Section 3 and also lines 5-6 of Algorithm 2 in Appendix D, we do not combine these metrics. **Rather, we consider each of them disjointly and a worker gets flagged if "any" of these metrics fall outside the threshold boundaries.** We have further clarified this point in the paper. We also have an ablation study on the impact of each deviation metric which we refer to in lines 2886-2893 and Table 18 of Appendix F.3.
>
> -----------------------------------------------------------------------
>
> > **(Q4) The ablation studies in Fig. 3 focus on warm-up steps, collusion, and delay, but do not explore the impact of varying EMA decay rates ((\beta_{h}) and (\beta_{g})). Could the authors provide additional experiments varying these hyperparameters to understand their effect on detection accuracy?**
>
> Initially, we provided an experiment where we removed the EMA by setting $\beta=0$ in Appendix G.3. **We have extended this experiment to use multiple different $\beta$ values as EMA decay rates.** As shown in the table below, modifying the decay rate has minimal impact on the performance. We have added these results to the revised version which you can find in Appendix F.3.
>
> | **Reference Point**                           | **Pr. (%) ↑** | **Re. (%) ↑** | **F1 (%) ↑** | **Val. Loss ↓** |
> |-----------------------------------------------|---------------|---------------|--------------|------------------|
> | **Average ($\beta_h = \beta_g =0$)**                     | 23.08         | 100.0         | 37.5         | 6.248            |
> | **SENTINEL ($\beta_h=0.9,~\beta_g=0.8$)**             | **100.0**     | **100.0**     | **100.0**    | **3.826**        |
> | **SENTINEL ($\beta_h=\beta_g=0.60$)**                 | 94.7          | **100.0**     | 97.3         | 3.894            |
> | **SENTINEL ($\beta_h=\beta_g=0.99$)**                 | 90.0          | **100.0**     | 94.7         | 3.875            |
>
> -----------------------------------------------------------------------
>
> We have incorporated the suggested ablation studies and clarifications into the revised manuscript. We hope these additions effectively resolve your questions regarding the method's robustness, and would appreciate it if you could reassess our work in light of these clarifications.

---

> > ### Author Response · Authors · 2025-11-21
> > **References**
> >
> > Blanchard et al. "Machine Learning with Adversaries: Byzantine Tolerant Gradient Descent." *NeurIPS*. 2017.
> >
> > El-Mhamdi et al. "The Hidden Vulnerability of Distributed Learning in Byzantium." *ICML*. 2018.
> >
> > Ryabinin et al. "Swarm parallelism: Training large models can be surprisingly communication-efficient." *ICML*. 2023.
> >
> > Koloskova et al. "On convergence of incremental gradient for non-convex smooth functions." *ICML*. 2024.
> >
> > Mayilvahanan et al. "LLMs on the Line: Data Determines Loss-to-Loss Scaling Laws." *ICML*. 2025.

---

> > > ### Author Response · Authors · 2025-11-27
> > > **Rebuttal Follow-up**
> > >
> > > Dear Reviewer ux6h,
> > >
> > > We are writing to gently follow up on our rebuttal response.
> > >
> > > Following your suggestions, we have added the requested ablation studies on EMA decay rates ($\beta$) and provided clearer definitions for the distance metrics used. We believe these additions significantly improve the reproducibility and clarity of the work. We would value your feedback on these revisions before the discussion period closes, and would appreciate it if you could reassess our work in light of these clarifications. We remain available to address any unresolved concerns.

---

> > > > ### Comment · Reviewer_ux6h · 2025-11-28
> > > >
> > > > Thank you for your detailed rebuttal. Overall, I consider the paper is well-motivated with comprehensive experiments that reasonably support the claims. While the work does not dramatically advance the field beyond its specific setting (I might be biased), it makes a positive effort about Byzantine resilience in pipeline-parallel training. I would maintain my recommendation of “marginally above the acceptance threshold.”

---

### Official Review · Reviewer_bjpd · 2025-10-30

**Soundness:** 2
**Presentation:** 2
**Contribution:** 2
**Rating:** 4
**Confidence:** 3

**Summary:**

The paper proposes SENTINEL, a lightweight verification mechanism for ensuring computational integrity in pipeline-parallel (PP) decentralized training, where traditional Byzantine-robust aggregation methods are inapplicable. SENTINEL introduces verifier nodes that monitor inter-stage activations and gradients using Exponential Moving Averages (EMAs) and IQR-based adaptive thresholds to detect anomalies caused by malicious workers. Experiments with Llama-3-0.6B and 1.2B models on decentralized frameworks (e.g., SWARM) show high (>90%) F1 scores in detecting various attack types with minimal overhead.

**Strengths:**

- Novel threat model: Addresses pipeline-parallel decentralized training security — an underexplored but increasingly relevant setting.

- Lightweight design: Verification via EMAs and statistical tests avoids costly redundancy or gradient aggregation.

- Comprehensive evaluation: Covers numerous attack types (activation, gradient, mixed, and adaptive attacks) across large-scale distributed setups.

**Weaknesses:**

- Trusted verifier assumption: SENTINEL depends critically on verifier nodes being honest and reliable. If a verifier node is compromised, it can both hide attacks and falsely flag benign workers, effectively collapsing the system’s security. The paper does not discuss mechanisms such as rotating verifiers, distributed verification, or cryptographic attestation to mitigate this.

- Incomplete threat model: The approach targets activation/gradient corruption but ignores broader adversarial behaviors such as data poisoning, backdoor insertion, or sybil collusion across multiple stages. These are common in decentralized systems and could bypass SENTINEL entirely.

- Limited evaluation scope: Experiments are conducted on medium-scale Llama-3 models (≤1.2B parameters) in simulated decentralized settings. It remains unclear how SENTINEL scales to truly large (>10B) models, heterogeneous networks, or high-latency cross-institutional environments where EMA synchronization could become costly.

- Parameter sensitivity and calibration cost: The verification relies on several empirically chosen hyperparameters (EMA decay rates, window size, IQR threshold k). These may require manual tuning per dataset and model. There is little analysis of robustness to these choices or automated adaptation beyond the initial “warm-up” period.

- False positives and training stability: While the paper reports high detection rates, it gives limited insight into false positive rates and the resulting training slowdowns or disruptions. Misidentification of honest nodes could degrade throughput or cause partial divergence in long training runs.

- Assumption-heavy theoretical guarantees: The proofs rely on simplified assumptions (e.g., independent random worker assignment, fixed detection thresholds). These are difficult to ensure in real decentralized networks, where collusion or heterogeneous bandwidth can break such guarantees.

**Questions:**

- How would the system behave if one or more verifier nodes are compromised or unavailable?

- Could the verification function be decentralized (e.g., through rotating verifiers or majority voting among neighboring nodes)?

- What is the quantitative computation and communication overhead of SENTINEL relative to redundancy-based baselines?

- How sensitive is performance to hyperparameter tuning (βₕ, βg, k, warm-up length)?

- Can the same framework detect semantic or backdoor-style attacks where activations remain statistically normal but maliciously biased?

- How generalizable is SENTINEL to other architectures (e.g., MoE models or non-transformer networks)?

---

> ### Author Response · Authors · 2025-11-21
> **Response to Reviewer bjpd**
>
> We sincerely thank the reviewer for providing feedback on our work. We are glad that they appreciate the "novelty of threat model" considered in this work, our "lightweight verification mechanism", and the "comprehensive evaluation" of the various attack types in our work. Below, we try to address their concerns and answer their questions.
>
> -----------------------------------------------------------------------
>
> >  **(W1) If a verifier node is compromised, it can both hide attacks and falsely flag benign workers, effectively collapsing the system’s security. The paper does not discuss mechanisms such as rotating verifiers, distributed verification, or cryptographic attestation to mitigate this.**
>
> We thank the reviewer for this interesting question. As pointed out in the paper, our threat model assumes that we do have "trusted verifier nodes." We emphasize that this assumption does not violate efficiency: due to the lightweight nature of verification, our verifier nodes are CPU nodes that can easily run the verification process. As pointed out in Appendix G, our verification mechanism sits on top of the trainer nodes within the SWARM eco-system, and the cost of running these verifiers by trusted entities is negligible.
>
> Concretely, for the 128 GPU node SWARM experiment outlined in the paper, we have 32 trainer nodes that are handling verification as part of their duties within the SWARM. While for the GPU nodes we had to utilize expensive nodes (8 `g5.2xlarge` instances for a price of $1.212/hr, 8 `g5.4xlarge` for a price of $1.624/hr, and 102 `g4dn.2xlarge` for a price of $0.752/hr, equalling to $99.39/hr), we can deploy all 32 trainer nodes using only 4 `c6a.8xlarge` instances with a total cost of $4.88/hr. As can be seen, the cost of running trusted trainer/verifier nodes is less than 5% of the total cost of our decentralized training, thus, it would make sense financially to operate and manage them.
>
> On a final note, we agree with the reviewer's point that there is a potential research direction here: *how to effectively delegate the verification process itself to untrusted entities.* While this is an interesting question, we believe its solution involves substantial research which is beyond the scope of the current work. We have discussed this point already in the paper in Q3 in Appendix A which we highlighted for your convenience.
>
> -----------------------------------------------------------------------
>
> > **(W2) The approach targets activation/gradient corruption but ignores broader adversarial behaviors such as data poisoning, backdoor insertion, or sybil collusion across multiple stages. These are common in decentralized systems and could bypass SENTINEL entirely.**
>
> As outlined in the paper, here we focus on training-disruption attacks, i.e., adversaries whose aim is to halt training. Indeed, considering all various forms of attacks that could arise is very challenging in a single work, and we have not claimed that SENTINEL is robust against all forms of adversarial threats. Historically, the adversarial ML literature has treated data poisoning, backdoor attacks, privacy attacks, training-interruption, etc. as distinct sub-fields due to their unique challenges. **While we agree a unified defense is the ultimate goal, tackling all threat vectors simultaneously is often infeasible in a single study**.
>
> We have pointed out in the paper that our threat model is targeting training-interruption attacks, but we have also initiated the discussions on other forms of adversaries that could arise within the decentralized training eco-system. Please see the limitation on lines 534-539 as well as a detailed discussion on various forms of adversaries that are not considered in this paper in Appendix B (lines 990-1025). **We share the same sentiment with the reviewer that this area is underexplored compared to the rest of adversarial ML domains, and we hope our paper could initiate such research directions within the ML security community.**
>
> Finally, would like to emphasize that we considered simultaneous attacking as a form of collusion in our experiments, which you can find in Section 5 under lines 436-439 which we have highlighted for your convenience.

---

> > ### Author Response · Authors · 2025-11-21
> > **Response to Reviewer bjpd (cont.)**
> >
> > -----------------------------------------------------------------------
> >
> > > **(W3) Experiments are conducted on medium-scale Llama-3 models (≤1.2B parameters) in simulated decentralized settings. It remains unclear how SENTINEL scales to truly large (>10B) models, heterogeneous networks, or high-latency cross-institutional environments where EMA synchronization could become costly.**
> >
> > There are several aspects to this question which we hope to clarify below:
> > 1. In our original submission we considered NanoGPT models as well, please see Appendix F.2.
> > 2. Due to reviewer's request on running experiments using MoE architecture, we have included Llama-4-0.4B and DeepSeekV3-1B which we defer to Q6 below.
> > 3. We would like to emphasize that training models with >10B parameters require computational resources (e.g., >24 H100 GPUs) that exceed the capacity of most academic laboratories. We believe our experiments on models up to 4B parameters, combined with the theoretical convergence guarantees, provide sufficient evidence of scalability without requiring industrial-scale infrastructure.
> > 4. **To demonstrate this scalability within our available means, we run SENTINEL against mixed activation attacks for a Llama-3 model with 4B parameters.** We use 8 H100 GPUs to deploy the 176 workers required for training model of this size with a small batch size of 12. We run our training until step 5k. We use a 3:5 ratio between honest to malicious workers at each stage, and randomly assign activation attacks to each worker. Each attacker coordinates with 20% of all attackers to send their malicious signals at the same time, where the attack times are decided at random. As seen in the table below, our method still successfully generalizes to this setting and prevents divergence. We have included the detection results alongside the training and validation loss figures to the experimental results in Section 5 (lines 451-459) and Appendix E.2 (lines 2634-2638).
> > | **Architecture**          | **Pr. (%) ↑** | **Re. (%) ↑** | **F1 (%) ↑** | **Val. Loss (SENTINEL) ↓** | **Val. Loss (Vanilla) ↓** |
> > |---------------------------|---------------|---------------|--------------|-----------------------------|-----------------------------|
> > | **Llama-3-4B**            | 94.9          | 68.6          | 79.6         | 3.714                       | 3.668                       |
> >
> >    **As can be seen, SENTINEL can successfully be applied to large-scale models and ensure convergence similar to vanilla training (in our experiment, the difference is <0.15%).**
> >
> > 5. Finally, we would like to respectfully point out that **we have considered the real-world use-case of decentralized training across heterogeneous networks.** For our SWARM experiments, the details of which have been outlined in Appendix G, each worker is located in a separate instance within AWS with no direct links. **For SWARM, the EMA does not need to be synchronized except once at the end of the warmup period** (see our response to Q2 from reviewer ux6h). As pointed out there and in the paper lines 501-507, each trainer/verifier uses their own version of EMA, and since SWARM uses stochastic routing, these EMAs remain close to each other. Please see the standard deviation between all 32 EMA replicas for activation and gradient EMA in Figure 21 in Appendix G. As seen, the variance for both cases are small. As such, we believe **there is no reason for our method to be affected by network conditions since no ongoing, global EMA synchronisation is needed in the real-world**. Please read our full implementation details in Appendix G and let us know if any further clarification is needed.

---

> ### Author Response · Authors · 2025-11-21
> **Response to Reviewer bjpd (cont.)**
>
> -----------------------------------------------------------------------
>
> >  **(W4) The verification relies on several empirically chosen hyperparameters (EMA decay rates, window size, IQR threshold k). These may require manual tuning per dataset and model. There is little analysis of robustness to these choices or automated adaptation beyond the initial “warm-up” period.**
>
> We would like to clarify a few points here.
> 1. First, we emphasize that **we have an extensive set of ablation studies on the impact of warm-up period, attacker collusion, delay, EMA, different distance metrics, and randomness in our experiments**. For a complete set of results please see Section 5 (lines 429-460) and Appendix F.3.
> 2. To complement our ablation study on the impact of a wider variety of EMA decay rates, we extend our ablation study in Table 15 to also include two additional betas, which would ablate the decay rate over $\beta \in \{ 0.0, 0.6, 0.8, 0.99 \}$. As seen below, apart from $\beta = 0.0$ which means instantaneous averaging and it is not working, SENTINEL remains stable across various choices of $\beta$.
> | **Reference Point**                           | **Pr. (%) ↑** | **Re. (%) ↑** | **F1 (%) ↑** | **Val. Loss ↓** |
> |-----------------------------------------------|---------------|---------------|--------------|------------------|
> | **Average ($\beta_h = \beta_g =0$)**                     | 23.08         | 100.0         | 37.5         | 6.248            |
> | **SENTINEL ($\beta_h=0.9,~\beta_g=0.8$)**             | **100.0**     | **100.0**     | **100.0**    | **3.826**        |
> | **SENTINEL ($\beta_h=\beta_g=0.60$)**                 | 94.7          | **100.0**     | 97.3         | 3.894            |
> | **SENTINEL ($\beta_h=\beta_g=0.99$)**                 | 90.0          | **100.0**     | 94.7         | 3.875            |
>
> 3. Please also note, for most of our experiments, we did not perform any extensive hyper-parameter selection and most of our hyper-parameters were used across different models/datasets without any changes. For example, **for the MoE experiments that were requested by the reviewer, we used the same set of hyper-parameters that we used for Llama-3-0.6B experiments. This highlights the transferability of our hyper-parameters across settings.**
> 4. Finally, we would like to clarify that the IQR threshold $k$ is being adaptively set. As mentioned in our methodology in Section 3.1 (lines 261-265 which we highlighted for your convenience), the IQR threshold is being set adaptively using the last 50 deviation history (which we used for all experiments without tuning). For more details on this automatic adaptation, please see Algorithm 5 in Appendix D.
>
> -----------------------------------------------------------------------
>
> > **(W5) While the paper reports high detection rates, it gives limited insight into false positive rates and the resulting training slowdowns or disruptions. Misidentification of honest nodes could degrade throughput or cause partial divergence in long training runs.**
>
> We thank the reviewer for this insightful question. We would like to emphasize that in real-world eco-systems such as SWARM (Ryabinin et al. 2023), we are not using a fixed mesh anymore. Workers join and leave the SWARM as they wish, and within each pipeline stage, there are numerous workers that are serving that stage. This combined with stochastic routing in SWARM (please see the details in Appendix G.1 and Figure 19 in G.3.1) ensures that workers would get close to their maximum utilization (please see Figure 1, Right from Ryabinin et al. 2023.) Thus, in real-world systems, we have an abundance of workers serving each stage where kicking workers out would not necessarily degrade throughput. However, we acknowledge that if workers keep getting kicked and no new workers join the SWARM, this could impact throughput and the TPS. We have added this point to Q10 in Appendix A. Please let us know if further clarification is needed.

---

> > ### Author Response · Authors · 2025-11-21
> > **Response to Reviewer bjpd (cont.)**
> >
> > -----------------------------------------------------------------------
> >
> > > **(W6) The proofs rely on simplified assumptions (e.g., independent random worker assignment, fixed detection thresholds). These are difficult to ensure in real decentralized networks, where collusion or heterogeneous bandwidth can break such guarantees.**
> >
> > Similar to all previous work within the Byzantine-tolerant literature (e.g., Gorbunov et al. 2023 and Malinovsky et al. 2024), we had to make relaxed assumptions to be able to provide a rigorous proof. Otherwise, representing the algorithm in its totality might have hindered the convergence theorem entirely.
> >
> > Additionally, we would like to emphasize that:
> > 1. In our experiments, we considered simultaneous coordinated attacking as a form of collusion. Please see our results in Section 5 (lines 436-439 and Figure 3.2).
> > 2. There is no evidence that network heterogeneity would impact our work. As pointed out to the reviewer's point above, in real-world scenarios there are no ongoing EMA synchronization required, and as such, there is no reason to believe that this would impact any of our results. Please see lines 501-507 in Section 5 and Appendix G.
> > 3. Finally, the "independent worker assignment" is indeed realistic. In decentralized training eco-systems such as SWARM (Ryabinin et al. 2023), workers usually get assigned to different stages randomly. This assumption in Lemma 1 is reflective of this behavior and aims to connect the honest majority in the initial pool of workers to the honest majority at each stage and how they would affect each other after "random worker assignment." However, *we do agree that other types of secure worker assignment might be possible which would be an interesting future direction.*
> >
> > -----------------------------------------------------------------------
> >
> > > **(Q1) How would the system behave if one or more verifier nodes are compromised or unavailable?**
> >
> > For compromised verifier nodes, please see our response to W1 earlier. On the availability of verifier nodes, that would not be an issue in decentralized systems built on SWARM parallelism. The reason is that our verification operation happens within trainer nodes that are overseeing batches of data end-to-end (please see Algorithm 7 from Appendix G). Each trainer is keeping their own local version of EMA, and in case they drop, no information from them is needed for other trainers to continue training (lines 501-507). For more information on trainer nodes, please see Algorithm 6 in Appendix G and the SWARM repository.
> >
> > -----------------------------------------------------------------------
> >
> > > **(Q2) Could the verification function be decentralized (e.g., through rotating verifiers or majority voting among neighboring nodes)?**
> >
> > As pointed out in W1, the cost of running verifier nodes centrally is a tiny fraction compared to compute nodes. Thus, economically it would not be hard to operate these. However, we agree that delegating verification to untrusted nodes is a promising research direction, involving techniques such as optimistic verification protocols and the use of TEEs to run verifiers securely, similar to approaches explored in TruBit (Teutsch and Reitwießner 2024). We have alluded to this in Q3 in our paper as well.
> >
> > -----------------------------------------------------------------------
> >
> > > **(Q3) What is the quantitative computation and communication overhead of SENTINEL relative to redundancy-based baselines?**
> >
> > The only redundancy-based baseline that considers decentralized training is the work of Lu et al. 2024. We have provided an extensive discussion and comparison with this work in Q9 in Appendix A. To summarize, **the work of Lu et al. 2024 creates a duplicate for each worker within the network and tries to replicate their work for verification. This would halve the network throughput**; e.g., if we have 320 compute nodes, we need to effectively use 160 of them for verification which would hamper training speed by half.
> >
> > In terms of communication overhead, as detailed in Appendix G, we use the information received by trainer nodes for verification. We do **not** need any additional costly communication between trainers as each trainer keeps their own replica of EMA (see Algorithm 7 in Appendix G). Please let us know if any further clarifications on this point is needed.
> >
> > -----------------------------------------------------------------------
> >
> > > **(Q4) How sensitive is performance to hyperparameter tuning (βₕ, βg, k, warm-up length)?**
> >
> > We have provided an ablation study on the impact of the decay rate in Appendix F.3 which we have extended after the reviewer's request. Please see the results in our response to W4.
> >
> > Also, as noted before, the IQR threshold $k$ is being set adaptively. Please see Algorithm 5 of Appendix D. Also, the effect of setting this threshold adaptively can be seen in Figure 12 of Appendix F.2.

---

> > > ### Author Response · Authors · 2025-11-21
> > > **Response to Reviewer bjpd (cont.)**
> > >
> > > -----------------------------------------------------------------------
> > >
> > > > **(Q5) Can the same framework detect semantic or backdoor-style attacks where activations remain statistically normal but maliciously biased?**
> > >
> > > As pointed out in our limitation discussion as well as Appendix B, backdoor attacks are indeed an interesting research direction. Unfortunately, *at the moment there is no existing backdoor attack literature on pipeline parallel settings where they show the effectiveness of such attacks in this environment.* Since only the first layer has access to the data and only the last layer has access to the labels, it indeed is challenging to delegate those layers to untrusted workers. However, *it remains an interesting research question to see if by only having access to intermediate layers a worker can successfully implant backdoors in the model.* This requires substantial non-trivial research which we leave for future work.
> > >
> > > -----------------------------------------------------------------------
> > >
> > > > **(Q6) How generalizable is SENTINEL to other architectures (e.g., MoE models or non-transformer networks)?**
> > >
> > > By far, *transformer models are the most common use-case of decentralized training* (Ryabinin et al. 2023 and Lu et al. 2024). This is the reason we considered LLMs as our primary application throughout the paper. To the best of our knowledge, there is limited work on decentralized training in domains other than text as LLMs are by far the largest size models which would benefit significantly from decentralized training.
> > >
> > > That being said, **we experiment with SENTINEL on alternative MoE architectures**. To this end, we consider three MoE models: Llama-4 with 400M parameters and DeepSeekV3 with 1B parameters. We then train these models using the Fineweb-EDU dataset against mixed activation attacks. **For hyper-parameters, we use the same set of hyper-parameters as our Llama-3-0.6B parameters to highlight the resilience of our proposed method to different settings (in response to W4 and Q4).**
> > >
> > > The results are given in the table below. As seen, **SENTINEL can be easily extended to this alternative architecture type under different families and train models successfully without divergence**. We have added these results to our ablation studies in the main paper (lines 443-451) and Appendix F.3.
> > >
> > > | **Architecture**          | **Pr. (%) ↑** | **Re. (%) ↑** | **F1 (%) ↑** | **Val. Loss (Sentinel) ↓** | **Val. Loss (Vanilla) ↓** |
> > > |---------------------------|---------------|---------------|--------------|-----------------------------|-----------------------------|
> > > | **Llama-4-0.4B**          | 73.5          | 92.3          | 81.8         | 3.617                       | 3.628                       |
> > > | **DeepSeek-V3-1B**        | 94.6          | 97.2          | 95.9         | 3.421                       | 3.393                       |
> > >
> > > -----------------------------------------------------------------------
> > >
> > > We trust that the new experiments on Llama-3-4B and MoE architectures resolve the concerns regarding scalability. We would appreciate a re-evaluation of our work based on these new results, and happy to address any further questions.

---

> > > > ### Author Response · Authors · 2025-11-21
> > > > **References**
> > > >
> > > > Teutsch and Reitwießner. "A scalable verification solution for blockchains." _Aspects of Computation and Automata Theory with Applications_. 2024. 377-424.
> > > >
> > > > Ryabinin et al. "Swarm parallelism: Training large models can be surprisingly communication-efficient." *ICML*. 2023.
> > > >
> > > > Gorbunov et al. "Secure distributed training at scale." *ICML*, 2022.
> > > >
> > > > Malinovsky et al. "Byzantine robustness and partial participation can be achieved at once: Just clip gradient differences." *NeurIPS*. 2024.
> > > >
> > > > Lu et al. "Position: Exploring the robustness of pipeline-parallelism-based decentralized training." *ICML*. 2024.

---

> > > > > ### Author Response · Authors · 2025-11-27
> > > > > **Rebuttal Follow-up**
> > > > >
> > > > > Dear Reviewer bjpd,
> > > > >
> > > > > We are writing to kindly follow up on our rebuttal response.
> > > > >
> > > > > Per your request, we have conducted and included new large-scale experiments on Llama-3-4B, as well as experiments on MoE architectures (Llama-4 and DeepSeek-V3). We believe these results address your concerns regarding the scalability and generalization of SENTINEL. We would greatly value your thoughts on these new results before the discussion period ends. We would appreciate a re-evaluation of our work based on these new results, and happy to address any further questions.

---

### Official Review · Reviewer_Byyi · 2025-11-01

**Soundness:** 3
**Presentation:** 3
**Contribution:** 3
**Rating:** 6
**Confidence:** 4

**Summary:**

The paper proposed SENTINEL, a verification mechanism for PP training without computation duplication. They provide theoretical convergence guarantees for this new setting that recovers classical convergence rates when relaxed to standard training. Experiments demonstrate successful training of billion-parameter LLMs across untrusted distributed environments with hundreds of workers while maintaining model convergence and performance.

**Strengths:**

1. The paper is claimed as the first comprehensive study of vulnerabilities unique to decentralized training with hybrid data–pipeline parallelism, and introduce a suite of training-interruption attacks that serve as benchmarks for evaluating the security of future systems.

2. The theoretical analysis demonstrates that undetected malicious workers have a negligible impact on the convergence properties.

3. The authors integrate our method with SWARM parallelism to demonstrate its remarkable versatility in real-world decentralized training ecosystems.

**Weaknesses:**

1. The authors claimed that "the paper considered the first comprehensive exploration of secure and verifiable PP decentralized training
by identifying". As for me, In this setting we can see that we need to train billionparameter LLMs through internet-scale communication among distributed nodes.  The paper does not discuss all possible the topology of the inter-connected distributed notes.

2.  Due the issue listed above, the advantage of using decentralized training with hybrid data–pipeline parallelism is largely weaken. And the proposed EMAs is not persuasive. Please explain.

3. The generality of "Data and Pipeline Parallel Threat Model" is not justified. I am not sure why it is typical in real practice.

4. Plus, the advantage of combing your scheme with SWARM needs to be discussed in many different scenarios.

**Questions:**

Please see in the weakness parts.

---

> ### Author Response · Authors · 2025-11-21
> **Response to Reviewer Byyi**
>
> We would like to thank the reviewer for their assessment of our work. We are encouraged that they value our work due to being "the first comprehensive study of vulnerabilities unique to decentralized training" and its "remarkable versatility" for its integration into SWARM. We hope to clarify their concerns around our work below.
>
> -----------------------------------------------------------------------
>
> > **(W1) As for me, In this setting we can see that we need to train billionparameter LLMs through internet-scale communication among distributed nodes. The paper does not discuss all possible the topology of the inter-connected distributed notes.**
>
> We would like to clarify that the focus of our work is not on "feasibility of decentralized training." Rather, **we considered the DP + PP as the most commonly used paradigm in decentralized training** (see Huang et al. 2019, Li et al. 2020, Yuan et al. 2020, and Ryabinin et al. 2023) to scale model training beyond a single device. Then, we tried to establish an interesting yet underexplored research direction by tapping into the security issues inherent within such widely used eco-systems. To this end, we not only consider the case of fixed mesh DP+PP, but we also generalize our methodology into SWARM which uses stochastic routing and DHT-based work allocation between workers. SWARM (Ryabinin et al. 2023) and its parent library, Hivemind (Ryabinin and Gusev 2020), are the most commonly used tools in decentralized training as of today, with which we have integrated SENTINEL. For more information on this integration and stochastic routing, please see Appendix G.
>
> On the feasibility of decentralized training, there are successful traces of training billion parameter LLMs within both research (please see Petals (Borzunov et al. 2023), SWARM (Ryabinin et al. 2023), TaskLets (Yuan et al. 2021)) and also industry (Gensyn Testnet (Gensyn Team 2025) and Prime Intellect (PrimeIntellect Research Team 2025)). These examples demonstrate the feasibility of training billion-parameter LLMs over internet-grade connections. In this paper, we aim to shed light on some of the security aspects of such eco-systems with the hope to foster an ongoing discussion among researchers around this important topic.
>
> - **Petals** (Borzunov et al. 2023): Actively running 100B+ parameter models across volunteer networks
> - **SWARM** (Ryabinin et al. 2023): Demonstrated billion-parameter training with realistic bandwidth constraints
> - **Gensyn testnet** (Gensyn Team 2025): Live infrastructure with operational "RL Swarm nodes" for collaborative LLM training
> - **INTELLECT-2 Release** (PrimeIntellect Research Team 2025): The first 32B parameter model trained through globally distributed reinforcement learning
> - **SWARM for Medical Applications** (Warnat-Herresthal et al. 2021): Published in Nature showing successful decentralized learning across medical institutions
>
> -----------------------------------------------------------------------
>
> > **(W2) Due the issue listed above, the advantage of using decentralized training with hybrid data–pipeline parallelism is largely weaken. And the proposed EMAs is not persuasive. Please explain.**
>
> > **(W3) The generality of "Data and Pipeline Parallel Threat Model" is not justified. I am not sure why it is typical in real practice.**
>
> As stated above, SWARM parallelism is the most commonly used form of parallelism used within decentralized training eco-systems (Ryabinin et al. 2023). In this paper, we first consider the case of fixed mesh DP+PP, but also extend our method into SWARM and its primitives. SWARM, which has been successfully used to train billion parameter LLMs over internet-grade connections (Ryabinin et al. 2023), is a generalization of fixed mesh DP+PP but uses stochastic routing between pipeline stages for work allocation. We have added a visual representation of stochastic routing and how this is a generalization of fixed mesh data + pipeline parallel (see Figure 19). For a detailed, step-by-step guide on SENTINEL integration with SWARM, please see Appendix G.
>
> We hope that our response has mitigated the reviewer's concerns. If further explanation on this point is needed, please let us know.

---

> > ### Author Response · Authors · 2025-11-21
> > **Response to Reviewer Byyi (cont.)**
> >
> > -----------------------------------------------------------------------
> >
> > > **(W4) Plus, the advantage of combing your scheme with SWARM needs to be discussed in many different scenarios.**
> >
> > As discussed above, we have extensively elaborated on the integration of SENTINEL into SWARM and how our approach fits perfectly within its eco-system. Details of this can be found in Appendix G. Please note that these experiments are quite expensive (>$100/hr), hence, we had to limit our tests to only a few. Please see our response to W1 from reviewer bjpd for a cost breakdown. Please let us know if the reviewer has any specific scenarios in mind, we will try to clarify further.
> >
> > -----------------------------------------------------------------------
> >
> > We hope the additional details regarding the practical integration of our work with SWARM have alleviated your concerns. We would appreciate it if you could reassess our work in light of these clarifications.

---

> > > ### Author Response · Authors · 2025-11-21
> > > **References**
> > >
> > > Huang et al. "Gpipe: Efficient training of giant neural networks using pipeline parallelism." _NeurIPS_. 2019.
> > >
> > > Li et al. "Pytorch distributed: Experiences on accelerating data parallel training." _VLDB_. 2020.
> > >
> > > Ryabinin and Gusev. "Towards crowdsourced training of large neural networks using decentralized mixture-of-experts." *NeurIPS*. 2020.
> > >
> > > Warnat-Herresthal et al. "Swarm learning for decentralized and confidential clinical machine learning." *Nature* 594.7862 (2021): 265-270.
> > >
> > > Yuan et al. "Decentralized training of foundation models in heterogeneous environments." *NeurIPS*. 2021.
> > >
> > > Diskin et al. "Distributed deep learning in open collaborations." *NeurIPS*. 2021.
> > >
> > > Borzunov et al. "Petals: Collaborative inference and fine-tuning of large models." *ACL*. 2023.
> > >
> > > Ryabinin et al. "Swarm parallelism: Training large models can be surprisingly communication-efficient." *ICML*. 2023.
> > >
> > > Gensyn Team. Gensyn Protocol Documentation: Machine Learning Compute Network. 2025.
> > >
> > > PrimeIntellect Research Team. INTELLECT-2 Release: The First 32B Parameter Model Trained Through Globally Distributed Reinforcement Learning. PrimeIntellect Blog. 2025.

---

> > > > ### Author Response · Authors · 2025-11-27
> > > > **Rebuttal Follow-up**
> > > >
> > > > Dear Reviewer Byyi,
> > > >
> > > > We are writing to gently follow up on our rebuttal response.
> > > >
> > > > We have provided additional details regarding the SWARM integration and stochastic topology to clarify the practicality of our setting. We hope these details address your concerns regarding the practical feasibility of the approach. We would appreciate it if you could reassess our work in light of these clarifications, and we look forward to your thoughts.

---

### Official Review · Reviewer_gMeg · 2025-11-01

**Soundness:** 2
**Presentation:** 4
**Contribution:** 3
**Rating:** 2
**Confidence:** 4

**Summary:**

Decentralized training is an emerging field that makes LLM training more accessible by allowing research groups and volunteers to pool together available compute and, potentially, match the performance of centralized GPU clusters. This paper discusses the problem of making such system Byzantine-tolerant, i.e. robust to the presence of malicious participants that contribute incorrect results (e.g. to disrupt the training run or get the incentive from participation without actually contributing their compute).

While prior work addressed this problem in case of using data parallelism, it is crucial that modern LLMs have a large number of parameters and require some form of model parallelism to be trained. This paper addresses Byzantine tolerance in case of pipeline parallelism, which is known to be one of the most practical forms of model parallelism in decentralized setups due to its low bandwidth requirements.

The paper proposes to use EMA-based metrics to detect anomalies in activations and gradients passes through the pipeline stages. The authors prove that these metrics catch attacks that are large enough to significantly affect the validation loss. They also report experiments showing that this approach withstands multiple simple attacks that might be applied by malicious workers.

**Strengths:**

1. **Significance.** The paper discusses decentralized training, a promising approach to make LLM training accessible for small research labs, academic and individual researchers that don't have access to massive centralized clusters. The authors address the problem of Byzantine tolerance, which is known to be a major roadblock to adopting decentralized training.
2. **Originality.** The paper goes beyond most prior work and addresses Byzantine tolerance in case of using pipeline parallelism, which is known to be one of the most practical approaches to model parallelism for decentralized training systems due to its low bandwidth requirements.
3. **Practical solution.** Unlike prior work, the proposed solution doesn't require to allocate a substantial share of GPU compute to verification. Instead, it suggests to use cheap CPU nodes to detect anomalies in activations and gradients passes through the pipeline stages. The authors propose a straightforward way of integrating their method into existing decentralized training frameworks.
4. **Clarity.** The paper is well-written and describes the proposed algorithm in a clear way.
5. **Realistic training setup.** The authors report experiments with various simple attacks on a realistic distributed training setup.

**Weaknesses:**

1. **No results for adversarially designed attacks.** The authors only evaluate common generic attacks (L162-173, L480-481), such as sending constants, random values, or transformations of true activations/gradients. They don't evaluate adversarial attacks specifically designed to bypass the proposed method (e.g. by sending random data mimicking the tracked EMA-based metrics). It is difficult to infer bounds on their validation loss impact from the provided theoretical derivations.
2. **No results for medium-strength attacks.** Figure 1 features only strong attacks (F1 score > 0.8) that get caught and weak attacks (F1 score < 0.2) that don't impact training, with only one datapoint in between. This suggests that medium-strength attacks (F1 score ≈ 0.5) might still slip through and significantly impact the validation loss.
3. **Too strong, less realistic assumptions.** The paper assumes that malicious workers don't collude with each other (L161) and only perturb activations/gradients while sending them through pipeline stages, not during gradient aggregation (L321). It also assumes a small enough number of malicious nodes so that majority of workers holding each pipeline stage are honest with a high probabilty (L285).
4. **Accounting for gradient aggregation attacks.** While the authors claim that protecting from gradient aggregation attacks is a "complimentary axis" (L321), they don't discuss how to combine their method with protecting from such attacks, and how this affects the method's assumptions (e.g. for the number of malicious workers).

**Questions:**

1. How does the proposed method withstand specially designed adversarial attacks, e.g. if attackers send random data mimicking the tracked EMA-based metrics or use a small MLP instead of the proper pipeline stage to save compute?
2. What is the effect of medium-strength attacks (e.g. with F1 scores 0.3, 0.4, 0.5, 0.6) on the validation loss?
3. Given the effect of weak and medium-strength attacks, does decentralized training still make sense? (e.g. if we get the validation loss of a smaller model with 10x training compute, the participants might rather choose to train the smaller model locally)
4. How can we combine the proposed method with methods to protect from gradient aggregation attacks? How would this impact the maximum number of malicious workers the system can withstand?

---

> ### Author Response · Authors · 2025-11-21
> **Response to Reviewer gMeg**
>
> We want to thank the reviewer for their assessment of our work. We are glad that they appreciate the "significance of decentralized training", "originality of our work compared to prior research", "the practicality of our solution", and "the clear and excellent presentation of the work." Below, we aim to address the reviewer's comments on our work and provide clarification.
>
> -----------------------------------------------------------------------
>
> > **(W1) The authors only evaluate common generic attacks (L162-173, L480-481), such as sending constants, random values, or transformations of true activations/gradients. They don't evaluate adversarial attacks specifically designed to bypass the proposed method (e.g. by sending random data mimicking the tracked EMA-based metrics).**
>
> > **(Q1) How does the proposed method withstand specially designed adversarial attacks, e.g. if attackers send random data mimicking the tracked EMA-based metrics or use a small MLP instead of the proper pipeline stage to save compute?**
>
> We appreciate the opportunity to clarify the novelty of our attack suite. To the best of our knowledge, our benchmark is the first to adapt these attacks specifically to the pipeline parallel setting:
> 1. Our benchmark is the first of its kind in the pipeline parallel literature that constitutes such a variety of attacks. The only prior work that we are aware of, Lu et al. 2024, considers negation attacks which is one of the attacks that we also consider in our work. We not only consider seven different attacks, but also create a realistic mixed attack where each worker uses a randomly chosen attack type to send invalid activations/gradients (please see the highlighted section L409-419). These attacks are common baselines inspired by the DP Byzantine-tolerant literature that are being constantly evaluated against, e.g., please see Gorbunov 2022.
> 3. We considered adaptive attacks that are aware of the defense design. In the paper, we assume an adversary that keeps accumulating a momentum of values that they send to the next layer and aim to gradually drift towards a noisy signal that interrupts training. As pointed out in the highlighted section in L461-479 and Table 6, our method is robust against such adaptive adversaries.
>
> We share the reviewer's point on the limited number of work dedicated to designing attacks specific to the pipeline parallel setting. We hope our work would encourage researchers in the field of adversarial ML to start working on this important yet underexplored setting.
>
> -----------------------------------------------------------------------
>
> > **(W2) It is difficult to infer bounds on their validation loss impact from the provided theoretical derivations.**
>
> Our theorem was never intended to provide bounds on the validation loss. Providing such bounds on non-convex LLMs with billions of parameters is indeed challenging. The theorem is providing convergence guarantees to show that under relaxed assumptions, we would end up in a neighborhood of the optimal solution, and that the radius of this neighborhood is directly related to the attack perturbation magnitude. Such bounds are common practice in the non-convex optimization literature, e.g., please see Koloskova et al. 2024. Please let us know if further clarification is needed on Theorem 1.
>
> -----------------------------------------------------------------------
>
> > **(W3) Figure 1 features only strong attacks (F1 score > 0.8) that get caught and weak attacks (F1 score < 0.2) that don't impact training, with only one datapoint in between. This suggests that medium-strength attacks (F1 score ≈ 0.5) might still slip through and significantly impact the validation loss.**
>
> > **(Q2) What is the effect of medium-strength attacks (e.g. with F1 scores 0.3, 0.4, 0.5, 0.6) on the validation loss?**
>
> We wish to clarify that Figure 1 does **not** show a sweep over the F1-score, and we explicitly did **not** control the attack strength. Instead, these clusters emerged naturally from the simulation. In particular, we simulated all various attack types given in the paper (with varying strengths, please see the full details in Tables 12-14), and at the end of training, we recorded the detection F1-score and the validation loss. Then, we observed that two clusters were formed: one belonging to disruptive attacks that could potentially lead to divergence, which SENTINEL can detect with high accuracy. The other one belonged to attacks that were naturally harmless and could deviate the final validation loss only limitedly, which we sometimes did not detect due to their minimal impact. This is directly related to our convergence theorem which we extensively discuss in Appendix E.1.6. Please see the highlighted points in lines 1971-1974 and let us know if any further clarification is needed.

---

> > ### Author Response · Authors · 2025-11-21
> > **Response to Reviewer gMeg (cont.)**
> >
> > -----------------------------------------------------------------------
> >
> > > **(W4) Too strong, less realistic assumptions. The paper assumes that malicious workers don't collude with each other (L161) and only perturb activations/gradients while sending them through pipeline stages, not during gradient aggregation (L321). It also assumes a small enough number of malicious nodes so that the majority of workers holding each pipeline stage are honest with a high probability (L285).**
> >
> > We break down this point into several discussion points as we believe they are related to different parts of the paper.
> >
> > 1. *No collusion Assumption*: We considered the no collusion assumption to relax our threat model and theoretical analysis. **However, we considered collusion attacks in ablation study 2 (highlighted in L436-439 and Figure 3b) where multiple adversaries coordinate their attack time**. We would like to note that all our experiments assume that at least 15% of all attackers coordinately attack at the same time. There might be other sophisticated forms of collusion which we believe are interesting research directions of their own and beyond the scope of our work.
> > 2. *Signal Perturbation between Pipeline Stages*: As highlighted in Figure 2, and Table 5 & Q0 in Appendix A (L832-863, highlighted for your convenience), the focus of our work is on the underexplored pipeline parallel attacks. Parameter gradient attacks are the topic of federated learning in DP which have been explored extensively (please see Gorbunov et al. 2022, Karimireddy et al., and Malinovsky et al. 2024 for such literature). Indeed the interaction between these two complementary axes would be an interesting research direction (which we alluded to in our paper's limitations, see L534-539).
> > 3. *Honest Majority Lemma*: We believe that the content of our Lemma 2 has been misinterpreted. The assumption made throughout the paper is the presence of an honest majority (>50%). We did **not** assume a "small enough" number of malicious nodes beyond this standard Byzantine-tolerant constraint.
> > 	1. Lemma 2 is not to ensure a "small enough malicious number with high probability." *It is related to the interplay between worker assignment and the initial pool of workers*. In simple terms, since we do **not** have a prior knowledge of which worker is adversarial or not, if we assign them to each stage "uniformly at random," we want to formulate what would be the number of workers to ensure that "at each stage" we also have an honest majority (>50% honest workers).
> > 	2. In our experiments, we satisfy the stage-wise honest majority to get the "maximum number of malicious workers per stage". Our mixed attack experiments consider a 3:5 malicious to honest ratio (37.5% malicious) at each stage, which is the "maximum number of malicious workers" to ensure an honest majority. We would like to point out that our assumptions are well above average of what is usually considered in the Byzantine-tolerant literature (please see Table 8 for a concrete overview of malicious workers considered in previous work).

---

> ### Author Response · Authors · 2025-11-21
> **Response to Reviewer gMeg (cont.)**
>
> -----------------------------------------------------------------------
>
> > **(W5) While the authors claim that protecting from gradient aggregation attacks is a "complimentary axis" (L321), they don't discuss how to combine their method with protecting from such attacks.**
>
> > **(Q4) How can we combine the proposed method with methods to protect from gradient aggregation attacks? How would this impact the maximum number of malicious workers the system can withstand?**
>
> Since the DP attacks happen during all-reduce, defenses in this area are easily integrable with our method which operates on a completely separate PP axis. In detail, one needs to replace the regular all-reduce with robust aggregation techniques that have been proposed in the Byzantine-tolerant DP literature (e.g., TrimmedMean (Yin et al. 2018)). **This would have no impact on how SENTINEL operates as these two operate on complete distinct axes during the training.**
>
> To demonstrate that this is feasible, we train a Llama-3-0.6B model against mixed attacks. We utilize two robust aggregators instead of the vanilla all-reduce operation, namely Krum (Blanchard et al. 2017) and Bulyan (El-Mhamdi et al. 2018). As can be seen below, the performance of our method has not been impacted by the presence of a robust aggregation technique, substantiating our claims regarding the "complementary nature of DP and PP axes."
>
> | **Method**                 | **Pr. (%) ↑** | **Re. (%) ↑** | **F1 (%) ↑** | **Val. Loss (Attacked) ↓** | **Val. Loss (Vanilla) ↓** |
> |---------------------------|---------------|---------------|--------------|-----------------------------|-----------------------------|
> | **SENTINEL + Krum**       | 93.6          | 80.6          | 86.6         | 3.873                       | 3.855                       |
> | **SENTINEL + Bulyan**     | 85.3          | 80.6          | 82.9         | 3.883                       | 3.855                       |
>
>
> **As shown in the table, the difference in validation loss between the attacked and vanilla settings is negligible (<0.5%), confirming that our method remains effective when combined with robust aggregation**. We have added these results to the paper in L480-488 and L2721-2732 in the Appendix.
>
> -----------------------------------------------------------------------
>
> > **(Q3) Given the effect of weak and medium-strength attacks, does decentralized training still make sense? (e.g. if we get the validation loss of a smaller model with 10x training compute, the participants might rather choose to train the smaller model locally)**
>
> We believe this points to the reviewer's earlier point on Figure 1 which we replied to earlier (see W3).
>
> As a side note, decentralized training has shown great promise to train LLMs over internet-grade connections. The presence of adversaries should not discourage us from stepping into this direction, but instead we should dedicate research in this direction to make it feasible. This is because decentralized training can open the door to training large, useful models outside big corporations. Indeed there are security challenges, but once necessary measures are put in place, they can deter malicious actors and provide access to honest workers that aim to participate in training such models. As pointed out by Yuan et al. 2022: "If we could make use of these [idle] devices in a decentralized open-volunteering paradigm for foundation model training, this would be a **revolutionary alternative** to the expensive solutions offered by data centers." We hope that our work would encourage active research in this exciting and rewarding area. Please see our Q1 in Appendix A for more on this point.
>
> -----------------------------------------------------------------------
>
> We hope our response provides sufficient clarifications around the scope of our attack taxonomy, theoretical bounds, and SENTINEL's compatibility with existing DP solutions. We hope these clarifications allow the reviewer to reconsider their assessment. We remain available if any further explanation is needed to fully address their concerns.

---

> > ### Author Response · Authors · 2025-11-21
> > **References**
> >
> > Blanchard et al. "Machine Learning with Adversaries: Byzantine Tolerant Gradient Descent." *NeurIPS*. 2017.
> >
> > El-Mhamdi et al. "The Hidden Vulnerability of Distributed Learning in Byzantium." *ICML*. 2018.
> >
> > Yin et al. "Byzantine-robust distributed learning: Towards optimal statistical rates." _ICML_. 2018.
> >
> > Yuan et al. "Decentralized training of foundation models in heterogeneous environments." *NeurIPS*. 2022.
> >
> > Gorbunov et al. "Secure distributed training at scale." *ICML*, 2022.
> >
> > Karimireddy et al. "Byzantine-robust learning on heterogeneous datasets via bucketing." *ICLR*. 2022.
> >
> > Malinovsky et al. "Byzantine robustness and partial participation can be achieved at once: Just clip gradient differences." *NeurIPS*. 2024.
> >
> > Lu et al. "Position: Exploring the robustness of pipeline-parallelism-based decentralized training." *ICML*. 2024.
> >
> > Koloskova et al. "On convergence of incremental gradient for non-convex smooth functions." *ICML*. 2024.

---

> > > ### Author Response · Authors · 2025-11-27
> > > **Rebuttal Follow-up**
> > >
> > > Dear Reviewer gMeg,
> > >
> > > We are writing to kindly follow up on our rebuttal response.
> > >
> > > We have provided a detailed discussion clarifying the points raised in the review including the theoretical findings of our work and the specific scope of our attack taxonomy. Additionally, we included new results on combining SENTINEL with robust aggregation to address your question regarding gradient protection compatibility with our method.
> > >
> > > We hope our response resolve the concerns raised, and we look forward to your feedback. We would appreciate it if you could reassess our work in light of our clarifications, and are more than happy to provide additional clarification should any issues remain unresolved.

---

### Author Response · Authors · 2025-11-21
**General Response**

We would like to thank the reviewers and the area chair for their academic service and feedback on our work. We are glad that the reviewers:

- *found our setting interesting* as we are addressing an underexplored yet important security issue within pipeline parallel-based decentralized training (reviewers gMeg, Byyi, bjpd, and ux6h),
- appreciate our *lightweight and practical verification* approach (reviewers gMeg and bjpd)
- value our *comprehensive attack taxonomy* and evaluation (reviewers bjpd and ux6h),
- appreciate the work that went into *real-world integration of SENTINEL with SWARM* for decentralized training (reviewers gMeg, Byyi, and ux6h)
- found our paper presentation *clear and well-written* (reviewers gMeg, Byyi, and ux6h)

-----------------------------------------------------------------------

## **Summary of Contributions**

We want to summarize key contributions of our paper:

- A suite of adversarial threats for training-interruption attacks with **more than 7 attacks**. To the best of our knowledge, the only prior work on pipeline parallel based attacks only considered negation and scale attacks (Lu et al. 2024)
- **Identifying cascading effect** within pipeline parallel attacks and formalizing this concept
- A practical, **lightweight verification** solution using EMA and IQR thresholds with adaptive thresholding that adjust itself based on changing training dynamics
- **Theoretical analysis** of the core method:
	1. Convergence analysis in the presence of training-interruption attacks demonstrating that small, undetected adversaries have limited impact on network convergence in the non-convex setting (formal statement: Theorem 3 in Appendix E.1.4)
	2. Showing that the above would recover well-known convergence bounds when the malicious actors are removed (Appendix E.1.5)
	3. Analyzing the relationship between the honest majority in the initial pool of workers and conditions under which random worker assignment to each stage would result in an honest majority (>50%) (formal statement: Lemma 1 in Appendix E.2)
-  A comprehensive, step-by-step adaptation guide for **integrating SENTINEL into SWARM** (Appendix G)
- **Comprehensive evaluation benchmark** involving (Section 5 and Appendices F & G)
	- Three different datasets (C4, FineWeb, OpenWebText)
	- Two different architectures (Llama-3 and NanoGPT) and adding two more MoEs (Llama-4 and DeepSeek-V3)
	- Seven initial activation AND activation gradient attacks + mixed attacks + adaptive EMA attacks
	- Ablation study on the impact of "warm-up", "collusion", "delay", "EMA", "distance metrics", and "randomness" (first three in the main paper Section 5 and the second three in Appendix G.3)
	- Real-world experiment on a fully functional decentralized training environment using SWARM involving *128 AWS GPU separate instances* showcasing the novel integration of SENTINEL into SWARM

-----------------------------------------------------------------------

## **Revised Version**

Following the discussions and reviewers' requests, we have made the following changes to the manuscript:

- Moved our detailed discussion on our distance metrics and how they are combined (in response to reviewer ux6h) to the main paper
- Added discussions on the worker availability in real-world eco-systems to our set of FAQs in Appendix A (following questions from bjpd and ux6h)
- Added a clear visualization on the difference between fixed mesh DP+PP vs. stochastic DP+PP (a.k.a. SWARM) to Appendix G.3.1 and discussions around this
- Added the following additional experiments:
	- Ablation study on the sensitivity to $\beta$ (requested by bjpd and ux6h)
	- A large-scale experiment on Llama-3 with 4B parameters (requested by bjpd)
	- MoE experiments on two different architectures, Llama-4-0.4B and DeepSeekV3-1B (requested by reviewer bjpd)
	- Impact of adding robust aggregation alongside SENTINEL (requested by gMeg and bjpd)

> LEGEND: we use the **color red** for any material (including tables and figures) added to this revision or moved from Appendix to main paper. We also use the **color blue** to highlight the questions that were answered in the original version of the paper.

Below, we address all reviewers’ comments and concerns. We hope this clarifies any misunderstandings. We remain available and committed to resolving any further points to ensure a thorough evaluation of our work.

### References

Lu et al. "Position: Exploring the robustness of pipeline-parallelism-based decentralized training." *ICML*. 2024.

---

### Meta-Review · Area_Chair_33Eq · 2026-01-07

**Summary:**

The author has tried to address the reviewer's concern about the relatively small model scale (increasing to 3B, which is still relatively small), ablation of the proposed approach; however, some of the concerns are not fully addressed, e.g., threat-model breadth limitations (data poisoning and Sybil attacks) are not sufficiently discussed. This is a really marginal case; I have decided to reject the paper.

**Reviewer Concerns:**

- The abolition study is helpful.
- The model scale has been increased, but the 4B model is still small.

**Reviewer Scores:**

I tend to believe the reviewers would keep their original score.

---

### Decision · Program_Chairs · 2026-01-26

Reject